# L2MAC: Large Language Model Automatic Computer for Extensive Code Generation

**Samuel Holt**
University of Cambridge
sih31@cam.ac.uk

**Max Ruiz Luyten**
University of Cambridge
mr971@cam.ac.uk

**Mihaela van der Schaar**
University of Cambridge
mv472@cam.ac.uk

## ABSTRACT

Transformer-based large language models (LLMs) are constrained by the fixed context window of the underlying transformer architecture, hindering their ability to produce long and coherent outputs. Memory-augmented LLMs are a promising solution, but current approaches cannot handle long output generation tasks since they (1) only focus on reading memory and reduce its evolution to the concatenation of new memories or (2) use very specialized memories that cannot adapt to other domains. This paper presents L2MAC[1], the first practical LLM-based general-purpose stored-program automatic computer (von Neumann architecture) framework, an LLM-based multi-agent system, for long and consistent output generation. Its memory has two components: the instruction registry, which is populated with a prompt program to solve the user-given task, and a file store, which will contain the final and intermediate outputs. Each instruction in turn is executed by a separate LLM agent, whose context is managed by a control unit capable of precise memory reading and writing to ensure effective interaction with the entire file store. These components enable L2MAC to generate extensive outputs, bypassing the constraints of the finite context window while producing outputs that fulfill a complex user-specified task. We empirically demonstrate that L2MAC achieves state-of-the-art performance in generating large codebases for system design tasks, significantly outperforming other coding methods in implementing the detailed user-specified task; we show that L2MAC works for general-purpose extensive text-based tasks, such as writing an entire book; and we provide valuable insights into L2MAC's performance improvement over existing methods.

## 1 INTRODUCTION

Transformer-based Large Language Models (LLMs), such as GPT-3 (Brown et al., 2020), Instruct GPT (Ouyang et al., 2022), and the most recent GPT-4 (OpenAI, 2023), have achieved unprecedented success in generating high-quality user-directed text. Despite their impressive capabilities, these models are inherently restricted by a fixed *context window* of size $c$, which limits the number of tokens and, consequently, the characters they can process. This limitation manifests itself critically in tasks that require the generation of extensive and logically coherent output structures, such as the generation of large codebases. Therefore, although existing LLMs excel in generating short isolated code snippets (Chen et al., 2021), they struggle to produce codebases that are both extensive and internally consistent due to the information loss and contextual truncation imposed by the finite context window. But practical code is rarely a short snippet, so this limitation undermines the applicability of LLMs to solving real-world tasks that require the dynamic integration of multiple outputs and attendance to distant information.

A natural solution is to extend the LLM agent with an external memory store. However, existing methods primarily serve to extend the implicit knowledge of LLMs through an external corpus (Zhong et al., 2022), to facilitate long conversational or document summarization tasks (Liang et al., 2023), or to maintain precise values for variables (Hu et al., 2023). In the latter case, the approaches simply interface the LLM with a dictionary or a database, which does not adjust to other tasks. In other settings, current approaches adopt simplistic memory management strategies that append new

---

[1]Full code at https://github.com/samholt/L2MAC.

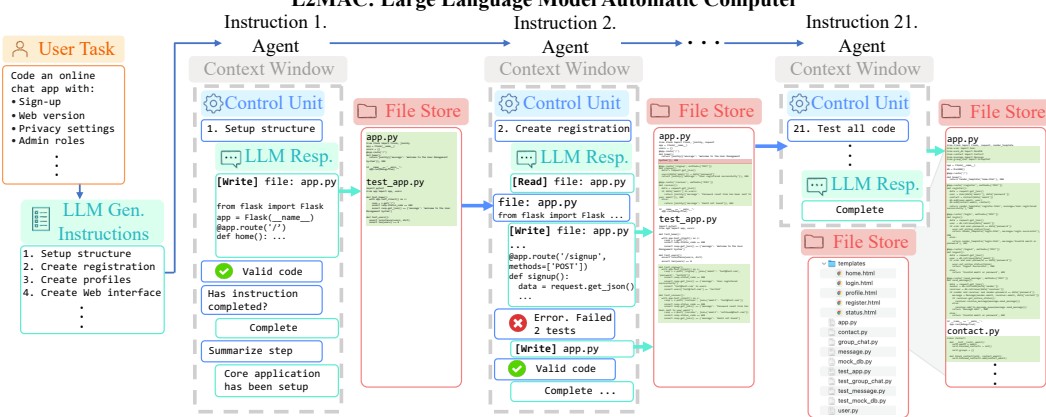

Figure 1: **L2MAC Overview**. Code-L2MAC is an instantiation of the LLM automatic computer (L2MAC) framework, an LLM-based multi-agent system, here for extensive code generation. First, it breaks down the user task into sequential instructions $\mathcal{I}$. The Control Unit (CU) manages the LLM's context window for each instruction (agent) and interacts with the external memory file store through read, write, and evaluate tools. It identifies and reads relevant files from the memory to generate or update files per instruction (P2). This ensures proper conditioning of existing files without losing vital context (P1). Automatic checks evaluate the LLM's outputs for correctness and completion (P3), with iterative error corrections involving both code syntactical checks of the code and running self-generated unit tests to check desired functionality. Overall, this produces a complete large codebase that fulfills the detailed user task in the file store. See Figure 5 for an expanded overview.

information sequentially without any provision for in-place modification. This makes any error essentially irreversible, a critical limitation when generating codebases, and restricts the possibility of adapting previously generated code as the task progresses. Compounding these shortcomings, these methods do not include mechanisms for maintaining syntactic or semantic consistency within the memory store, a vital requirement for the generation of coherent and interdependent code structures. Thus, existing methods are ill-suited for extensive large code-generation tasks.

An effective method for extensive code generation requires the following three core properties:
**(P1) Task-Oriented Context Management:** At each computational step, the context for the LLM agent should contain the information required to complete the current instruction. The context is dynamically managed to prevent exceeding the fixed context window size of tokens.
**(P2) Precise Read/Write Tools for Entire Memory:** The LLM agent should possess read/write operations that can interact precisely with the memory store to *fetch* and *update* relevant files.
**(P3) Checking the Generated Output:** The outputs of the LLM agent are checked for both mistakes and when the current instruction has been completed. When mistakes are detected, such as syntactically invalid code, or failing self-generated functional unit tests, they can attempt to be fixed by iterating the discovered errors with the LLM agent.

With these considerations, we introduce L2MAC, the first practical LLM-based general-purpose stored-program computer (von Neumann architecture) framework, an LLM-based multi-agent system, and instantiate it for long code generation tasks as Code-L2MAC. A Control Unit (CU) orchestrates the execution of the individual LLM agents and their interaction with the memory store, thus satisfying the three stipulated properties. As outlined in Figure 1, an LLM agent first generates a task-oriented instruction list $\mathcal{I}$ from a detailed user-specified task. The CU tailors the LLM agent's context (P1), so that it always includes the next unresolved instruction in $\mathcal{I}$ and information about the execution of past iterations (agents), and declutters the context when approaching its limit. It also endows the LLM agent with the ability to read and update any existing region of the memory store or extend it with new outputs (P2). Furthermore, the CU plays a crucial role in checking the generated output (P3). It feeds the LLM agent with syntactical checker errors and requests the LLM agent to generate checks alongside generating output, here unit tests when generating code, which are verified at each update of the memory file store to trigger corrective actions if needed, thereby ensuring that the extensive output in memory is both syntactically and functionally consistent.

**Contributions:** ① We introduce the L2MAC framework, the first practical LLM-based general-purpose stored-program automatic computer (von Neumann architecture) framework, an LLM-based multi-agent system, for long output generation tasks. ② We provide a practical implementation of this framework for code generation tasks called Code-L2MAC. This uses a Control Unit to control the input and output of the LLM and the use of entire memory file store read/write tools and highlights

that code checks are key to correcting generated code that has syntactic and functional errors within whilst conditioning on and integrating with the existing codebase. ③ We empirically validate Code-L2MAC, demonstrating its state-of-the-art capabilities in generating large codebases to solve for the first time high-level system design tasks of creating entire applications and for generating code achieving a 90.2% Pass@1 score on the HumanEval benchmark (Chen et al., 2021). Also, we show L2MAC works for general-purpose extensive text-based tasks, such as writing an entire book. Additionally, we gain insight and understanding of how Code-L2MAC can leverage tools to execute and verify code by self-generating its own unit tests and use these to correct for generation errors whilst generating large interrelated code structures that fulfill detailed complex user-specified task feature requirements.

## 2 BACKGROUND

**Large Language Models (LLMs).** Essentially, an LLM is a probabilistic function $l : \Sigma^c \to \mathcal{P}(\Sigma)$, ingesting a list of elements of the alphabet $\Sigma$ of lengths $c$ and outputting a distribution over the elements from which one is drawn (Vaswani et al., 2017). We refer to $c$ as the length of the context. For example, GPT-4 and GPT-3 have fixed context windows of $8,192$ and $4,097$, respectively.

Consequently, as discussed in Schuurmans (2023), an LLM at inference time can be formalized as a finite-state machine (Dolfing & Hetherington, 2001). Given the well-known and exceptional extension from finite-state machines to Turing machines Turing et al. (1936) that leads to the practical framework of stored-program computers, it is natural to envision a parallel extension for LLMs.

**Stored-Program Computer (SPC).** Originally termed the von Neumann architecture (Von Neumann, 1945), is a combination of a processor (arithmetic and logic unit), a Control Unit (CU), and memory capable of storing both instructions and data, SPCs set the foundation for modern computer design. Where the control unit manages the interaction between the processor and the data. A fundamental property of a general-purpose SPC is that it can be reprogrammed to automatically execute a program (ideally to solve a specified task) without manual intervention[2]. Specifically, the CU extracts from memory, data and instructions and correspondingly sets the input and state of the processor (P1) and then overwrites a memory register with the output (Appendix A). Thus, the SPC extends the processor's arithmetic and logic abilities with the ability to manipulate memory (P2). An often overlooked but vital function of the CU is error-checking of the processor's output (P3). "Stochastic" effects of temperature, voltage change, or radiation can cause errors in a processor (Nicolaidis, 2010), including misspecified inputs leading to overflows. Therefore, CUs implement output checks such as parity bit checks and usually corresponding error-correcting mechanisms (Harris & Harris, 2010).

## 3 L2MAC FRAMEWORK

Now we outline the L2MAC framework for the first practical LLM-based SPC, with an instantiation for coding illustrated in Figure 1. L2MAC consists of three main components: the LLM processor, the memory file store, and the Control Unit (CU) that controls the flow of the execution, thus endowing the LLM agent with read-and-write capabilities, among other capacities—this is illustrated in Figure 2. Some choices are deliberately left open to differentiate the key functionalities needed from how we tackle them in our implementation for code generation, which we detail in the next section.

### 3.1 LLM-BASED PROCESSOR

An inherent element of L2MAC is the Large Language Model (LLM), which is responsible for the actual generation of the output for the task. An LLM can be visualized as a more complex atomic unit of computation $f : \Sigma^c \to \mathcal{P}(\Sigma)$, where, for instance, $|\Sigma| = 50,257$ tokens (Radford et al., 2019)[3]; rather than being restricted to only deterministic arithmetic and logical operations on binary sequences, that is, $f : \{0,1\}^{64} \times \{0,1\}^{64} \to \{0,1\}^{64}$, e.g., for a standard 64-bit processor (Hennessy & Patterson, 2011). This allows for a more flexible and powerful (yet more expensive) computation unit, which can be used to solve a different range of tasks.

---

[2]As a side note, the Church-Turing Thesis posits that both Universal Turing Machines (UTMs) and SPCs are capable of simulating each other and, therefore, are models of the same class of computational processes Von Neumann (1945) (Appendix A.1). Recently, it has been shown that augmenting an LLM with an external read-and-write memory allows it to simulate a UTM (Schuurmans, 2023). However, we do not aim to reproduce the capabilities of the current hardware SPC in arithmetic and logic operations but rather explore the powerful abilities of LLM-SPCs as automatic computers.

[3]The token vocabulary size is an implementation detail of the underlying LLM used.

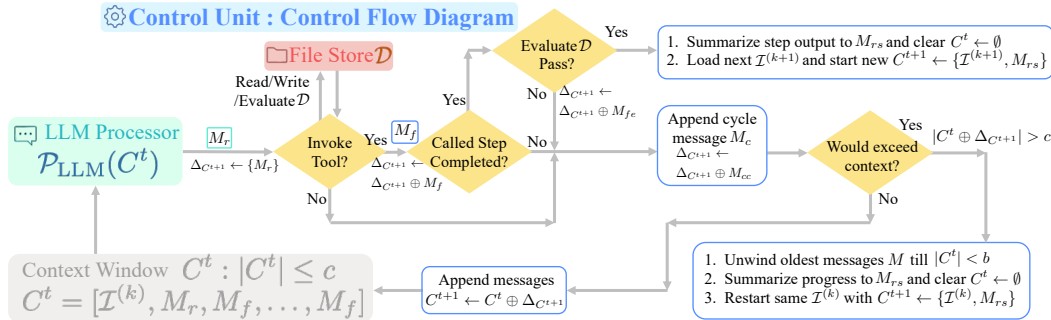

Figure 2: **Control Unit—Control flow diagram for one dialog turn** $t$. Here this executes one current instruction $\mathcal{I}^{(k)}$. It starts by loading the first instruction into the context window $C^0 \leftarrow \{\mathcal{I}^{(0)}\}$ and iterates it automatically until all instructions in $\mathcal{I}$ have been executed. First, $C^t$ is processed by the LLM Processor $\mathcal{P}_{\text{LLM}}(C^t)$ to output $M_r$. The CU stores this in a buffer $\Delta_{C^{t+1}} \leftarrow \{M_r\}$, and checks if $M_r$ has called a tool, and if so, it executes the tool with the specified input in $M_r$, which includes reading, writing and evaluating $\mathcal{E}(D)$ the file store $\mathcal{D}$—outputting $M_f$, which is appended to the buffer $\Delta_{C^{t+1}}$. The CU performs additional control flow as detailed in Section 3.3, for checking if an instruction has been completed, continuing an instruction beyond the context window (P1), and continuing executing the current instruction.

Leveraging an LLM within the L2MAC framework offers distinct advantages to exploit and challenges to overcome, categorized to the properties of P1, P2, and P3. Here, L2MAC benefits from the LLM's *awareness of external tools* with which it can interact with assisted by the Control Unit. This capability enables requests for memory reads/writes (P2) and additional output checks (P3).

In contrast, the use of an LLM also imposes constraints that need to be addressed, such as the impediments of a limited context window to prevent incongruencies caused by the lack of attention to distant information; that is, we have to handle context correctly (P1). Furthermore, the *stochastic nature of LLM's output* ($\mathcal{P}(\Sigma)$) is a significant risk in situations where precision and correctness are key. Thus, crucial to effectively updating an interrelated memory is the ability *to enforce periodic checks* on its output to ensure a level of correctness and consistency (P3) (Liventsev et al., 2023).

## 3.2 MEMORY

Following an SPC, we distinguish between two types of memory, that of the *instruction registry* $\mathcal{I}$ and the *file store* $\mathcal{D}$. On the one hand, the *instruction registry* $\mathcal{I}$ contains the prompt program that will be used to determine the state of the processor. In L2MAC, given the LLM processor, this corresponds mainly to the strings that will be incorporated into the context of the LLM agent at each execution. In a basic implementation, $\mathcal{I}$ lists the sequential steps needed to complete the task, either specified by the user or automatically generated—where each step will be executed by a separate LLM agent. On the other hand, the *file store* $\mathcal{D}$ stores the rest of the information relevant for the processor to read, write, and evaluate, with the final output ultimately stored in $\mathcal{D}$.

## 3.3 CONTROL UNIT

The control unit (CU, cf. Figure 2) is responsible for managing the context window for the LLM, encompassing both its inputs and outputs, executing the LLM, checking its outputs for errors, and enabling it to call tools (functions)—including reading and writing. We further detail the CU, including a block diagram figure and pseudocode for its operation in Appendix B. First, we start with how the CU interacts with the LLM.

### 3.3.1 TASK-ORIENTED CONTEXT MANAGEMENT (P1)

**Context formalism**. The CU uses the LLM as a multi-turn dialog system, filling its context window $C$ with a combination of messages $m$ which can come from the user $M_u$, an LLM response $M_r$, a function (tool) output $M_f$, or the CU $M_c$, so that $m \in \{M_u, M_r, M_f, M_c\}$. Consequently, at turn $t$ then the context window[4] $C^t \in \text{List}(M)$ is of the form $C^t = (m^1, m^2, \ldots, m^{n_t})$.

To make L2MAC an automatic computer[5], the CU prompts the LLM to fill the initially empty instruction registry $\mathcal{I}$ with a list of instructions $\{\mathcal{I}^{(1)}, \ldots, \mathcal{I}^{(K)}\}$ where each will be executed in

---

[4]Given a set $A$, let $\text{List}(A) = \{(a^i)_{i=1}^n, a^i \in A \,\forall i \in [n], n \in \mathbb{N}\}$ denote the set of lists of arbitrary length composed of elements in $A$.

[5]We define what we mean by an *'automatic'* computer in Appendix A.2.

the LLM processor[6]. L2MAC then loads an empty context window of an LLM agent with the first instruction $C^0 \leftarrow \{\mathcal{I}^{(0)}\}$ and iterates the CU control flow loop (Figure 2) until all instructions have been achieved. The LLM can signal when the current instruction $\mathcal{I}^{(i)}$ has been completed through calling a special function 'step_complete' at which point the CU evaluates the file store $\mathcal{D}$ using its evaluator module $\mathcal{E}$ (discussed in Section 3.3.3) to check for any introduced errors. If none are found, it asks the LLM to summarize the generated output in the current context window $C^t$ as a message $M_{rs}$ and resets the context window as $C^{t+1} \leftarrow \{\mathcal{I}^{(k+1)}, M_{rs}\}$.

**Overcoming the fixed context window constraint**. The input to the LLM cannot exceed the context window constraint $c$: the combined length of the initial context $C^t$ and the additional messages buffer $\Delta_{C^{t+1}} = \{m^0, \dots, m^n\}$ must fit in the context window, that is[7], $|C^t \oplus \Delta_{C^{t+1}}| \leq c$. However, the length of $\Delta_{C^{t+1}}$ is not known a priori, so the CU should have a way of handling the cases where $\Delta_{C^{t+1}}$ exceeds the context margin $c - |C^t|$. This can be achieved through a combination of three different strategies: (1) minimize the occurrence by promoting the task at each time step to be small enough and economizing the filling of the context $C$; and if the situation occurs, (2) store in the file store $\mathcal{D}$ as much relevant output as possible from the current $C^t$ and (3) update or include a new summary message with $\mathcal{I}^{(k)}$ as in-context tuning for the next iteration.

Regarding (1), through appropriate crafting $C^t$, the CU can prompt the LLM to plan sub-steps for the current instruction (most likely the original task prompt given by the user) and then target each sub-step in the following iterations. For illustration, in a coding setting, (2) can be achieved by storing the generated code so far to avoid rewriting it in the next iteration, and (3) by initializing a new prompt with a summary $M_{rs}$ of the current progress and helpful information to complete the current instruction, e.g., which files should be read or modified, or the current progress made fixing errors—(3) is further detailed at the bottom right of Figure 2.

### 3.3.2 PRECISE READ/WRITE TOOLS FOR ENTIRE MEMORY (P2)

The need for a reading mechanism that retrieves the relevant information at each iteration is evident and has been reasonably explored in previous literature. In contrast, previous work on memory (as shown in Section 5) has paid little attention to the writing component, which gets mostly reduced to the appending of new prompts and LLM outputs (Liang et al., 2023; Zhong et al., 2022; Cheng et al., 2023; Wu et al., 2022) or updating the values of very structured and thus restrictive forms of memory (Modarressi et al., 2023), e.g., variables or tables (Hu et al., 2023).

These approaches make sense for summarization, dialogs, and database manipulation tasks but are not suitable for long interconnected output generation tasks, such as generating large codebases for system design tasks. Indeed, in such settings, the possibility of downstream subtasks $\mathcal{I}^{(j)}$ demanding extensions of previous outputs (such as modules in a codebase) due to imperfect planning, plus the non-determinism and possible hallucination of LLMs, make it probable to require modifications of previously stored memories $\mathcal{D}$ to rectify these defects, as shown in Section 6.2.

In L2MAC it is thus key to implement read/write interactions with any part of the memory. We want the agent to be able to scan on demand $\mathcal{D}$, retrieve parts of the memory that it considers relevant, and potentially update them. In the next Section 4, we detail our implementation of an LLM with a write component that allows it not only to add new information to $\mathcal{D}$ but also to delete and update any of its contents, an essential element that allows L2MAC to succeed in long output generation tasks.

### 3.3.3 CHECKING THE GENERATED OUTPUT (P3)

As discussed in Section 3.1 and 3.3.2, the intrinsic stochasticity of LLMs and the well-known phenomenon of hallucination (OpenAI, 2023) make it likely that incoherent or erroneous outputs occur during long interactions, which can be disastrous, for example, in coding. More profoundly, changes (e.g., to a function) to satisfy a given instruction $\mathcal{I}^{(j)}$ can hamper the solution to formerly completed instructions $\mathcal{I}^{(i)}$, $i < j$. Therefore, it is essential to incorporate two key checks, one to check the generated outputs for errors using a given evaluator module $\mathcal{E}$, and the other to check when the current instruction has been completed in the current context $C^t$ (c.f. top diamond in Figure 2).

---

[6]We consider the simplest case of sequential instructions of a prompt program and discuss more complicated control flow paradigms in Appendix I.

[7]We use $\oplus : \text{List}(A) \times \text{List}(A) \rightarrow \text{List}(A)$ as the concatenation of two lists on the set $A$. We abuse notation by considering any $a \in A$ as a singleton $\{a\}$.

**Error checking and error correction**. Using a given evaluator module $\mathcal{E}$, which can process the existing file store $\mathcal{D}$, i.e., $\mathcal{E}(D)$, allows when run, errors to be detected and returned to the LLM as an evaluator message $M_{fe}$. The evaluator is domain-specific; for example, in coding tasks, this can correspond to syntactical code checkers or self-generated unit tests that verify the correctness of the output for an instruction $\mathcal{I}^{(i)}$. Crucially, these self-generated unit tests also help test the existing functionality of previously generated instructions $\mathcal{I}^{(j)}$. Naturally, evaluation checks should be enforced on the file store $\mathcal{D}$ after each writing operation to ensure that new additions are correct and consistent with previous files. These result in messages $M_{fe}$ that are provided for in-context learning so that the LLM can correct the errors, $\Delta_{C^{t+1}} \leftarrow \Delta_{C^{t+1}} \oplus M_{fe}$, and iterate by rewriting $\mathcal{D}$ until the evaluator checks pass, if any are present.

**Checking for current instruction completion**. To ensure continued execution in a multi-turn dialogue LLM system until completion, we request the LLM to decide on the next step to take, which can involve executing a tool (Wang et al., 2023a). This is achieved through a cycle prompt message $M_{cc}$ that also asks the LLM if the instruction has been completed. Cycle prompting is necessary to account for different instructions requiring a variable number of turns to complete, and to protect against hallucinations where the LLM agent only discusses the instruction $\mathcal{I}^{(i)}$ and does not generate a solution or store it in memory. Overall, this ensures that the LLM provides a solution for the given instruction within the current context (P3).

## 4    CODE-L2MAC

Now, we use our LLM automatic computer (L2MAC) framework and instantiate it to complete large codebase generation tasks, which we call **Code-L2MAC**. We distinguish the general-purpose task long-generation framework from the code instantiation to detach the core components from task domain decisions that can be appropriately adapted to other task domains. We provide the full details of the implementation in Appendix C; yet here we highlight some notable design decisions on the memory layout (in particular, $\mathcal{D}$) and the read logic. There are different potentially valid alternatives for the read component in L2MAC (e.g., Wu et al. (2022)). However, to promote a *transparent read component*, we make the following choices.

We specify the memory file store $\mathcal{D}$, be composed solely of files, each shorter than the residual margin of the context window after accounting for preliminary messages in any iteration[8] and instruct the LLM agent to assign each file a semantically meaningful descriptive path. For example, a file name `models/user.py` suggests that it contains the data model for the user class. This choice is not only valuable for a human reader but also crucial for our Read-and-Write implementation, as it allows the LLM to infer the content of existing files $\mathcal{D}$ and prioritize those that it might need to read to complete its current instruction. Given the list of file paths, the LLM can request that the contents of any be appended into its context $C^t$. Although the path-to-content link is not absolute, empirically, this read implementation can perform well (Section 6), when coupled with the context management logic of the CU. Specifically, if the LLM reads the content of certain files and reaches the context window limit, it can include in its subsequent iteration summary indications of which files should be read and which should be excluded in order to generate the code necessary to complete the current instruction. This approach thereby enables the LLM to systematically scan all memory, with the scanning order guided by priorities previously established from the file path names.

## 5    RELATED WORK

In the following, we review the existing memory-augmented LLM-based approaches, focusing on approaches applicable to completing large-generation coding tasks—and summarize their main differences in Table 1. We provide an extended discussion of additional related works, including those of other single-turn code generation methods and memory-augmented LLMs in Appendix D.

**Single LLMs**. These are limited to generating an output of the maximum size of the context window and form the most popular baseline widely used for coding. Examples include Codex (Chen et al., 2021), GPT4 (OpenAI, 2023), GPT-Engineer (Osika, 2023) and Code LLama (Rozière et al., 2023).

**Reflecting LLMs**. These enhance the performance of LLMs by having the LLM reflect retrospectively on the output of an evaluator and the actions taken (Shinn et al., 2023; Madaan et al., 2023). These verbal reflections are used as in-context learning (Dong et al., 2022) to improve performance when attempting the task again. By using a validator, such as self-generated unit tests, this simple idea has

---

[8]More precisely, no longer than half such margin to ensure the LLM can rewrite any file within its context.

Table 1: Comparison of related works. Columns: *Context Management (P1)*—can it generate a larger output than $c$, and if so, can it access the necessary files to generate the next file correctly without knowledge of the file falling out of context? *Precise Read/Write for Entire Memory (P2)*—can it write to the full memory and update it in any order? *Checking the Generated Output (P3)?*—can it check for mistakes in the output? References:[1](Richards, 2023)[2](Shrestha & Watkins, 2023)[3](Nakajima, 2023)[4](Chen et al., 2021)[5](Osika, 2023)[6](Shinn et al., 2023)[7](Madaan et al., 2023).

| Approach | Methods | Context Management (P1)? | Precise Read/Write for Entire Memory (P2)? | Checking the Generated Output (P3)? |
|---|---|---|---|---|
| Autonomous Agent LLMs with memory | [1,2,3] | ✗—forgets which files were generated | ✓ | ✗—No checks |
| Single LLMs | [4,5] | ✗—limited to $c$ | ✗ | ✗ |
| Reflecting LLMs | [6,7] | ✗ | ✗—limited to append-only read always memory | ✓ |
| Code-L2MAC | **This work** | ✓ | ✓ | ✓ |

provided state-of-the-art small code snippet generation performance (Shinn et al., 2023). Still, the output of these approaches is also confined to the context window, whereas Code-L2MAC uses an evaluator not only for in-context learning but also for consistency of the codebase.

**Memory-augmented LLMs**. Most work on this topic focuses on long conversational/text summarization tasks (Liang et al., 2023; Zhong et al., 2022; Cheng et al., 2023), or mathematical/database reasoning (Hu et al., 2023). Methods in the first group usually focus exclusively on the read component of memory. The same happens with (Wu et al., 2022), which, through $k$-NN retrieval for next-token predictions, enhances perplexity for multiple tasks, including coding. However, read-only implementations are sensitive to the inconsistency and "stochasticity" of LLMs, which might break code, that the method cannot then fix. Modarressi et al. (2023) proposes updating memories, but they use tuples with two subjects and their relation, so they have no clear application to coding tasks. Similarly, the memory of mathematical/database reasoning methods use a dictionary of values or a structured database, which again is not applicable to different settings (e.g., coding).

**Autonomous agent LLMs with memory**. Richards (2023); Shrestha & Watkins (2023); Nakajima (2023) formulate a fully *autonomous* LLM-based agent in a reinforcement learning environment to complete a given task (Wang et al., 2023a). They build and update their own plans from a few user-given high-level goals. When coding, these agents reprogram and reformulate their step plan at runtime without safeguarding the original intentions, resulting in frequent deviations from the task (Wang et al., 2023a; Sato et al., 2023). In comparison, Code-L2MAC is designed to be *automatic*, i.e., it fulfills the detailed user-specified task without altering or forgetting parts. The only existing autonomous agent LLM with applicable memory capable of writing, reading, and executing code is AutoGPT (Richards, 2023). It tackles the context window constraint by iteratively summarizing previous actions completed at every $k$-th turn into a rolling summary kept in the context. Like other agents, the first step of AutoGPT is to summarize the user-specified task into a name, role, and five one-sentence goals for the agent to complete; this summary is kept in context throughout execution. AutoGPT and associated methods have two key limitations compared to Code-L2MAC (cf. Figure 3): (1) they compress the user-specified task into a mere six-sentence description, and (2) they compress the previous action history, thus losing crucial information. In code generation tasks, (2) indicates forgetting which files exist and their content, which is compounded by continual re-planning.

## 6 EXPERIMENTS AND EVALUATION

In this section, we evaluate Code-L2MAC and verify that it significantly outperforms state-of-the-art methods for generating code and large codebases for system design tasks. Due to the absence of an existing benchmark for large codebase generation tasks, we introduce one to enable comparison with other methods. Furthermore, there are also no automated tools to validate the generation of codebases for system design tasks either, therefore we propose evaluation metrics to compare the methods.

**Benchmark tasks**. We evaluate against three standard system design codebase generation tasks whose prompt questions are taken from real-world system design interview questions (Xu & Lam, 2020; Martin, 2023), and the HumanEval benchmark (Chen et al., 2021); with all details in Appendix E.

**Evaluation metrics**. Large-scale codebase generation is unique in that the generated code can satisfy the high-level user-specified task feature requirements through various possible implementation approaches. To quantify the degree to which the user-specified features in the initial prompt are effectively implemented in the generated code, we introduce a performance metric named **Features %**. This metric numerically represents the proportion of input features that are fully and functionally implemented in the output codebase. The **Features %** is obtained by using a separate GPT-4 API call, which iteratively examines the entire generated code to verify the functional implementation of

Table 2: Codebase generation system design task results showing the percentage of functional features specified that are fully implemented (**Features %**), the number of syntactical errors in the generated code (**# Errors**), the number of lines of code (**LOC**), and the number of passing tests (**Tests Passed**). Code-L2MAC fully implements the highest percentage of user-specified task feature requirements across all tasks by generating fully functional code that has minimal syntactical errors and a high number of passing self-generated unit tests. The results are averaged over 10 random seeds.

| Method | URL Shortener App | | | | Online Social Media App | | | | Online Chat App | | | |
| | Features % ↑ | # Errors ↓ | LOC | Tests Passed ↑ | Features % ↑ | # Errors ↓ | LOC | Tests Passed ↑ | Features % ↑ | # Errors ↓ | LOC | Tests Passed ↑ |
| --- | --- | --- | --- | --- | --- | --- | --- | --- | --- | --- | --- | --- |
| GPT4 | 53.6±10.5 | 0±0 | 119±21.1 | 2.56±0.95 | 19.5±8.28 | 4.09±3.32 | 116±31.5 | 0.818±0.785 | 11±2.26 | 0.3±0.346 | 127±24.1 | 1.2±1 |
| CodeT | 52.9±6.74 | 0.05±0.105 | 110±11.8 | 3.6±0.513 | 19.5±5.19 | 0.4±0.603 | 106±17.7 | 2.6±1.76 | 10.5±4.61 | 0±0 | 91.6±25.9 | 3.32±1.57 |
| Self-Refine | 47.9±8.53 | 0.05±0.105 | 124±15.7 | 3.65±1.15 | 16.4±2.62 | 0.938±0.714 | 110±19.6 | 1.81±0.938 | 14.2±4.19 | 0.211±0.304 | 111±13.8 | 1.42±0.927 |
| Reflexion | 38.8±6.02 | 0.1±0.209 | 96.2±9.11 | 2.35±0.631 | 15.2±8.05 | 2.53±1.69 | 122±24 | 1.33±2.44 | 10.2±3.08 | 0±0 | 76±6.88 | 2.85±0.822 |
| AutoGPT | 25.3±19.6 | 0±0 | 136±41.9 | 3.3±1.91 | 33.3±18 | 0.6±0.369 | 148±35.5 | 3±2.86 | 23.1±11.8 | 1.85±2.47 | 220±65.8 | 3.08±3.34 |
| Code-L2MAC | **91.6±8.22** | **0±0** | **330±47.6** | **14±6.71** | **82.4±14.6** | **0±0** | **395±52.9** | **18.3±6.8** | **59.4±25.9** | **0±0** | **374±123** | **18.8±9.11** |

all input features, counting the number of fully implemented features. We quote this as a percentage of the features implemented over the total features specified by the user. A detailed description and a motivating example of this implementation are provided in Appendix G. We also incorporate standard code generation evaluation metrics (Hasemer, 2018), such as the number of lines of code generated **LOC** and the number of errors **# Errors** in the codebase as determined by a code syntactical analyzer (Thénault, 2023). Furthermore, each method is instructed to generate unit tests to check that the generated code is valid; therefore, we quote how many of these self-generated unit tests pass as **Tests Passed**. We give each metric with their 95% confidence intervals throughout. Moreover, we detail these metrics and the experimental setup in more detail in Appendix G.

**Benchmark methods**. To assess whether Code-L2MAC is state-of-the-art, we compare it with the most competitive and popular autonomous agent LLM with memory **AutoGPT** (Richards, 2023), based on GPT4 and capable of reading, writing, and persisting, using a vector embedding memory to complete given tasks autonomously. We also compare with code reflecting LLM methods of **CodeT** (Chen et al., 2022), **Self-Refine** (Madaan et al., 2023), **Reflexion** (Shinn et al., 2023) and a single GPT4 LLM (**GPT4**) (OpenAI, 2023)—we make all these competitive by providing them with the same tools that Code-L2MAC uses. We provide the method implementation details, hyperparameters, and experimental details in Appendix F.

## 6.1 Main results

We evaluated all the benchmark methods across all our system design tasks with results tabulated in Table 2. Code-L2MAC fully implements the highest percentage of user-specified task feature requirements across all tasks by generating fully functional code that has minimal syntactical errors and a high number of passing self-generated unit tests—therefore, Code-L2MAC is state-of-the-art for completing these system design large codebase generation benchmark tasks. We further evaluated Code-L2MAC on the standard HumanEval benchmark (Chen et al., 2021) and observe that it achieves a state-of-the-art score of 90.2% Pass@1 (Appendix H.1). Also, we show L2MAC works for general-purpose extensive text-based tasks, such as writing an entire book (Appendix H.2).

## 6.2 Insight experiments

This section provides an in-depth empirical analysis of the efficacy of Code-L2MAC compared to its benchmark counterparts, AutoGPT and GPT-4. Specifically, we examine the core properties we suggested an effective extensive code generation framework should possess: task-oriented context management (P1), precise read/write tools (P2), and output error checking and correcting (P3).

**Can Code-L2MAC correctly perform task-oriented context management? (P1)**. To explore if the benchmarked methods during operation contain the information within their context to complete the task directly, we adapted our *Features %* metric to count the number of user-specified task feature requirements that are retained within the methods task instructions instead, i.e., those instructions that are eventually fed into its context window during its operation, as shown in Figure 3 (a). Empirically, we observe that Code-L2MAC is able to retain a high number of user-specified task feature requirements within its instructions $\mathcal{I}$ and perform instruction-oriented long-running tasks. We note that AutoGPT also initially translates the user-specified task feature requirements into task instructions; however, it does so with higher compression—condensing the information into a mere six-sentence description. This process results in the loss of crucial task information necessary for completing the overall task correctly, such that it aligns with the detailed user-specified task.

**Can Code-L2MAC perform precise read/write operations for the entire file store (P2)?** We wish to understand, during the operation of executing a task instruction, if Code-L2MAC can understand the existing generated code files within the codebase—which could have been created many instructions

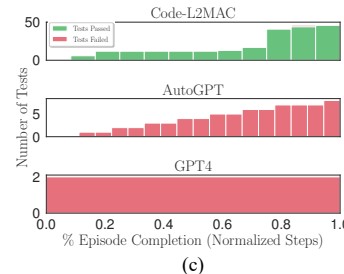

Figure 3: **Experimental Insight during operation**—of generating a codebase for an Online Chat App task. **(a)** Percentage of user-specified feature requirements that are retained within the methods task instructions and used in context. **(b)** Number of syntactical errors within the codebase. **(c)** Stacked histograms of passing and failing self-generated unit tests.

ago, and through its understanding, create new files that interrelate with the existing files, and most importantly update existing code files as new features are implemented. To derive insight, we plot a heatmap of the reading, writing, and when files are created at each write operation step during one episode in Figure 4. We observe that Code-L2MAC has an understanding of the existing generated code that allows it to update existing code files, even those originally created many instruction steps ago, and can view the files when it is not certain and update the files through writing to the files. In contrast, AutoGPT often only writes to files once, when initially creating them, and can only update files that it knows about that are retained within its current context window. Although it also has a read file tool, it often forgets about the files that it created many iterations ago due to its context window handling approach of summarizing the oldest dialog messages in its context window, i.e., a continual lossy compression of the previous progress made during operation of completing the task.

**Can Code-L2MAC check the generated output and error correct (P3)?** When using a probabilistic model (LLM) as a generator to output code, errors can naturally occur in its outputs. Therefore, we wish to verify if, when errors do appear, the respective benchmark methods can error-correct the codebase. We plot the number of syntactical errors in the codebase during a run where errors are made in Figure 3 (b). We observe that Code-L2MAC can correctly error correct the previously generated codebase that has errors contained within, which could arise from syntactical errors from the last file written or other files that depend on the most recent file written, which now contain errors. It does this by being presented with the error output when it does arise and modifying the codebase to resolve the error whilst still completing the current instruction. In contrast, AutoGPT cannot detect when an error in the codebase has been made and continues operating, which can compound the number of errors forming within the codebase.

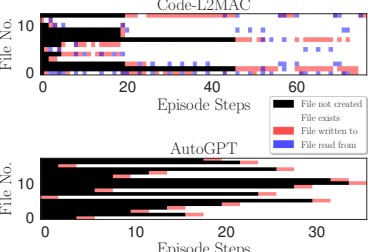

Figure 4: Heatmap of file access. Indicating reading, writing, and when files are created at each write operation step during one episode for the Online Chat App task.

Moreover, Code-L2MAC generates unit tests alongside the functional code and uses these as an error checker to inspect the functionalities of the codebase as it is generated and can use these errors to fix the codebase to pass unit tests that now fail after updating part of an existing file. We show this in Figure 3 (c) and observe that AutoGPT, whilst prompted to also write unit tests for all code generated, is unable to use these tests as an integrity error check, which could be compounded by the observation that AutoGPT forgets which files it has previously created and hence unable to modify the existing forgotten code files as new modifications are made, leading to incompatible code files.

## 7 CONCLUSION AND FUTURE WORK

In this paper, we present L2MAC, the first LLM-based general-purpose stored-program computer framework that effectively and scalably augments LLMs with a memory store for long output generation tasks where this was not previously successfully achieved. Specifically, Code-L2MAC, an application for long code generation tasks, surpasses existing solutions—and is an immensely useful tool for rapid development. This work is not without limitations. L2MAC's performance is inherently bounded by its underlying LLM across various tasks, from planning to tool use. Further, the current design imposes an implicit constraint on the scale of the codebase by listing all file names within the context window, which could readily be improved. These, and among many other aspects such as complex instruction flows, re-programming the instructions, and interpretable prompt programs pose exciting open directions for future work to build upon as detailed in Appendix I.

**Ethics statement**. This paper proposes the first practical framework for an LLM-based stored-program computer and instantiates a version to generate large codebases, named Code-L2MAC. However, misuse or use of such a tool with an incorrectly specified task could lead to undesirable outputs. Furthermore, due to the use of an LLM, such a tool is prone to hallucinations, and hence, such results should always have a content filter on the final output.

**Reproducibility statement**. All code is available at https://github.com/samholt/L2MAC. Seeking full reproducibility, we include an extensive appendix with all implementation and experimental details to re-create the benchmark tasks and baseline methods. These are detailed in the following: for benchmark task environment details, see Appendix E; for benchmark method implementation details, see Appendix F; regarding Code-L2MAC implementation details, for high-level details, see Appendix C, and for low-level details, see Appendix F.1; and for evaluation metric details, see Appendix G.

**Acknowledgements.** The authors would like to acknowledge and thank their corresponding funders, where SH and ML are funded by AstraZeneca. Moreover, we would like to warmly thank all the anonymous reviewers, alongside research group members of the van der Schaar lab (www.vanderschaar-lab.com) and Andrew Rashbass, for their valuable input, comments, and suggestions as the paper was developed.

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

# Appendix

## Table of Contents

**Code.** All code is available at https://github.com/samholt/L2MAC. We have a broader research group codebase at https://github.com/vanderschaarlab/L2MAC.

## A  STORED-PROGRAM COMPUTERS (SPC)

**Stored-Program Computer (SPC)**. Originally termed the von Neumann architecture, was first proposed by Von Neumann (1945) as a combination of a processor (arithmetic and logic unit), a Control Unit (CU), and memory capable of storing both instructions and data, SPCs set the foundation for modern computer design. With a control unit (CU) managing the interaction between both components, this design is capable of being reprogrammed to solve various tasks without manual intervention [9]. In more detail, the CU uses instructions from the memory to set the processor to a specific state (e.g., load two numbers into the processor's (arithmetic logic unit) input registers and then execute addition and store the result in memory. The computer is not only able to perform arithmetic and logic operations, but it can also manipulate the memory. One unique aspect often overlooked in this paradigm that is crucial to the reliable operation of the processor is error-checking of the processor's outputs. Errors can arise in a typical processor from hardware influences such as temperature, voltage variations, or radiation (e.g., cosmic rays) (Nicolaidis, 2010), or from miss-specified inputs leading to overflows (e.g., divide by zero errors). Therefore, typical processors involve output checks, the simplest being parity bit checks, and can combine these with error-correcting mechanisms (Harris & Harris, 2010).

### A.1  SPC IS A UNIVERSAL TURING MACHINE

**Stored-program computer (SPC) and Automata Theory**. We can consider an SPC as a specific type of Turing machine $T$. Formally, a Turing machine is defined by the 7-tuple $T = (Q, \Sigma, \Gamma, \delta, q_0, b, F)$, where:

- $Q$ is a finite set of states.
- $\Sigma$ is the finite input alphabet not containing the special blank symbol $b$.
- $\Gamma$ is the finite tape alphabet, where $b \in \Gamma$ and $\Sigma \subseteq \Gamma$.
- $\delta : Q \times \Gamma \to Q \times \Gamma \times \{L, R\}$ is the transition function. Given a state and a tape symbol, it specifies the next state, the symbol to write on the tape, and the direction in which the machine should move its head (Left or Right).
- $q_0 \in Q$ is the start state.
- $F \subseteq Q$ is the set of accepting states.

In the context of a stored-program computer, the memory holds both the instructions (akin to the transition function $\delta$) and the data (akin to the input sequence from $\Sigma$). The program counter, which determines the next instruction to execute, can be represented by the machine's current state in $Q$. The execution of a program on an SPC is then analogous to the computation performed by a Turing machine when transitioning from one state to another based on the input read and the defined transition function $\delta$.

The universality of an SPC, similar to a universal Turing machine, means that given the appropriate transition function (or set of program instructions) in its memory, it can simulate any other Turing machine and perform any computable task (Turing et al., 1936).

### A.2  WHAT IS AN AUTOMATIC COMPUTER?

The term 'automatic computer,' as used in this paper, comes from the original conception presented by Turing et al. (1936), where Turing introduced the concept of an *automatic*-machine—commonly known today as a Turing machine. In this context, 'automatic' refers to a system capable of executing a pre-defined set of instructions without human intervention (Turing et al., 1936; Von Neumann, 1945).

---

[9]As a side note, the Church-Turing Thesis posits that both Universal Turing Machines (UTMs) and SPCs are capable of simulating each other and, therefore, are models of the same class of computational processes Von Neumann (1945) (Appendix A.1). Recently, it has been shown that augmenting an LLM with external read-and-write memory allows it to simulate a UTM (Schuurmans, 2023). However, we do not aim to reproduce the capabilities of the current CPU-SPC in arithmetic and logic operations but instead explore the powerful abilities of LLM-SPCs as automatic computers.

In the context of LLMs, there are methods whereby LLMs can autonomously execute tasks, as reviewed by Wang et al. (2023a). However, it is crucial to differentiate between the concepts of 'autonomous' and 'automatic'. Autonomous agents are characterized by their ability to independently devise and adapt plans, implying a level of self-determination and initiative in executing instructions. In contrast, our focus with L2MAC is on developing a system that meticulously and automatically executes pre-defined instructions contained in the prompt program $\mathcal{I}$. This execution requires no human intervention during the process until all instructions have been successfully executed. In this regard, L2MAC aligns more with the traditional concept of an automatic computer. It functions as a general-purpose, instruction-following LLM-based computer (Von Neumann, 1945) dedicated to efficiently executing the provided instructions, thereby ensuring adherence to the completion of the originally specified prompt program. The term 'general-purpose' here implies that L2MAC can accommodate any prompt program as input and execute it without necessitating system redesign for each new prompt-program introduced.

## B  CONTROL UNIT OPERATION

The Control Unit, manages the entire context of the LLM and is itself a finite state machine. Here we expand on Section 3.3, provide pseudocode in Algorithm 1, provide an extend block-diagram figure of L2MAC in Figure 5, and a data flow diagram in Figure 6.

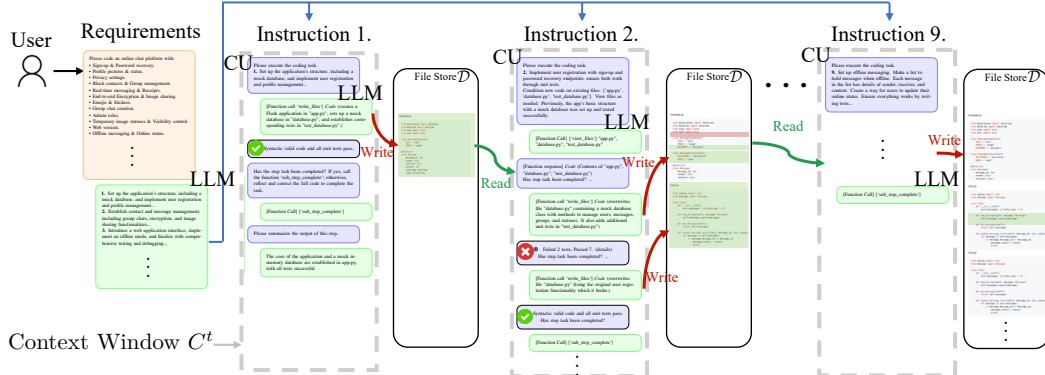

Figure 5: **Code-L2MAC Extended Overview**. Code-L2MAC is an instantiation of the LLM automatic computer (L2MAC) framework, here for extensive code generation. First, it breaks down the user task into sequential instructions $\mathcal{I}$. The Control Unit (CU) manages the LLM's context window $C^t$ for each instruction and interacts with the external memory file store $\mathcal{D}$ through read, write, and evaluate tools. It identifies and reads relevant files from the memory to generate or update files per instruction (P2). This ensures proper conditioning of existing files without losing vital context (P1). Automatic checks evaluate the LLM's outputs for correctness and completion (P3), with iterative error corrections involving both code syntactical checks of the code and running self-generated unit tests to check desired functionality. Overall, this produces a complete large codebase that fulfills the detailed user task in the file store $\mathcal{D}$.

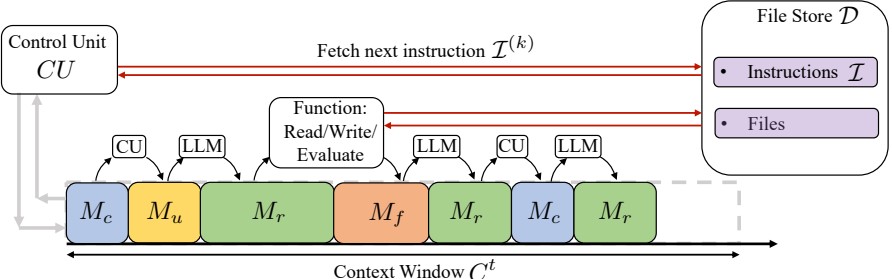

Figure 6: **Data flow of L2MAC executing one prompt instruction** $\mathcal{I}^{(k)}$ **from a self-programmed prompt-program** $\mathcal{I}$. The first practical LLM-based stored-program automatic computer (L2MAC) framework. The L2MAC consists of a file store $\mathcal{D}$, an entire memory write component $\mathcal{W}$, an entire memory read component $\mathcal{R}$, and a program control flow mechanism that of a Control Unit $CU$ (c.f., Figure 2) to manage the LLM's context window—encompassing both its inputs and outputs. Here, the context window $C^t$ (dashed gray box) consists of a sequence of messages, where each message has one of the following message types: control $M_c$, user $M_u$, LLM response $M_r$ and function response $M_f$.

## C CODE-L2MAC HIGH LEVEL IMPLEMENTATION DETAILS

In the following we describe the high-level implementation details of Code-L2MAC, and we provide the verbose low level implementation details, such as the function JSON definitions in Appendix F.1. We follow the same structure as the L2MAC outline and focus on the specific design choices we made for each item. For the **LLM Processor**, we use GPT-4-0613 OpenAI (2023), which trains GPT-4 to receive in-context descriptions of functions (tools) and call them as needed, where the descriptions follow a JSON format (OpenAI, 2023).

The functions we provide are the following:

1. 'provide_detailed_sub_task_steps_for_sub_agents'. For producing a step-by-step plan, where each step paragraph is a detailed sub-task step for a separate sub-agent LLM to complete.

2. 'sub_task_step_complete'. For the LLM to call when the user specified sub task step has been completed.

3. 'view_files'. Prints out all the file contents into the response to view.

4. 'run_python_file'. Runs a python file and returns the output to the response to view.

5. 'write_files'. Writes out multiple files that will be combined into the existing codebase.

6. 'delete_files'. Deletes files, where their names are specified.

Next, we discuss specific choices to develop a method based on our LLM Automatic Computer (L2MAC) framework specifically designed for large code generation tasks, which we call **Code-L2MAC**.

For the **LLM Processor**, we use GPT-4-0613 OpenAI (2023), which trains GPT-4 to receive in-context descriptions of functions and call them as needed following a JSON format (OpenAI, 2023). See Appendix F.1 for a low-level JSON definition of the functions that we provide to the LLM, most of which we also mention below.

In terms of the **instructions**, we initialize it[10] with $\mathcal{I}^0 = \emptyset$, and the CU queries the LLM only in the first iteration with the user-specified feature requirements (task description) and requests the LLM to define a sequential prompt program $\mathcal{I} = [\mathcal{I}^{(0)}, \ldots, \mathcal{I}^{(K)}]$.

Regarding the data $\mathcal{D}$, we determine that it encompasses only the code generated so far so that upon termination of Code-L2MAC, $\mathcal{D}$ corresponds to its output. However, we impose the codebase $\mathcal{D}$ to

---

[10]Rigorously, $\mathcal{I}$ is initialized with a set of pre-defined messages for the LLM to be effective as part of Code-LLM (including the functions the CU provides to the LLM).

---

**Algorithm 1** Control Unit Pseudocode for L2MAC

---

1: **Input:** Initial user input prompt $p_{\text{in}}$; file store $\mathcal{D}$; Domain Evaluator unit $\mathcal{E}$; Domain system message $M_s$; Domain bootstrap message $M_{cb}$; Context window constraint $c$; Unwind token margin $b$

2: **Output:** External memory store $\mathcal{D}$.

3: $\mathcal{I} \leftarrow \emptyset, M_{rs} \leftarrow \emptyset$

4: $M_u \leftarrow p_{\text{in}}$

5: $M_r \leftarrow LLM(\{M_s, M_{cb}, p_{\text{in}}\})$        ▷ **Self-program** $\mathcal{I}$, by bootstrap instruction

6: $(\mathcal{I}, \mathcal{D}) \leftarrow$ process_into_prompt_program$(M_r, \mathcal{D})$

7: **while** $\mathcal{I}$ is not empty **do**

8:      $C^t \leftarrow \{M_s, M_{rs}\}$        ▷ **Clear Context Window: It always contains the system and summary message**

9:      $(\mathcal{I}^{(k)}, \mathcal{D}) \leftarrow$ pop_front_of_$\mathcal{I}(\mathcal{I}, \mathcal{D})$      ▷ **Fetch:** next instruction from external memory $\mathcal{D}$

10:      $C^t \leftarrow C^t \oplus \mathcal{I}^{(k)}$

11:      has_instruction_completed $\leftarrow$ False

12:      **while** has_instruction_completed is False **do**

13:          $M_r \leftarrow \mathcal{P}_{\text{LLM}}(C^t)$

14:          $\Delta_{C^{t+1}} \leftarrow \{M_r\}$

15:          **if** 'function_call' in $M_r$ **then**

16:              $(M_f, \mathcal{D}) \leftarrow$ execute_function$(M_r, \mathcal{D})$

17:              ▷ **Invoke tool:** Includes functions (tools) to read/write and evaluate the file store $\mathcal{D}$

18:              $\Delta_{C^{t+1}} \leftarrow \Delta_{C^{t+1}} \oplus M_f$

19:              **if** 'step_completed' in $M_r$ **then**

20:                  $M_{fe} \leftarrow \mathcal{E}(\mathcal{D})$        ▷ Run evaluator module on file store

21:                  **if** $M_{fe}$ is $\emptyset$ **then**        ▷ If evaluator checks pass

22:                      **while** $|C^t \oplus \Delta_{C^{(t+1)}}| > b$ **do**      ▷ **Summarize instruction output to** $M_{rs}$. Unwind oldest messages to make room for summary control message $M_{cs}$ and summary response $M_{rs}$

23:                          $(\_, C^t) \leftarrow$ pop_back_of_$C^t(C^t)$

24:                          $M_{rs} \leftarrow \mathcal{P}_{\text{LLM}}(C^t \oplus \Delta_{C^{(t+1)}} \oplus M_{cs})$

25:                      has_instruction_completed $\leftarrow$ True

26:                  $\Delta_{C^{t+1}} \leftarrow \Delta_{C^{t+1}} \oplus M_{fe}$

27:          $\Delta_{C^{t+1}} \leftarrow \Delta_{C^{t+1}} \oplus M_{cc}$        ▷ Append the cycle message

28:          **if** $|C^t \oplus \Delta_{C^{(t+1)}}| > c$ **then**        ▷ **Would exceed context check**

29:              **while** $|C^t \oplus \Delta_{C^{(t+1)}}| > b$ **do**      ▷ **Summarize progress to** $M_{rs}$. Unwind oldest messages to make room for summary control message $M_{cs}$ and summary response $M_{rs}$

30:                  $(\_, C^t) \leftarrow$ pop_back_of_$C^t(C^t)$

31:              $M_{rs} \leftarrow \mathcal{P}_{\text{LLM}}(C^t \oplus \Delta_{C^{(t+1)}} \oplus M_{cs})$

32:              $C^{t+1} \leftarrow \{M_s, M_{rs}, \mathcal{I}^{(k)}\}$

33:          **else**

34:              $C^{t+1} \leftarrow C^t \oplus \Delta_{C^{t+1}}$        ▷ Append Messages

35: **return** $\mathcal{D}$

---

be partitioned into files no longer than the context window[11], and we name the files with a path from a root directory / where the name of folders and files suggests its content. This choice is not only useful for a human reader but will be crucial for our Read-and-Write implementations.

The remaining details are in the arguably richest component of the method, the **control unit**.

**Execution flow of the LLM**. In Code-L2MAC, we can describe the execution flow as the following iteration. For each instruction $\mathcal{I}^{(k)} \in \mathcal{I}$, while $\mathcal{I}^{(k)}$ is not satisfied, do the following—following the CU pseudocode Algorithm 1. First, load the fixed messages that provide in-context learning to the LLM, the description of the current instruction $\mathcal{I}^{(k)}$, and the summary of the last execution $M_{rs}$. We

---

[11]More precisely, no longer than half the margin left in the context after the initial messages are provided in any iteration in the execution. This is to ensure that the LLM can read and rewrite any given file inside its context.

let the LLM iterate (work on completing the instruction) while in context, and if approaching out of context, summarize its current context and iterate.

**Interfacing with tools**. During execution, the LLM might request reading and writing operations. After each write (which possibly introduces a new test), a syntactical checker and all existing tests in $\mathcal{D}$ are run, and the output is provided to the LLM—through $M_{fe}$. If, after receiving this feedback, the LLM considers the task completed, it can signal it to the CU through calling a special function 'step_complete', which summarizes the current context output into $M_{rs}$ and continues executing the next instruction with this summary $M_{rs}$. After all $K$ instructions are completed, the process halts (stops), and $\mathcal{D}$ is returned.

We are only left to detail the process of **reading and writing**. Given the instruction $\mathcal{I}^{(k)}$, the LLM must be able to know what modules in $\mathcal{D}$ it can call, how to call them, and where in the directory to write new code. However, the size of $\mathcal{D}$ might exceed the context window.

Due to the restrictions mentioned on the files in $\mathcal{D}$, the LLM can read and write any single full file as desired and, if necessary for context preservation, summarize this information for the next iteration. However, the decision regarding which files are read can be heuristically based on the semantic information provided by the path and name of the file, which encodes its role in $\mathcal{D}$.

Specifically, we enable the read component $\mathcal{R}$ of Code-L2MAC by implementing the two functions. `list_files()`, which lists the file paths in breadth-first search order. This could be refined to only list a subset of files within a subdirectory, but for simplicity, in our implementation, we automatically list all file paths in $\mathcal{D}$ for the LLM as contexts to enable it to be aware of existing code and help it determine where to include newly generated files. And `view_files(files)`, which takes a list of file paths and returns strings with their content—appending this to the context window.

We also provide the LLM at all times with a system message $M_s$ to inform it that it is to complete a coding task, that it must act without human input, and a few details about the coding environment that exists within—for example, that it can use the python test framework Pytest, and can only mock in-memory databases as none are installed in the local environment. Although this could be resolved by installing and providing a clean, fresh version for most common database providers, we leave this extension as a future work to explore.

We also provide the LLM with a pre-prompt message that it uses with the user-specified requirements, to construct the initial plan, we also detail this and the system message in Appendix F.1.

## C.1 READ / WRITE IMPLEMENTATION DETAILS

Here we expand on the read/write implementation details of Code-L2MAC. This builds on the existing descriptions contained within Section 4 and appendices C and F.1.

The LLM interfaces with the control unit (CU) through calling functions, and thus, the read and write implementation can be fully described through a discussion of the functions that are provided to Code-L2MAC to that end, which are 'read_files' and 'write_files'.

It is worth noting the control unit always exposes the LLM to the list of file paths that are already part of the codebase (e.g., "file paths: ['app.py', 'tests/test_app.py']"). The name of directories and files (i.e. file path) provides a semantical prior for the order in which the LLM will read each file, depending on the functionality it is looking for (among other elements, including always the current instruction $\mathcal{I}_t$, and possibly previous dialog turn outputs and error responses).

Note that if the LLM initially reads a file that does not contain what it was looking for or the desired functionality is spread across multiple files, the LLM can continue reading other files. Recall that when the context window is full, the CU prompts the LLM to summarize the relevant parts of the context window for the current instruction $\mathcal{I}_t$. In this case, it could summarize which are the relevant files it needs to read if the functionality is spread across multiple files.

Assume the LLM has already determined the name of the file 'file_path' that it wants to read. Let us zoom into the functions (tools) provided to the LLM to signal intentions to the CU.

- 'read_files(file_path)' requests the CU to output into the context window $C^t$ the contents of 'file_path'.

- 'write_files(file_path, content)' requests the CU to create or overwrite 'file_path' and populate it with 'content'.

As the empirical results demonstrate, these functions perform well as components of Code-L2MAC. The future work appendix (Appendix I) discusses how these functions could be optimized.

We find Code-L2MAC's reading and writing functionality different from previous memory-augmented LLM (where we refer to the extended related work for more details, Appendix D) as these can be categorized as:

- **Append only memory:** These methods explicitly extend the implicit knowledge of LLMs through an external corpus (Zhong et al., 2022) to facilitate long conversational or document summarization tasks (Liang et al., 2023).

- **Key-value memory:** These methods interface the LLM with a dictionary or a database (Hu et al., 2023), which does not apply to our automatic coding tasks.

## D  EXTENDED RELATED WORK

### D.1  VANILLA LLMS

The usage of only a single LLM without a memory augmentation approach is limited to only producing an output that can be maximally as long as the underlying LLM's context window constraint $c$. Although not directly applicable, they are still related since the main focus of all coding tasks to date in the literature focuses on using an LLM to solve a code generation task within the context window constraint $c$. Notable examples include Codex (Chen et al., 2021), GPT4 (OpenAI, 2023), GPT-Engineer (Osika, 2023) and Code LLama (Rozière et al., 2023).

**Vanilla LLMs with longer context windows**. One might try to directly extend the context window. However, such an approach scales quadratically in computational complexity (Vaswani et al., 2017), an impracticality worsened by the observation that LLMs with long context windows attend information from the middle of the window poorly, substantially decreasing their task performance as the length increases (Liu et al., 2023a). Another naive approach would be to fine-tune the LLM in the environment at hand so that it implicitly memorizes it in its weights, but this should be done recurrently as the state evolves, which is also too computationally expensive to be practical (Brown et al., 2020).

### D.2  TRANSFORMERS FOR LONG INPUTS

There are architectural approaches to directly allow LLMs to ingest larger windows. Tay et al. (2022) offers a comprehensive review on this topic of "efficiency-flavored" transformers.

Child et al. (2019); Beltagy et al. (2020); Zaheer et al. (2021) present sparse attention mechanisms that scale subquadratically with the length of the sequence, Guo et al. (2022); Phang et al. (2022) draw on local/global attention mechanisms to scale to longer inputs, Wang et al. (2020) approximate the attention mechanism with a low-rank matrix to achieve linear complexity, Choromanski et al. (2022) provide an unbiased estimator of the attention that converges uniformly, Peng et al. (2021) use random features to approximate soft-max attention with a decreased computational burden. Although this list is not exhaustive, the precise impact of these optimizations on the performance of LLM is not yet fully understood, and the adoption of these approaches in practice is not widespread.

In Press et al. (2022), the authors discuss biasing attention scores with a penalty proportional to their distance instead of using positional embeddings, which allows them to extrapolate to longer sequences in inference time effectively. Consequently, they achieve significantly more efficient training. However, this does not address the challenges of long context windows at inference time.

### D.3 MEMORY AUGMENTED LLMS

#### D.3.1 LLMS WITH BUILT-IN MEMORY

These are works that expose the LLM to memory at training time. Yogatama et al. (2021) extend the usual attention architecture in two ways. First, they extend the context window without much burden at training time by omitting the backward step for distant tokens. Second, they update the next token probability distribution by incorporating information about the next token for historical data for the nearest neighbors in latent space. Borgeaud et al. (2022) essentially incorporates for each training example the k-NN (based on an embedding fixed through training) from an external corpus. Zhong et al. (2022) propose a boost to the next token probability distribution based on the similarity of the current embedding to the previous embeddings where this token appeared. These enable the LLM to be trained with this memory component but can essentially be thought of in the same spirit as embedding NN lookup memory. However, none of these methods enables the memory to be updated.

Finally, (Bulatov et al., 2022) embeds into a sequence of tokens memory for the next inference. This approach, however, has a low-interpretable, non-symbolic, and limited memory, thereby not fully addressing arbitrarily long tasks.

#### D.3.2 INFERENCE-TIME MEMORY-AUGMENTED LLMS

The plainness of the limitations imposed by a limited context window has motivated several efforts to overcome it, which has inevitably led to the concept of memory. However, most of such works focus on long conversational / text summarization tasks (Liang et al., 2023; Modarressi et al., 2023; Zhong et al., 2022; Cheng et al., 2023), or mathematical/database reasoning (Hu et al., 2023). The methods from the first group, due to the domain, focus exclusively on the read component of memory. The same happens with (Wu et al., 2022), which explores the $k$-NN lookup to retrieve memory for next-token predictions and observes improvements in perplexity for multiple tasks, including coding. However, such read-only implementations are more sensitive to the erratic fallibility and the "stochastic" nature of LLMs to break code and cannot interact with memory in any other way. An exception to this is Modarressi et al., 2023, but its memory (and thus update capabilities) is composed of tuples of the form (Subject 1, Subject 2, Relationship), for which there is no apparent application to coding tasks. Similarly, mathematical / database reasoning methods such as (Hu et al., 2023) reduce memory to a dictionary of values or a structured database, which again are not applicable to different settings (e.g., coding).

#### D.3.3 AUTNOMOUS AGENT LLMS WITH MEMORY

These works formulate the LLM as an agent in a reinforcement learning environment, where it observes a state and then takes an action, often calling a particular tool with an argument and interacting with the environment until the task is solved (Wang et al., 2023a). Unique to these works is they act fully *autonomously*, where they are often given a few high-level goals and construct their own plans, which are continuously updated to have self-guided operation without external input, to complete their goals or run continuously (Richards, 2023; Shrestha & Watkins, 2023; Nakajima, 2023). Regarding code generation tasks, this implies that these agents can re-program and re-formulate their step plan at run-time; however, this can often lead to the agents going off-topic instead of completing the specified given task and getting stuck in infinite loops (Wang et al., 2023a; Sato et al., 2023). However, in L2MAC we specify that it is *automatic* rather than fully *autonomous* so that the specified user input commands can be performed without manual intervention whilst achieving the input specified given task by the user. Furthermore, of the existing Autonomous Agent LLMs with memory, the only directly applicable method capable of writing, reading, and executing code of files is AutoGPT (Richards, 2023). This overcomes the context window constraint to complete long-running tasks iteratively *lossly* summarizing the previous actions completed after a given number of turns and only keeping the running summary in the context window. Prevalent with AutoGPT and others is that they contain a first step of taking the user-specified task, and *lossly* summarizing it into a name, role, and five one-sentence goals for the agent to complete; this goal message also is always in the agent's context window. At each dialog step, the LLM is prompted with a ReACT (Yao et al., 2022) message template to reason, plan, and act by selecting the next tool to call until the goals have been achieved, then stops. Overall, these methods, such as AutoGPT have two key limitations: (1) they lossily compress the initial detailed user-specified task into a six-sentence

description (a sentence role and five goal sentences), and (2) they lossily compress the previous action history, which for code generation tasks equates to forgetting which files were created and their use, which is further compounded by re-planning with a separate new plan at dialog step. Whereas our proposed LLM automatic computer (L2MAC) can (1) create a detailed plan initially and use that single plan throughout to align to complete the original user-specified task, and (2) L2MAC is aware of the existing files generated, which encapsulates all the previous actions up to the current state (where the files track the current memory state), allowing it to know and integrate new code with the existing files already generated.

### D.4 TOOL AUGMENTED LLMS

Schick et al. (2023) teach LLMs to use external tools through API calls, for example, to use a calculator, a search engine, or a translation system to cover LLM weaknesses or reduce computation cost. The methods by which the LLM chooses when to read from memory (instead of automatically receiving the information obtained from memory) can basically be regarded as providing a new tool to the LLM (e.g., Hu et al. (2023); Modarressi et al. (2023)). This is the case for the L2MAC implementation of reading and writing, but not all the functionalities of the control unit can be seen under this lens, as is the case with the monitoring of the context window or the automatic evaluation of the file store, unless we stretch it excessively. In any case, as argued in Modarressi et al. (2023), this similarity with Schick et al. (2023) does not undermine the relevance of the development of new and influential tools.

### D.5 MULTI-STEP REASONING & REFLECTING LLMS

**Reflecting LLMs**. These methods can improve the task performance of existing single LLMs by having access to an evaluator and using the LLM to reflect on the actions taken (current trajectory part) to derive actionable verbal feedback that is persisted in an append-only memory (Shinn et al., 2023; Madaan et al., 2023). These verbal reflection learnings are a form of in-context learning (Dong et al., 2022) that are then used to improve the task or sub-task performance when the task or sub-task is restarted anew. This simple, and practical reflection idea, when having access to a possible external validator, such as self-generated unit tests for code generation, have demonstrated improving single LLMs for small code snippet generation tasks performance to be state-of-the-art (Shinn et al., 2023). A unique characteristic our our checks is that they are not only aimed at validating the immediate output but rather to impose coherence on the memory as a whole. This implies that while in previous settings, a failing test would require a change in the LLM output, in L2MAC, it can also motivate revisiting and refactoring a preexisting component in the file store.

Specifically, we find the following differences between Code-L2MAC and reflecting LLM methods, such as Self-Refine and Reflexion to be:

- Self-Refine (Madaan et al., 2023) refines the most recent output from the LLM and Reflexion (Shinn et al., 2023) the recent outputs that are only within the LLM's current context window $C^t$; whereas Code-L2MAC can refine the entire file store, encompassing all previous outputs from the LLM—allowing it to fix or improve earlier outputs from multiple instructions back, outside its context window, which are now erroring due to a new code implementation change. This enables Code-L2MAC to generate long, consistent, and interrelated code structures.

- Self-Refine and Reflexion are constrained to operating on an output within the context window constraint $c$, whereas Code-L2MAC can manage a total output greater than the context window constraint through the L2MAC framework.

**Multi-step reasoning**. Methods that leverage a hierarchical flow of information between LLMs can also handle information that does not fit the context length, thanks to the distribution of information. For example, (Zhang et al., 2022; Wu et al., 2021) iteratively and hierarchically perform the summarization of long documents. We refer to (Dohan et al., 2022) for the unification of such methods under a common formalism. Code-L2MAC uses simple instantiations of this family to populate $\mathcal{I}$ from the initial prompt or to summarize the execution when cleaning the context window, but future iterations could build on more complex cascades to improve our implementation of L2MAC. Multistep reasoning is only a small part of L2MAC, and we envision L2MAC enhancing multistep

reasoning methods through the CU and memory (c.f. Appendix I). Furthermore, there exist standard RL multi-step planning frameworks (Holt et al., 2024; 2023) which could be adapted to LLMs in future works.

### D.5.1 LLM-BASED MULTI-AGENT SYSTEMS

LLM-based Multi-Agent Systems, comprise of multiple LLM-powered agents that interact with one another to achieve a desired task (Wang et al., 2024; Guo et al., 2024). L2MAC is an example of an LLM-based multi-agent system, as it consists of separate LLM-based agents that work sequentially each getting a separate next instruction (where the instructions were generated by an initial LLM-agent) and persisting memory of the previous agent's outputs between the agent's, as illustrated in Figure 1. Broadly LLM-based multi-agent systems, facilitate sophisticated interactions and decision-making (Wang et al., 2024; Guo et al., 2024), proposed through collective intelligence and specialized profiles and skills of multiple agents, to allow them to collaboratively complete tasks. Such works include (Wang et al., 2023b; Du et al., 2023; Zhuge et al., 2023; Hao et al., 2023; Akata et al., 2023; Liu et al., 2023b; Park et al., 2023; Cai et al., 2023; Hong et al., 2024; Wu et al., 2023; Li et al., 2024; Packer et al., 2023).

### D.6 TURING MACHINES, STORED PROGRAM COMPUTERS AND MACHINE LEARNING

There are instantiations of Universal Turing Machines and Stored-Program Computers with LLMs (Schuurmans, 2023; Giannou et al., 2023), respectively, which are theoretically insightful. However, in both cases, their method with a transformer at the core only aims to simulate a UTM or SPC, thereby not achieving a method functionally superior to either a UTM or SPC (and none of them new), which can be implemented much more cheaply. In contrast, we capitalize on the SPC framework to boost the LLM's unique capabilities as a stochastic NLP to automatically solve potentially long tasks from a single user prompt, which is altogether a different focus.

Although (Graves et al., 2014) does not use LLMs, and the focus is also completely different to ours (since it aims to reproduce algorithms by supervised learning), it is worth mentioning that they extend a Recursive Neural Network with memory to achieve a Turing Machine or von Neumann architecture capable of inferring simple algorithms when trained to reproduce them.

## E  BENCHMARK TASK ENVIRONMENT DETAILS

### E.1  CODEBASE GENERATION SYSTEM DESIGN TASK

Given the absence of pre-existing benchmark task environments suitable for evaluating long code generation tasks, we have introduced a new benchmark tailored for standard system design tasks, which are typically assigned to humans. Traditional system design tasks often require high-level conceptualization, such as devising the architecture, modules, components, interfaces, and data structures needed for an online application to meet specific user feature requirements (Xu & Lam, 2020). Unlike these conventional tasks that predominantly focus on high-level system outlines, our benchmark mandates the implementation of fully functional code. This code aims to instantiate a minimal system satisfying the user-specified requirements in a practical manner.

For meaningful and realistic evaluation, our benchmark encompasses three standard system design codebase generation tasks. The prompt questions for these tasks are derived from actual system design interview questions, providing a realistic basis for assessment (Xu & Lam, 2020; Martin, 2023). The tasks included in the benchmark are as follows:

1. **URL Shortener App**: This task necessitates the implementation of a comprehensive online URL shortening service, a utility that enables users to submit lengthy URLs and subsequently receive a shortened variant for simplified sharing. The user-specified feature requirements for this task are extensive. The envisaged system should facilitate URL shortening services, allowing users to input, validate, and, if desired, customize their URLs. The shortened URLs should redirect users to the original addresses. Users should have access to analytics, revealing data such as click counts, timestamps, and geographical locations linked to each URL. The platform should also support user account creation and management, enabling

users to edit, delete, or view analytics related to their URLs. An administrative dashboard for monitoring user activity, user accounts, and overall system performance should be in place. Furthermore, users should have the ability to set expiration parameters for their shortened URLs.

2. **Online Microblogging App**: The task entails the implementation of a web-based microblogging service where registered users can post short textual messages, follow or unfollow others, and interact with the posts. Each user can create, edit, and manage their profile with options to set it as private or public. Users can post text messages limited to 280 characters, optionally include images in the posts, and have the ability to delete their posts. The platform will support interactions with posts, such as liking, retweeting, and replying, with a nested structure for the comments. Users can also search and filter posts or other users using keywords, hashtags, mentions, or view trending topics. Furthermore, users can follow and unfollow others, with a timeline view displaying posts from the users they follow and receiving notifications for new followers, likes, retweets, replies, and mentions. The service will also facilitate private messaging between users, with options to block or unblock users. Finally, the platform will offer user recommendations for following based on interests, activity, and mutual followers and showcase trending topics, which can be viewed globally or sorted by location.

3. **Online Chat App**: This task involves implementing a real-time online chat application. The GCS must support user registration and authentication functionalities, including email sign-up and forgotten password recovery. Users should be able to set profile pictures and status messages, with privacy settings to control visibility. The application should provide options for contact management (block/unblock contacts and group management), real-time text messaging with read receipts, end-to-end encryption, image sharing, and support for emojis, GIFs, and stickers. Group chat functionality should allow the creation of group chats with assigned names and pictures, participant management, and admin role assignments and permissions. Users should be able to post image statuses that are visible for a limited duration, with the ability to control viewer access. The service should be accessible as a web application and support offline messaging, with queued messages sent upon connectivity restoration and visible online/offline user statuses.

4. **Recipe App**: This task focuses on creating a service that allows users to submit, share, and discover recipes. Key features include the ability for users to submit recipes complete with ingredients, instructions, and images; options for categorizing recipes by cuisine type or dietary restrictions; and functionalities for editing or deleting submissions. The platform also supports searching for recipes based on various criteria such as ingredients or recipe name, and categorizing recipes by meal type or dietary needs. User account creation and management are integral, allowing for saving of favorite recipes and displaying user profiles with their submitted and favorite recipes. Additional features include a rating system where users can rate recipes on a 5-star scale and write reviews, a community aspect allowing users to follow others and see updates in a feed, and the sharing of recipes on social media platforms. An administrative dashboard for managing content, monitoring site usage, and user engagement is also included, alongside a system for generating recipe recommendations based on user preferences and past activity.

5. **Event Planner App**: This task involves developing a web-based application to assist users in organizing and managing various aspects of event planning. The application enables users to create and manage events, specifying details like event type, date, time, and customizations such as themes and color schemes. A calendar view for managing multiple events is included. Users can search for and book venues through the application, with integration of maps for venue locations. Guest list management is facilitated with features for creating, importing, exporting, and RSVP tracking. The platform connects users with event service providers like caterers and decorators, offering vendor profile viewing, comparison, and in-app messaging for coordination. Budget management tools allow users to set, track, and receive alerts for budget overruns. User account features include personal profile creation, customization, and access to past and upcoming events. Automated notifications and reminders for event milestones and tasks are part of the system. Reporting and analytics capabilities are provided for event success metrics, along with feedback collection from guests and vendors post-event. An administrative dashboard allows for the monitoring and management of user activities,

system performance, and vendor listings. Additionally, the tool emphasizes security and data privacy, including secure payment gateway integration.

6. **Financial Tracking App**: This task involves developing a comprehensive tool for managing personal finances. It should include functionalities for tracking expenses, incomes, and investments, as well as setting and adjusting budget goals. Key features include the ability for users to create and manage a personal account with secure linking of bank accounts and multi-factor authentication. The application should support both manual and automatic import of expenses and incomes, categorization of these entries, and visualization of their history. Budget management is a crucial aspect, with capabilities to set monthly budget goals, receive alerts when nearing these limits, and analyze spending patterns to suggest adjustments. The integration with investment accounts to track performance and balance, along with an overview of asset allocation, is also required. Additionally, the application should generate financial reports, like monthly summaries, and provide customizable alerts for unusual spending or important reminders.

Each method receives the same user-specified feature requirements, which are given below for each environment task, and produces a codebase output, which we then evaluate using our evaluation methodology, which is fully detailed in Appendix G. The following user-specified input prompts for each task are:

**URL Shortener App**, user-specified feature requirements input prompt:

```
### **Online URL Shortening Service**

**Overview**:
A service that allows users to submit long URLs and then receive a shortened version of that URL for ease of sharing.

**Functional Requirements to implement**:

1. **URL Shortening**:
   - [ ] 1.1. Users can input a URL to be shortened.
   - [ ] 1.2. The system validates that the URL is active and legitimate.
   - [ ] 1.3. The system generates a unique shortened URL.
   - [ ] 1.4. Users can choose custom short links (subject to availability).

2. **Redirection**:
   - [ ] 2.1. Accessing the shortened URL redirects to the original URL.

3. **Analytics**:
   - [ ] 3.1. Users can view statistics about their shortened URLs.
   - [ ] 3.2. View number of clicks.
   - [ ] 3.3. View date/time of each click.
   - [ ] 3.4. View geographical location of the clicker.

4. **User Accounts**:
   - [ ] 4.1. Users can create accounts.
   - [ ] 4.2. Account holders can view all their shortened URLs.
   - [ ] 4.3. Account holders can edit or delete their shortened URLs.
   - [ ] 4.4. Account holders can view analytics for all their shortened URLs.

5. **Admin Dashboard**:
   - [ ] 5.1. Administrators can view all shortened URLs.
   - [ ] 5.2. Administrators can delete any URL or user account.
   - [ ] 5.3. Administrators can monitor system performance and analytics.

6. **Expiration**:
   - [ ] 6.1. Users can set an expiration date/time for the shortened URL.
```

**Online Microblogging App**, user-specified feature requirements input prompt:

```
**Online Microblogging Service (OMS) - Description & Requirements**

*Description:*
A web-based platform where registered users can post short text-based messages, view messages from others, follow/unfollow other
    users, and interact with posts. The platform provides features for both individual users and administrative management.

**Functional Requirements to implement**:

1. **User Management:**
   1. **Registration & Authentication:**
      - [ ] Allow users to register using email, username, and password.
      - [ ] Option to reset forgotten passwords.
      - [ ] Secure authentication using JWT or similar protocols.
   2. **Profile Management:**
      - [ ] Users can edit their profile information: profile picture, bio, website link, and location.
      - [ ] Option to make profile private or public.

2. **Posting & Content Management:**
   1. **Creating Posts (Tweets):**
      - [ ] Allow users to create text-based posts with a limit of 280 characters.
      - [ ] Option to include images in posts.
      - [ ] Users can delete their own posts.
   2. **Interacting with Posts:**
      - [ ] Users can like, retweet, and reply to posts.
      - [ ] Nested comment structure for post replies.
   3. **Content Filtering & Search:**
      - [ ] Users can search for specific posts or users using keywords.
      - [ ] Filter option based on hashtags, user mentions, or trending topics.
```

3. **Social Interaction:**
    1. **Following & Followers:**
        – [ ] Users can follow/unfollow other users.
        – [ ] A timeline view displays posts from followed users.
        – [ ] Users receive notifications for new followers.
    2. **Direct Messaging:**
        – [ ] Private conversation threads between users.
        – [ ] Option to block/unblock users from messaging.
    3. **Notifications:**
        – [ ] Users are notified of likes, retweets, replies, and mentions.

4. **Trending & Discovery:**
    1. **Trending Topics:**
        – [ ] System identifies and displays trending hashtags or topics based on volume and velocity of mentions.
        – [ ] Trending topics can be sorted based on location or globally.
    2. **User Recommendations:**
        – [ ] Recommend users to follow based on interests, activity, and mutual followers.

## Online Chat App, user-specified feature requirements input prompt:

**Global Chat Service (GCS)**

**Overview**:
A real-time online chat application allowing users to send text messages, images, and create group chats.

**Functional Requirements to implement**:

User Registration and Authentication:
– [ ] 1.1. Sign up using email.
– [ ] 1.2. Forgotten password recovery.

User Profile:
– [ ] 2.1. Allow users to set profile pictures and status messages.
– [ ] 2.2. Privacy settings for who can see user details or last seen status.

Contact Management:
– [ ] 3.1. Block/unblock contacts.
– [ ] 3.2. Create, edit, and manage groups.

Messaging:
– [ ] 4.1. Send and receive real-time text messages.
– [ ] 4.2. Message read receipts (blue ticks or equivalent).
– [ ] 4.3. End-to-end encryption for security.
– [ ] 4.4. Image sharing.
– [ ] 4.5. Emojis, GIFs, and stickers support.

Group Chats:
– [ ] 5.1. Create group chats with a name and picture.
– [ ] 5.2. Add or remove participants.
– [ ] 5.3. Admin roles and permissions.

Status/Story Feature:
– [ ] 6.1. Allow users to post image statuses visible for a limited time.
– [ ] 6.2. Control who can see the status.

Web Application:
– [ ] 7.1. Web-based version accessible from browsers.

Connectivity and Offline Mode:
– [ ] 8.1. Message queuing for when the user is offline; messages are sent once connectivity is restored.
– [ ] 8.2. Display online/offline status.

## Recipe App, user-specified feature requirements input prompt:

### **Recipe Sharing Platform**

**Overview**:
A service that allows users to submit, share, and discover recipes. It includes features for categorizing recipes, user ratings and reviews, and personal recipe management.

**Functional Requirements to implement**:

1. **Recipe Submission and Management**:
    – [ ] 1.1. Users can submit recipes with ingredients, instructions, and images.
    – [ ] 1.2. Recipe submissions include options for categorization (e.g., cuisine type, dietary restrictions).
    – [ ] 1.3. Users can edit or delete their submitted recipes.
    – [ ] 1.4. Recipe format validation to ensure complete information.

2. **Search and Categorization**:
    – [ ] 2.1. Users can search for recipes based on ingredients, recipe name, or categories.
    – [ ] 2.2. Categorization of recipes by type (e.g., breakfast, lunch, dinner), cuisine, or dietary needs (e.g., vegan, gluten-free).

3. **User Accounts and Profiles**:
    – [ ] 3.1. Users can create and manage accounts.
    – [ ] 3.2. Account holders can save favorite recipes.
    – [ ] 3.3. Profile pages showing submitted recipes and favorite recipes.

4. **Ratings and Reviews**:
    – [ ] 4.1. Users can rate recipes on a 5-star scale.
    – [ ] 4.2. Users can write reviews for recipes.
    – [ ] 4.3. Display of average rating on recipe pages.

5. **Community Features**:
    – [ ] 5.1. Users can follow other users or chefs.
    – [ ] 5.2. Feed showing recent activity of followed users (new recipes, ratings).
    – [ ] 5.3. Option to share recipes on social media platforms.

6. **Admin Dashboard**:
    – [ ] 6.1. Administrators can manage all submitted recipes.
    – [ ] 6.2. Administrators can remove inappropriate content.
    – [ ] 6.3. Monitoring of site usage statistics and user engagement.

7. **Recipe Recommendations**:

```
    – [ ] 7.1. System generates recipe recommendations based on user preferences and past activity.
    – [ ] 7.2. Users receive notifications for new recipes in their interest areas.
```

**Event Planner App**, user-specified feature requirements input prompt:

```
### **Custom Event Planner Tool**

**Overview**:
A web-based application designed to assist users in organizing and managing various aspects of event planning. This tool should
    provide functionalities for selecting event types, managing guest lists, sourcing venues, and coordinating with service
    providers.

**Functional Requirements to Implement**:

1. **Event Creation and Management**:
    – [ ] 1.1. Users can create a new event, specifying details like event type, date, and time.
    – [ ] 1.2. The system allows for customization of events (e.g., themes, color schemes).
    – [ ] 1.3. Users can update or modify event details as needed.
    – [ ] 1.4. A calendar view is available for users to manage multiple events.

2. **Venue Sourcing**:
    – [ ] 2.1. Users can search for venues based on location, capacity, and type.
    – [ ] 2.2. Integration of maps for venue locations.
    – [ ] 2.3. Users can book venues directly through the application.

3. **Guest List Management**:
    – [ ] 3.1. Users can create and manage guest lists.
    – [ ] 3.2. Import/export guest list feature.
    – [ ] 3.3. RSVP tracking and management.

4. **Vendor Coordination**:
    – [ ] 4.1. Platform to connect with various event service providers (caterers, decorators).
    – [ ] 4.2. Users can view and compare vendor profiles and reviews.
    – [ ] 4.3. In-app messaging system for vendor communication.

5. **Budget Management**:
    – [ ] 5.1. Users can set a budget for the event.
    – [ ] 5.2. Budget tracking and breakdown by categories (venue, catering, etc.).
    – [ ] 5.3. Alerts for budget overruns.

6. **User Accounts and Profiles**:
    – [ ] 6.1. Users can create personal profiles.
    – [ ] 6.2. Profile customization to reflect event planning preferences.
    – [ ] 6.3. Saving and accessing past and upcoming events.

7. **Notifications and Reminders**:
    – [ ] 7.1. Automated email/SMS notifications for event milestones.
    – [ ] 7.2. Customizable reminders for tasks and deadlines.

8. **Reporting and Analytics**:
    – [ ] 8.1. Generate reports on event success metrics (attendance, budget adherence).
    – [ ] 8.2. Feedback collection from guests and vendors post-event.

9. **Admin Dashboard**:
    – [ ] 9.1. Administrators can monitor and manage user activities.
    – [ ] 9.2. System performance analytics and user engagement statistics.
    – [ ] 9.3. Manage vendor listings and platform content.

10. **Security and Data Privacy**:
    – [ ] 10.1. Ensuring user data protection and privacy.
    – [ ] 10.2. Secure payment gateway integration for transactions.
```

**Financial Tracking App**, user-specified feature requirements input prompt:

```
### **Personal Finance Tracking Application**

**Overview**:
A comprehensive tool for managing personal finances, including tracking expenses, incomes, investments, and setting budget goals.

**Functional Requirements to implement**:

1. **Account and Security**:
    – [ ] 1.1. Users can create and manage their personal account.
    – [ ] 1.2. Secure linking of bank accounts.
    – [ ] 1.3. Multi-factor authentication for enhanced security.

2. **Expense and Income Tracking**:
    – [ ] 2.1. Manual and automatic import of expenses and incomes.
    – [ ] 2.2. Categorization of expenses and income sources.
    – [ ] 2.3. Visualization of expense and income history.

3. **Budget Management**:
    – [ ] 3.1. Setting and adjusting monthly budget goals.
    – [ ] 3.2. Alerts for nearing budget limits.
    – [ ] 3.3. Analysis of spending patterns to suggest budget adjustments.

4. **Investment Overview**:
    – [ ] 4.1. Integration with investment accounts.
    – [ ] 4.2. Tracking investment performance and balance.
    – [ ] 4.3. Overview of asset allocation.

5. **Reports and Alerts**:
    – [ ] 5.1. Generation of financial reports (e.g., monthly summaries).
    – [ ] 5.2. Customizable alerts for unusual spending or important reminders.
```

## E.2    HUMANEVAL BENCHMARK

We use the standard HumanEval benchmark, as introduced by Chen et al. (2021). This benchmark evaluates the task of the underlying LLM-based method to generate standalone Python functions

from a given docstring, and evaluates the correctness of the generated code function through held out unit tests. The benchmark consists of 164 hand-written programming problems with unit tests, which aim to assess language comprehension, algorithms, and simple mathematics, of which the authors (Chen et al., 2021) compare these to simple software interview questions.

# F   BENCHMARK METHOD IMPLEMENTATION DETAILS

In the following, we detail all implementation details of our benchmark methods, including that of the full low-level implementation details of Code-L2MAC in Appendix F.1.

**AutoGPT**. We chose to compare against the most competitive and popular autonomous agent LLM with memory method, that of **AutoGPT** (Richards, 2023). This uses GPT4 as the underlying LLM. Like other autonomous agents, the first step of AutoGPT is to summarize the user-specified task into a name, role, and five one-sentence goals for the agent to complete; this summary is kept in context throughout execution. At each step, until the goals have been achieved, the underlying LLM receives a ReACT (Yao et al., 2022) message template to reason, plan, and act by selecting the next tool to call. Specifically, we used the latest stable version of AutoGPT, 0.4.7, enabled local tool execution, and disabled searching the web for answers to ensure a fair comparison. It is interesting to note that AutoGPT also has tool functionality and has tools to read, write, and list files in a local file store—making it the most competitive and applicable baseline to compare against. Although other autonomous agent LLM methods exist, they do not possess the ability to read or write to an external file store in the same manner as AutoGPT. Importantly, AutoGPT, when running, would sometimes have runs where it would fail to complete the task, i.e., fail to output any code or get stuck in an infinite loop. When such failures occurred, we classified these runs failures and excluded them from the results. This method defaults to use the LLM with a temperature of 0.

**GPT4**. We also chose to compare with the single GPT4 LLM (**GPT4**) (OpenAI, 2023), which we adapt to make it competitive by providing it with the same tools that Code–L2MAC uses, forming an ablation of just using an LLM processor without the control unit. Specifically, we provide it with exactly the same messages and experimental setup as Code-L2MAC, including the same system message and prompt format message; however, it is limited to only respond with one long single response that can fill up the entire context window $C^t$.

**CodeT**. We build upon our GPT4 method implementation above and implement CodeT a recent state-of-the-art method for code snippet generation (Chen et al., 2022). CodeT samples independent generated code outputs, and uses the same LLM to generate test cases for the code samples, it them executes the code samples using the generated test cases and performs a dual execution agreement, to return the output code sample that performs the best against the generated tests. To make this method more competitive when applied to large codebase generation tasks, it is possible generate a given codebase statisfying the user given feature requirements, however with many different possible implementation approaches (of splitting components, classes and across files and folders)—therefore instead of sampling tests independently of the generated codebase, we follow our setup with GPT4 and Code-L2MAC and generate a codebase and unit tests at the same time so they match the given implementation. We repeat this generation independently for $n = 3$ times, and run the codebase against the generated tests, and rank the codebases in order of the number of tests that pass, and return the codebase that passes the highest number of self generated tests.

**Self-Refine**. Similarly, we also build upon our GPT4 method implementation above and implement Self-Refine, another recent state-of-the-art method for code snippet generation (Madaan et al., 2023). Self-Refine, uses iterative self-refinement that alternates between using an underlying LLM to provide feedback and refine a given generated LLM output, to create a higher quality output. Specifically, given an initial LLM generated codebase output, we use the same LLM to provide verbal feedback on how it could have improved the codebase to the given original task, and then using this verbal feedback within the same context window, we refine the previously generated codebase. This process is repeated $n = 3$ times, as in line with $n = 3$ used in by Madaan et al. (2023). We followed the same coding task setup as in Madaan et al. (2023) to make this method competitive. Furthermore to make it more competitive, after generating the codebase, we use the same error checking methods in Code-L2MAC and include in context the result of any code errors found and the result of any tests generated that are failing, so it can also use this signal to improve upon the generated codebase.

**Reflexion**. Likewise, we also build upon our GPT4 method implementation above and implement Reflexion, another recent state-of-the-art method for code snippet generation (Madaan et al., 2023). Reflexion, another reflecting LLM approach, uses the same LLM to reflect on the trajectory (outputs of an LLM) once an environment episode (task) has been completed to provide verbal feedback on how the task could have been completed better. These verbal feedback signals are then persisted in an episodic append-only (list) memory buffer and are given in context upon the next episode of completing the task from scratch (re-starting the environment). Reflexion converts a binary or scalar feedback from the environment into its verbal feedback, that is then used as additional context for the LLM in the next episode. To make Reflexion more competitive, after generating the codebase we use the same error checking methods in Code-L2MAC and include in context the result of any code errors found and the result of any tests generated that are failing appended to the context after generating the codebase. Specifically, Reflexion follows an iterative process where it generates a codebase given the same initial prompt as Code-L2MAC and existing methods, then it is asked to evaluate the codebase for the number of features fully and functionally implemented as listed in the input prompt, giving a scalar score (the reward signal for the environment), which is apended to the context window. This is followed by the self-reflection step, where it is asked to analyze the trajectory (codebase, the evaluated scalar feedback) to produce a summary on how it could improve (verbal experience feedback for that episode). This summary is then stored in an external buffer. We then reset the context window window, starting with the initial task prompt plus the verbal lessons from the external buffer, and repeat this iterative process $n = 3$ times be be competitive. We follow Reflexion's code programming setup and prompts, adapting them to our task to be more competitive.

**Code-L2MAC**. Our proposed method uses the message and function definitions defined in Appendix F.1. Using these, it then follows the pseudocode for the control unit, as outlined in Algorithm 1, and generates a prompt program of instructions $\mathcal{I}$ and iterates each dialog turn of the control unit control flow loop until all instructions are complete. There are a few other minor implementation details, which we also detail here. We impose a maximum number of times an instruction can be re-tried $r_{\text{Max}}$, with a new context window when, during the execution of that instruction, it attempts to exceed the context window, forcing the context window to restart the instruction with the summary of the current progress made—we empirically set $r_{\text{Max}} = 30$. When the current number of retries exceeds $r_{\text{Max}}$, we class this run as a failure and exclude its results. Importantly, at such a failure, the method could ideally either await human input for correction or continue with an errored state. We used the LLM of GPT4-0613, and when using it throughout, set the temperature to 0.01. Additionally, another implementation detail is that when the CU detects that it is stuck in a loop repeating the same two messages over again by comparing the most recent two messages in the context window, it increases the temperature of the LLM by 0.1 and continues until the temperature caps at 1.0, and then after it exits the loop reducing the temperature back to 0.01. This is to have a form of simulated annealing of introducing randomness to escape the local minima. In the following, Appendix F.1, we detail special message templates exactly.

## F.1 CODE-L2MAC LOW LEVEL IMPLEMENTATION DETAILS

We follow the same setup as outlined in Appendix C, and in the following, we provide exact low-level implementation details. We use GPT-4-0613, which has fine-tuned support for function calls, and use the function definition file of the following.

Function definitions:

```
{
"name": "provide_detailed_sub_task_steps_for_sub_agents",
"description": "For producing a step-by-step plan, where each step paragraph is a detailed sub-task step for a separate
        sub-agent (large language model agent) to complete. Within each detailed step paragraph, always include a last
        sentence to create and run tests when implementing or writing code in that same step.",
"parameters": {
    "type": "object",
    "properties": {
        "steps": {
            "type": "array",
            "description": "List of strings, where each string is a separate step sub-task paragraph for a separate sub-
                agent to complete. Within each detailed step paragraph, always include a last sentence to create and
                run tests when implementing or writing code in that same step.",
            "items": {
                "type": "string",
            }
        },
    },
    "required": ["steps"],
},
},
{
"name": "sub_task_step_complete",
```

```
        "description": "Call this function when the user specified sub task step has been completed.",
        "parameters": {
            "type": "object",
            "properties": {},
        },
    },
    {
    "name": "view_files",
    "description": "Print out the file contents into the response to view.",
    "parameters": {
        "type": "object",
        "properties": {
            "files": {
                "type": "array",
                "description": "list of the files to view",
                "items": {
                    "type": "string"  # assuming each file is represented as a string
                }
            },
        },
        "required": ["files"],
    },
    },
    {
    "name": "run_python_file",
    "description": "Run python file and return the output to the response to view. That is with 'python3 file_name_to_run
            '.",
    "parameters": {
        "type": "object",
        "properties": {
            "file_name_to_run": {
                "type": "string",
                "description": "file name to run",
            },
            "arguments": {
                "type": "array",
                "description": "optional run arguments",
                "items": {
                    "type": "string"
                }
            },
        },
        "required": ["file_name_to_run"],
    },
    },
    {
    "name": "pytest_files",
    "description": "Run pytest on the input file names and print out the results to the response to view. If no file names
            are provided, pytest runs on all files.",
    "parameters": {
        "type": "object",
        "properties": {
            "files_to_test": {
                "type": "array",
                "description": "file names to run pytest on",
                "items": {
                    "type": "string"
                }
            },
        },
    },
    },
    {
    "name": "write_files",
    "description": "Write out multiple files and it will be combined into the existing codebase. Always output the whole
            file. You always indent code with tabs.",
    "parameters": {
        "type": "object",
        "properties": {
            "files_and_contents": {
                "type": "array",
                "description": 'list of files and their contents.',
                "items": {
                    "type": "object",
                    "properties": {
                        "file_path": {
                            "type": "string",
                            "description": "Path to the file"
                        },
                        "file_contents": {
                            "type": "string",
                            "description": "Contents of the file"
                        }
                    },
                    "required": ["file_path", "file_contents"]
                }
            }
        },
        "required": ["files_and_contents"],
    },
    },
    {
    "name": "delete_files",
    "description": "Delete files. Specify the file names, and these files will be deleted. If you specify the file name '-1'
            all files in the folder will be deleted.",
    "parameters": {
        "type": "object",
        "properties": {
            "files": {
                "type": "array",
                "description": "list of the files to delete. If you provide a file name of '-1' all files in the folder will
                        be deleted.",
                "items": {
                    "type": "string"
                }
            },
        },
    },
```

```
        "required": ["files"],
    },
}
```

With Code-L2MAC we use the following system message (OpenAI, 2023) $M_s$, and include it all context windows $C^t$, which is standard to do so (OpenAI, 2023).

System message $M_s$:

```
Objective: Write code for a large system design task.
Please note that the code should be fully functional. No placeholders.
Only use the functions you have been provided with.
Only use the 'write_files' to output code.

You must act autonomously and you will receive no human input at any stage. You have to return as output the complete code for
    completing this task, and correctly incorporate it into the existing codebase.
You always write out the whole file contents. You always indent code with tabs.
Please always view the files before writing to them, to make sure you are writing to the correct files.
When writing a test, make the filename start with the prefix 'test_'.

Provide the minimal code necessary to achieve the task conditioned on the existing generated code———including changing the existing
    generated code.

You cannot visualize any graphical output. You exist within a Actor Model machine, and when you list out steps, each step will be
    taken by a new separate sub-ChatGPT model. When you list out a sub-task steps, you can optionally specify the sub-task
    validation to check that it has been completed successfully.

You cannot use any databases as none are setup in the local environment, instead mock a database with an in memory dictionary to
    store data. No data saved to disk will persist between steps or write operations.

If a test is failing the error could be the code, or the test is incorrect, so feel free to overwrite and change the tests when they
    are incorrect, to make all tests pass.

Use the functions provided. When calling functions only provide a RFC8259 compliant JSON request following this format without
    deviation.
```

When Code-L2MAC self-programs its instructions initially, we provide it with the following prompt template to do so, which encapsulates the user message $M_u$, as 'user_specified_feature_requirements'.

$M_{cb}$ bootstrap instruction:

```
You will get instructions for code to write.
First lay out the names of the core classes, functions, methods that will be necessary, As well as a quick comment on their purpose.
Do not comment on what every file does. Please note that the code should be fully functional. No placeholders.

You will start with the "entrypoint" file, then go to the ones that are imported by that file, and so on.
Please note that the code should be fully functional. No placeholders.

Follow a language and framework appropriate best practice file naming convention.
Make sure that files contain all imports, types etc. The code should be fully functional. Make sure that code in different files
    are compatible with each other.
When writing code if you are unsure, write a plausible implementation.
Include module dependency or package manager dependency definition file.

Useful to know:

For Python, you always create an appropriate requirements.txt file.
Always add a comment briefly describing the purpose of the function definition.
Add comments explaining very complex bits of logic.
Always follow the best practices for the requested languages for folder/file structure and how to package the project.
You can use any package and any other packages you wish to install.
You cannot use any databases as none are setup in the local environment, instead mock a database with an in memory dictionary to
    store data. No data saved to disk will persis between steps or write operations.
When writing a test, make the filename start with the prefix 'test_'.

Python toolbelt preferences:
- pytest
- dataclasses
- flask

Objective:```
{user_specified_feature_requirements}
```

Understand the problem, by creating an extremely detailed step-by-step plan, where each step is long (multiple sentences) and in
    total includes every single feature requirement specified above, feel free to copy directly from it. Use no more than 10
    steps in the plan. Create additional tests, checks and evaluation at each step when applicable to help make an excellent code
    implementation, where all the code is fully functional. Use best software design practices, and you can output large amounts
    of code at once. Please include a last sentence to create and run tests when implementing or writing code in that same step.
    You will receive no human input at any stage, so you cannot use a human to test. Only create a detailed plan to begin with,
    which includes designing and running tests to check that they all pass. Please be sure to include all of the specified
    feature requirements in the following plan.
```

Code-L2MAC also involves control messages of the following form.

$M_{cc}$ Control cycle message of starting the instruction:

```
Objective: Execute sub task step: {instruction}.\n\n Note: Condition any new code files on the existing code files: {list_files()}.
    Fully implement these features in the code, no placeholders. You can now optionally view the existing files if you need to
    view them to complete the current task step. You have a limited context window so be selective about which files you view,
    only view the files you think you might need to view.\n\nSummary output of previous step: ""{
    previous_instruction_output_summary}""\n\nRespond now only with a function call of one of the following functions provided: {
    function_names()}, and if you want to output code only use the 'write_files' function to output code.
```

Here instruction is the current instruction of operation, 'list_files()' a list files tool function, previous_instruction_output_summary is the summary message $M_{rs}$, and 'function_names()' lists the current function names that the LLM can request.

$M_{cc}$ Control cycle message of continuing the instruction:

```
Has the sub task step been completed of: ```
{instruction}
``` \n\n If yes, call the function `sub_task_step_complete`, otherwise reflect and correct the full code to complete the task. Only
     use the functions you have been provided with, and if you want to output code only use the `write_files` function to output
     code. Condition it on existing code: {list_files()} Fully implement these features in the code, no placeholders. If you have
     not viewed the files before writing to them, please view them, to make sure you are writing to the correct files.\nRespond
     now only with a function call of one of the following functions provided: {function_names()}, and if you want to output code
     only use the `write_files` function to output code.
```

Here instruction is the current instruction of operation, 'sub_task_step_complete' is the function the LLM calls to indicate that it has completed the current instruction, 'list_files()' a list files tool function, and 'function_names()' lists the current function names that the LLM can request.

$M_{cs}$ Control summarization message for restarting the same instruction:

```
Please provide a one or two sentence summary of the output of this step, which is useful for the next step. Your response will be
     used when starting the next step without any of the previous messages.
```

$M_{cs}$ Control summarization message, for summarizing the instruction output—which is used when continuing the next instruction.

```
You have exhausted your context window. Reflect on your progress. Provide a short concise response, of two sentences maximum, this
     will be used to restart this step from the beginning without the previous messages.
```

# G  EVALUATION METRICS

There are no automated tools to validate the output of system design tasks, so we propose evaluation metrics to compare the output of different methods. Large-scale code generation is unique in that the generated code can satisfy the high-level user feature requirements specified in the input through various possible implementation approaches.

To quantify the degree to which user-specified features in the initial prompt are effectively implemented in the generated code, we introduce a performance metric named **Features %**. This metric numerically represents the proportion of input features that are fully and functionally implemented in the output codebase. The **Features %** is obtained by using a separate GPT-4 API call, which iteratively examines the entire generated code to verify the functional implementation of all input features, counting the number of fully implemented features. We quote this as a percentage of the features implemented over the total features specified by the user. Specifically, we implement this metric by first collecting all the generated code files from a method's generated output and outputting all the code files as named strings of the format file name, followed by the file contents, and pass it into the prompt template below for the GPT-4 API to count the number of fully implemented features. We then use a regex parser to extract the final number from the LLM response message. We performed a human study in Appendix H.6 and find that this evaluation metric correlates to that performed by human professional software engineers performing a manual code review counting the same number of features implemented.

We now provide this prompt template and then provide an example of it in action.

**Features %** prompt template

```
Objective: Based on the numbered features given, you are to evaluate the following code and return a numeric value for how many (a
     count) of those numbered features are implemented in the provided code. Give the numeric answer as "FEATURES_FUNCTIONAL=
     num_features_functional" in the final line.

Numbered Features Specified:```
{user_specified_feature_requirements}
```

Code to evaluate for the amount of features fully implemented:"""
{code_files}
"""
```

Here 'user_specified_feature_requirements' is the user-specified task description, which are provided in full in Appendix E, and 'code_files' is the entire contents of the codebase printed out with filenames and the contents.

A worked example, can be seen in the following, for example Code-L2MAC for one of the runs for the environment task URL Shortener App, with the complete user-specified feature requirements is given, including the GPT4 call's response.

## GPT4 API call.

```
To evaluate the provided codebased on the requirements for the "Online URL Shortening Service", you can use the following numbered
        checklist. For each feature present in the code, you can increment a numeric value:

```
**Evaluation Checklist for Online URL Shortening Service**:

1. **URL Shortening**:
    - [ ] 1.1. Users can input a URL to be shortened.
    - [ ] 1.2. The system validates that the URL is active and legitimate.
    - [ ] 1.3. The system generates a unique shortened URL.
    - [ ] 1.4. Users can choose custom short links (subject to availability).

2. **Redirection**:
    - [ ] 2.1. Accessing the shortened URL redirects to the original URL.

3. **Analytics**:
    - [ ] 3.1. Users can view statistics about their shortened URLs.
    - [ ] 3.2. View number of clicks.
    - [ ] 3.3. View date/time of each click.
    - [ ] 3.4. View geographical location of the clicker.

4. **User Accounts**:
    - [ ] 4.1. Users can create accounts.
    - [ ] 4.2. Account holders can view all their shortened URLs.
    - [ ] 4.3. Account holders can edit or delete their shortened URLs.
    - [ ] 4.4. Account holders can view analytics for all their shortened URLs.

5. **Admin Dashboard**:
    - [ ] 5.1. Administrators can view all shortened URLs.
    - [ ] 5.2. Administrators can delete any URL or user account.
    - [ ] 5.3. Administrators can monitor system performance and analytics.

6. **Expiration**:
    - [ ] 6.1. Users can set an expiration date/time for the shortened URL.

**Total Implemented Features**: [Count the checked boxes to get the numeric value]
```

You can go through the provided code and check off each item that has been implemented. At the end, count the checked boxes to get
        the numeric value for the number of features implemented.
```

Code to evaluate for the amount of features fully implemented:"""
test_analytics.py
```python
import analytics
import datetime

def test_track_click():
        """Test that click events are tracked correctly."""
        analytics.track_click('abc123', '8.8.8.8')
        data = analytics.get_analytics('abc123')

        assert len(data) == 1
        assert isinstance(data[0]['time'], datetime.datetime)
        assert data[0]['ip_address'] == '8.8.8.8'

def test_get_analytics():
        """Test that analytics data is retrieved correctly."""
        data = analytics.get_analytics('abc123')

        assert len(data) == 1
        assert isinstance(data[0]['time'], datetime.datetime)
        assert data[0]['ip_address'] == '8.8.8.8'

```

analytics.py
```python
import datetime

# Mock database
analytics_data = {}

def track_click(short_url, ip_address):
        """Track a click event."""
        if short_url not in analytics_data:
                analytics_data[short_url] = []

        # Record the click event
        click_event = {
                'time': datetime.datetime.now(),
                'ip_address': ip_address
        }
        analytics_data[short_url].append(click_event)

def get_analytics(short_url):
        """Retrieve analytics data for a short URL."""
        return analytics_data.get(short_url, [])

def get_system_performance():
        """Retrieve system performance data.

        Returns:
```

```
                dict: A dictionary containing system performance data.
        """
        # For simplicity, we'll just return the number of URLs and the number of click events.
        return {
                'num_urls': len(analytics_data),
                'num_clicks': sum(len(events) for events in analytics_data.values())
        }
```

test_user_accounts.py
```python
import pytest
from user_accounts import UserAccount

def test_create_account():
        user_account = UserAccount()
        assert user_account.create_account('test', 'password') == 'Account created successfully.'
        assert user_account.create_account('test', 'password') == 'Username already exists.'

def test_view_urls():
        user_account = UserAccount()
        user_account.create_account('test', 'password')
        assert user_account.view_urls('test') == []
        assert user_account.view_urls('nonexistent') == 'Username does not exist.'

def test_add_url():
        user_account = UserAccount()
        user_account.create_account('test', 'password')
        assert user_account.add_url('test', 'http://example.com') == 'URL added successfully.'
        assert user_account.add_url('nonexistent', 'http://example.com') == 'Username does not exist.'

def test_delete_url():
        user_account = UserAccount()
        user_account.create_account('test', 'password')
        user_account.add_url('test', 'http://example.com')
        assert user_account.delete_url('test', 'http://example.com') == 'URL removed successfully.'
        assert user_account.delete_url('test', 'http://nonexistent.com') == 'URL does not exist.'
        assert user_account.delete_url('nonexistent', 'http://example.com') == 'Username does not exist.'
```

url_shortener.py
```python
import hashlib
import requests
from datetime import datetime, timedelta

# Mock database to store the mapping between the shortened URLs and the original URLs.
url_mapping = {}

def generate_short_url(url, expiration_minutes=15):
        """Generate a unique shortened URL.

        Args:
                url (str): The original URL.
                expiration_minutes (int): The number of minutes until the URL expires.

        Returns:
                str: The shortened URL.
        """
        hash_object = hashlib.md5(url.encode())
        short_url = hash_object.hexdigest()[:10]

        # Store the mapping between the shortened URL and the original URL.
        url_mapping[short_url] = {
                'url': url,
                'expiration': datetime.now() + timedelta(minutes=expiration_minutes)
        }

        return short_url

def validate_url(url):
        """Validate a URL.

        Args:
                url (str): The URL to validate.

        Returns:
                bool: True if the URL is valid, False otherwise.
        """
        try:
                response = requests.get(url)
                return response.status_code == 200
        except requests.exceptions.RequestException:
                return False

def get_original_url(short_url):
        """Get the original URL associated with the shortened URL.

        Args:
                short_url (str): The shortened URL.

        Returns:
                str: The original URL, or None if the shortened URL does not exist or has expired.
        """
        url_data = url_mapping.get(short_url)
        if url_data and url_data['expiration'] > datetime.now():
                return url_data['url']

def delete_url(short_url):
        """Delete a shortened URL.
```

```
        Args:
                short_url (str): The shortened URL.
        """
        if short_url in url_mapping:
                del url_mapping[short_url]

def get_all_urls():
        """Get all shortened URLs.

        Returns:
                list: A list of all shortened URLs.
        """
        return list(url_mapping.keys())
```

test_url_shortener.py
```python
import url_shortener
from datetime import datetime, timedelta
import time

def test_generate_short_url():
        """Test the generate_short_url function."""
        url = 'https://www.example.com'
        short_url = url_shortener.generate_short_url(url)
        assert len(short_url) == 10
        assert isinstance(short_url, str)

def test_validate_url():
        """Test the validate_url function."""
        valid_url = 'https://www.example.com'
        invalid_url = 'https://www.invalidurl.com'
        assert url_shortener.validate_url(valid_url) is True
        assert url_shortener.validate_url(invalid_url) is False

def test_url_expiration():
        """Test that URLs expire correctly."""
        url = 'https://www.example.com'
        short_url = url_shortener.generate_short_url(url, expiration_minutes=0)
        assert url_shortener.get_original_url(short_url) is None

        short_url = url_shortener.generate_short_url(url, expiration_minutes=1)
        assert url_shortener.get_original_url(short_url) == url

        # Wait for the URL to expire
        time.sleep(61)
        assert url_shortener.get_original_url(short_url) is None
```

user_accounts.py
```python
class UserAccount:
        def __init__(self):
                self.accounts = {}

        def create_account(self, username, password):
                if username in self.accounts:
                        return 'Username already exists.'
                self.accounts[username] = {'password': password, 'urls': []}
                return 'Account created successfully.'

        def view_urls(self, username):
                if username not in self.accounts:
                        return 'Username does not exist.'
                return self.accounts[username]['urls']

        def add_url(self, username, url):
                if username not in self.accounts:
                        return 'Username does not exist.'
                self.accounts[username]['urls'].append(url)
                return 'URL added successfully.'

        def delete_url(self, username, url):
                if username not in self.accounts:
                        return 'Username does not exist.'
                if url not in self.accounts[username]['urls']:
                        return 'URL does not exist.'
                self.accounts[username]['urls'].remove(url)
                return 'URL removed successfully.'

        def delete_account(self, username):
                if username in self.accounts:
                        del self.accounts[username]
                        return 'Account deleted successfully.'
                else:
                        return 'Username does not exist.'

        def get_all_users(self):
                return list(self.accounts.keys())
```

requirements.txt
```text
flask
pytest
maxminddb-geolite2
```

test_app.py
```python
import pytest
from app import app

@pytest.fixture
def client():
        app.config['TESTING'] = True
        with app.test_client() as client:
                yield client

def test_home_page(client):
        response = client.get('/')
        assert response.status_code == 200
        assert response.data == b'Hello, World!'

def test_redirect_to_original_url(client):
        response = client.get('/abc123')
        assert response.status_code == 404

def test_shorten_url(client):
        response = client.post('/shorten', data={'url': 'https://www.google.com'})
        assert response.status_code == 200
        assert 'short_url' in response.get_json()

def test_view_analytics(client):
        response = client.get('/analytics/abc123')
        assert response.status_code == 200
        assert 'analytics' in response.get_json()

def test_manage_account(client):
        response = client.post('/account', data={'action': 'create', 'username': 'testuser', 'password': 'testpass'})
        assert response.status_code == 200
        assert response.get_json()['message'] == 'Account created successfully.'

        response = client.post('/account', data={'action': 'view_urls', 'username': 'testuser'})
        assert response.status_code == 200
        assert 'urls' in response.get_json()

        response = client.post('/account', data={'action': 'delete', 'username': 'testuser'})
        assert response.status_code == 200
        assert response.get_json()['message'] == 'Account deleted successfully.'

def test_admin_dashboard(client):
        response = client.get('/admin')
        assert response.status_code == 200
        assert 'urls' in response.get_json()
        assert 'users' in response.get_json()
        assert 'performance' in response.get_json()

def test_delete_url(client):
        client.post('/admin', data={'action': 'delete_url', 'short_url': 'abc123'})
        response = client.get('/abc123')
        assert response.status_code == 404

def test_delete_user(client):
        client.post('/admin', data={'action': 'delete_user', 'username': 'testuser'})
        response = client.get('/admin')
        assert 'testuser' not in response.get_json()['users']

```

app.py
```python
from flask import Flask, redirect, url_for, request
from url_shortener import get_original_url, delete_url, get_all_urls, generate_short_url, validate_url
from user_accounts import UserAccount
from analytics import get_system_performance, get_analytics, track_click

app = Flask(__name__)

user_account = UserAccount()

@app.route('/')
def home():
        return 'Hello, World!'

@app.route('/<short_url>')
def redirect_to_original_url(short_url):
        original_url = get_original_url(short_url)
        if original_url is None:
                return 'URL not found', 404
        else:
                track_click(short_url, request.remote_addr)
                return redirect(original_url)

@app.route('/shorten', methods=['POST'])
def shorten_url():
        url = request.form.get('url')
        if not validate_url(url):
                return 'Invalid URL', 400
        short_url = generate_short_url(url)
        username = request.form.get('username')
        if username:
                user_account.add_url(username, short_url)
        return {'short_url': short_url}

@app.route('/analytics/<short_url>')
```

```
def view_analytics(short_url):
        return {'analytics': get_analytics(short_url)}

@app.route('/account', methods=['POST'])
def manage_account():
        action = request.form.get('action')
        username = request.form.get('username')
        if action == 'create':
                password = request.form.get('password')
                return {'message': user_account.create_account(username, password)}
        elif action == 'view_urls':
                return {'urls': user_account.view_urls(username)}
        elif action == 'delete':
                return {'message': user_account.delete_account(username)}

@app.route('/admin', methods=['GET', 'POST'])
def admin_dashboard():
        if request.method == 'POST':
                action = request.form.get('action')
                if action == 'delete_url':
                        short_url = request.form.get('short_url')
                        delete_url(short_url)
                elif action == 'delete_user':
                        username = request.form.get('username')
                        user_account.delete_account(username)
                return redirect(url_for('admin_dashboard'))
        else:
                all_urls = get_all_urls()
                all_users = user_account.get_all_users()
                performance_data = get_system_performance()
                return {'urls': all_urls, 'users': all_users, 'performance': performance_data}

if __name__ == '__main__':
        app.run(debug=True)

...
```

### GPT4 API response

```
The provided code implements the following features:

1. **URL Shortening**:
   - [x] 1.1. Users can input a URL to be shortened.
   - [x] 1.2. The system validates that the URL is active and legitimate.
   - [x] 1.3. The system generates a unique shortened URL.
   - [ ] 1.4. Users can choose custom short links (subject to availability).

2. **Redirection**:
   - [x] 2.1. Accessing the shortened URL redirects to the original URL.

3. **Analytics**:
   - [x] 3.1. Users can view statistics about their shortened URLs.
   - [x] 3.2. View number of clicks.
   - [x] 3.3. View date/time of each click.
   - [ ] 3.4. View geographical location of the clicker.

4. **User Accounts**:
   - [x] 4.1. Users can create accounts.
   - [x] 4.2. Account holders can view all their shortened URLs.
   - [x] 4.3. Account holders can edit or delete their shortened URLs.
   - [x] 4.4. Account holders can view analytics for all their shortened URLs.

5. **Admin Dashboard**:
   - [x] 5.1. Administrators can view all shortened URLs.
   - [x] 5.2. Administrators can delete any URL or user account.
   - [x] 5.3. Administrators can monitor system performance and analytics.

6. **Expiration**:
   - [x] 6.1. Users can set an expiration date/time for the shortened URL.

Counting the checked boxes, we get a total of 15 features implemented.

FEATURES_FUNCTIONAL=15
```

The final line of the response, 'FEATURES_FUNCTIONAL' is then extracted by a regex parser. We note that the use of the feature requirements specified as a checklist helps the LLM check which features are fully implemented and which features are not and aids in counting the number of features that have been fulfilled in the generated codebase according to the user-specified input feature requirements.

Furthermore, we also incorporate standard code generation evaluation metrics (Hasemer, 2018), such as the number of lines of code generated **LOC** and the number of errors **# Errors** in the codebase determined by a code syntactical analyzer (Thénault, 2023). Specifically, we use PyLint, a well-known Python syntactical analyzer, and configure it to only detect and count the number of errors in the code, which becomes static syntax errors that arise from the generated outputs. We configure it with the following options of '–disable=all –enable=E'.

Moreover, as the methods are instructed to generate unit tests to check that the generated code is valid, we also quote how many self-generated unit tests pass with the method as **Tests Passed**. We compute this metric by running a Python test tool, that of PyTest (Okken, 2022). This is configured to automatically find all relevant python tests (default configuration), and the LLM can call this separately as a function on only code path input.

We also include the metric of code coverage (Miller & Maloney, 1963) as **Cov %**. This measures the percentage of lines of code that are executed during running all the self-generated unit tests for the generated codebase out of the total executable lines of code. A codebase with a high test coverage percentage has more of its code executed during running the unit tests, indicating that it has a lower chance of containing undetected software bugs compared to a codebase with a low test coverage percentage. We practically test this with a python code coverage tool, that of Coverage (Batchelder & Contributors to Coverage.py, 2023), and only run this at evaluation time when evaluating the codebase.

Experimental setup. We complete 10 random seed runs for each task environment for each benchmark method. We then compute each metric alongside its 95% confidence intervals throughout.

## G.1 MOTIVATION FOR FEATURE %

To elucidate the motivation of Feature %, we note that the metric "Is it fully functional?" is very sparse and does not reflect the degrees of correctness that an implementation can have. As an extreme case, it is tough to prove that a given codebase contains no bugs, and in practice, most codebases contain bugs. This does not imply that they are not functional, although each time a bug is correctly fixed, we can agree that the codebase has gotten more functional. Consequently, asking, "To what degree is it functional?" is more reasonable. A way to quantify this is to ask "Of the functions/features that I expect this codebase to fulfill correctly, how many are actually fulfilled?". This is what feature % quantifies. As a proxy, and in line with previous literature (Chiang & Lee, 2023), we quantify this by having GPT4 assess what features specified by the user were implemented fully as code (see Appendices E and G for example prompts).

## H ADDITIONAL RESULTS

### H.1 HUMANEVAL BENCHMARK RESULTS

We also evaluated Code-L2MAC on the standard HumanEval benchmark (Chen et al., 2021) and found that it achieves a state-of-the-art score of 90.2% Pass@1. We briefly describe the benchmark here (with a complete description given in Appendix E.2), and provide a comparative analysis of existing methods here.

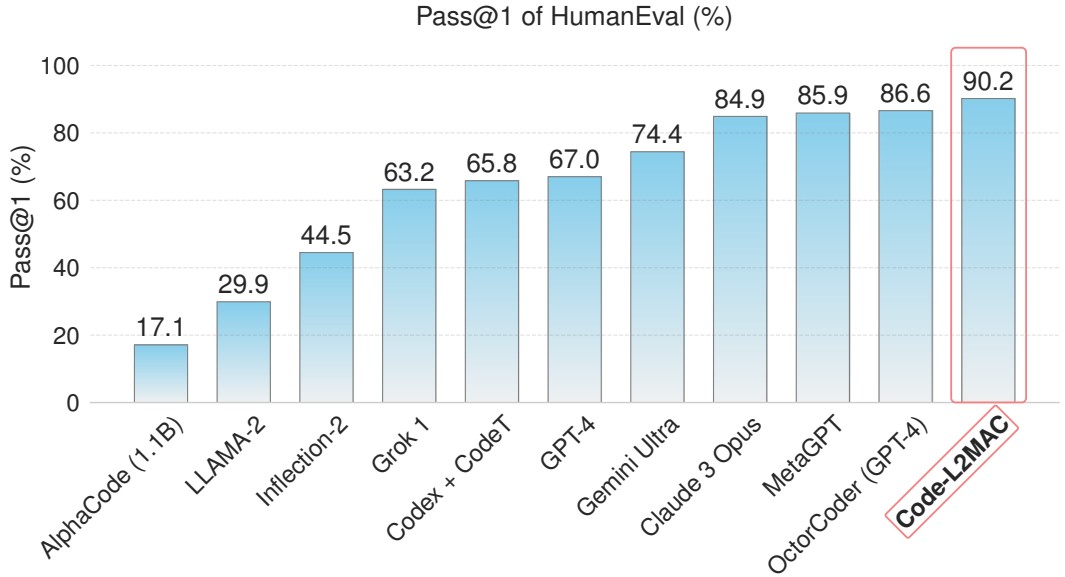

Figure 7: **HumanEval Benchmark**—Pass rates with a single attempt.

**HumanEval Benchmark.** As introduced by Chen et al. (2021), consists of 164 hand-written programming problems, where each task provides a docstring from which the LLM-based method

needs to generate the rest of the python function. Each task generation is then evaluated for functional correctness by testing with held-out unit tests, which they all have to pass for that task to be marked as complete.

**Evaluation Metric.** To evaluate the functional correctness of the programs generated the benchmark uses the pass@$k$ metric as presented by Chen et al. (2021), to evaluate the functional accuracy of the top-$k$ generated programs.

$$\text{pass@}k := \mathop{\mathbb{E}}_{\text{Problems}} \left[ 1 - \frac{\binom{n-c}{k}}{\binom{n}{k}} \right] \tag{1}$$

**Baselines.** As HumanEval is an established benchmark, we quote the existing baseline results from respective papers, from the current global leaderboard. Such methods include AlphaCode (Li et al., 2022), LLAMA-2 (Touvron et al., 2023), Inflection-2 (Inflection AI, 2023), Grok-1 (xai-org, 2024; Team et al., 2023), Codex + CodeT (Chen et al., 2022), GPT-4 (OpenAI, 2023), Gemini Ultra (Team et al., 2023), Claude 3 Opus (Anthropic, 2024), MetaGPT (Hong et al., 2024) and OctorCoder (Muennighoff et al., 2023).

We modify the prompt to include the function docstring and provide the instruction to only write code to complete the given function for each problem within the benchmark. We also instruct in the initial prompt that all tests should be comprehensive, interpretable, and cover edge cases. We also performed this evaluation using the GPT-4 model of 1106-Preview.

**Main Result.** Figure 7 shows that Code-L2MAC outperforms all baselines listed above on the HumanEval benchmark. Crucially, Code-L2MAC that uses GPT-4 within its multi-agent framework significantly outperforms a standalone GPT-4 model on the pass@1 metric on the HumanEval benchmark.

## H.2 L2MAC FOR WRITING AN ENTIRE BOOK

In the following we show that a general-purpose L2MAC works for completing general-purpose long output text-based tasks, such as writing an entire book. Specifically, we posed the task to write a complete book based on the following input prompt below of

```
Write a complete recipe book for the following book title of "Twirls & Tastes: A Journey Through Italian Pasta"

Description: "Twirls & Tastes" invites you on a flavorful expedition across Italy, exploring the diverse pasta landscape from the sun-drenched hills of Tuscany to the bustling streets of Naples. Discover regional specialties, learn the stories behind each dish, and master the art of pasta making with easy-to-follow recipes that promise to delight your senses.
```

Specifically, as we use the same L2MAC framework, as presented in Sections 3 and 4, where we now generate output text instead of code. Specifically, we replace the error checking mechanism $\mathcal{E}(D)$ from a code specific one as in Code-L2MAC to a general consistency check, for coherence of the generated output to the existing generated previous outputs. We additionally use the GPT-4 model of 1106-Preview.

The full result of the raw generated book is given here:

```
title_page_twirls_tastes.txt
```
Twirls & Tastes: A Journey Through Italian Pasta

by [Author's Name]
```

foreword_twirls_tastes.txt
```
Twirls & Tastes: A Journey Through Italian Pasta

Foreword

Dear Reader,

Welcome to a world where the simple act of twirling a fork can transport you to a place rich with history, culture, and flavor. Italian pasta is not just food; it's a narrative woven into the very fabric of Italy's heritage, a series of stories told through each shape, each sauce, and each bite.

As you turn these pages, you will embark on a journey that spans the snowy peaks of the North to the sun-kissed shores of the South. You will discover the artistry behind the humble noodle and the myriad ways it has been transformed by the hands of grandmothers, chefs, and artisans alike.

I invite you to savor the tales of pasta's past, to revel in the creativity of its present, and to anticipate the innovations that will shape its future. This book is a tribute to the traditions that have made Italian pasta a cornerstone of the culinary world and a celebration of the joy it brings to tables around the globe.
```

My own love affair with pasta began in my nonna's kitchen, where I learned that the secret ingredient to any dish is the love with which it's made. I hope to share a piece of that love with you through the recipes and stories contained within these pages.

Prepare to be delighted by the diversity of flavors and textures that await you. From the robust ragus of the North to the zesty seafood pastas of the South, your palate will be your passport to the regions of Italy.

Buon viaggio and buon appetito!

[Author's Name]
```

introduction_twirls_tastes.txt
```
Twirls & Tastes: A Journey Through Italian Pasta

Introduction

Italian pasta: a tapestry of history, culture, and culinary artistry. From the rolling hills of Tuscany to the bustling streets of Naples, pasta is the heartbeat of Italian cuisine. Its origins trace back to ancient times, and over the centuries, it has evolved into a myriad of shapes and forms, each with its own story to tell.

In this book, we delve into the rich tapestry of Italian pasta, exploring its storied past, the diversity of its forms, and its revered place in Italian culture. We will journey through the regions of Italy, discovering how geography and history have shaped the nation's beloved dish. Along the way, we will encounter traditional recipes passed down through generations, as well as innovative creations that continue to push the boundaries of flavor and form.

The structure of this book is designed to guide you through Italy's pasta landscape one region at a time. Each chapter focuses on a different area, providing historical context, cultural anecdotes, and a selection of recipes that highlight the region's unique contributions to pasta cuisine. From the hearty, butter-laden dishes of the North to the vibrant, olive oil-based recipes of the South, you will gain a comprehensive understanding of Italy's pasta tradition.

To use this book, simply begin with the introduction to set the stage for your culinary adventure. Then, proceed through the chapters at your own pace, savoring the stories and flavors of each region. The pasta-making techniques section will equip you with the skills to create your own pasta masterpieces, while the glossary will help you navigate the rich vocabulary of Italian pasta terms.

Prepare to embark on a flavorful journey through time and taste. 'Twirls & Tastes' is more than a cookbook; it's a passport to the soul of Italian cuisine. Buon viaggio e buon appetito!
```

chapter1_twirls_tastes.txt
```
Twirls & Tastes: A Journey Through Italian Pasta

Chapter 1: The Heart of Italy – Tuscany

Nestled in the heart of Italy, Tuscany is a region renowned for its breathtaking landscapes, rich history, and culinary excellence. The rolling hills, dotted with vineyards and olive groves, set the stage for a gastronomic adventure that is deeply intertwined with the land itself.

Pasta plays a pivotal role in Tuscan cuisine, embodying the simplicity and rustic elegance that the region is known for. One cannot think of Tuscany without envisioning the wide, flat ribbons of pappardelle, often served with hearty, game-based sauces such as the iconic Pappardelle al Cinghiale – a wild boar ragu that is as robust in flavor as the Tuscan countryside.

Recipes:

Pappardelle al Cinghiale

Ingredients:
– 400g pappardelle pasta
– 2 tablespoons extra-virgin olive oil
– 1 onion, finely chopped
– 2 carrots, finely chopped
– 2 celery stalks, finely chopped
– 2 garlic cloves, minced
– 500g wild boar meat, minced
– 1 cup red wine
– 400g canned tomatoes, crushed
– Salt and pepper to taste
– Grated Parmesan cheese, for serving

Instructions:
1. Heat the olive oil in a large pan over medium heat. Add the onion, carrots, celery, and garlic, and saute until softened.
2. Increase the heat to high, add the wild boar meat, and cook until browned.
3. Pour in the red wine and let it reduce by half.
4. Add the crushed tomatoes, salt, and pepper. Lower the heat, cover, and simmer for at least 2 hours, stirring occasionally.
5. Cook the pappardelle in a large pot of salted boiling water until al dente, then drain.
6. Toss the pasta with the wild boar ragu and serve with a sprinkle of Parmesan cheese.

This dish tells a story of the Tuscan hunt, tradition, and the deep connection between the land and the table. The robust flavors and hearty texture are emblematic of the region's cuisine, offering a taste of Tuscany's culinary heritage.

As you recreate these recipes, imagine the Tuscan landscape and the generations of cooks who have perfected these dishes over centuries. Each ingredient, each step, is a testament to the love and respect for the food that is at the heart of Tuscan life.

[Note: High-quality images of the Tuscan landscape and the completed dishes will be included in the final publication to enhance the reader's experience.]

Review for accuracy, clarity, and consistency with the book's theme has been conducted to ensure that this chapter aligns with the journey through Italian pasta that 'Twirls & Tastes' promises to deliver.
```

chapter2_twirls_tastes.txt
```
Twirls & Tastes: A Journey Through Italian Pasta

Chapter 2: The Riches of Emilia-Romagna

Emilia-Romagna, a region celebrated for its culinary contributions, is a treasure trove of Italian gastronomy. The fertile plains and the Po River create a lush environment for agriculture, giving rise to a variety of ingredients that form the backbone of its rich cuisine.

In Emilia-Romagna, pasta is not just food; it's a work of art. The region is the birthplace of many pasta shapes and sizes, each with its own story. Tagliatelle, tortellini, and lasagne are just a few examples of the pasta that have earned this region a worldwide reputation.

Recipes:

Tagliatelle al Ragu Bolognese

Ingredients:
– 400g tagliatelle pasta
– 2 tablespoons extra-virgin olive oil
– 1 onion, finely chopped
– 1 carrot, finely chopped
– 1 celery stalk, finely chopped
– 2 garlic cloves, minced
– 300g ground beef
– 150g ground pork
– 1 cup red wine
– 800g canned tomatoes, crushed
– Salt and pepper to taste
– Grated Parmigiano-Reggiano, for serving

Instructions:
1. Heat the olive oil in a large pan over medium heat. Add the onion, carrot, celery, and garlic, and saute until softened.
2. Increase the heat to high, add the ground beef and pork, and cook until browned.
3. Pour in the red wine and let it reduce by half.
4. Add the crushed tomatoes, salt, and pepper. Lower the heat, cover, and simmer for at least 2 hours, stirring occasionally.
5. Cook the tagliatelle in a large pot of salted boiling water until al dente, then drain.
6. Toss the pasta with the ragu and serve with a sprinkle of Parmigiano-Reggiano.

This classic dish is a testament to the culinary expertise of Emilia-Romagna, where the slow cooking of the ragu brings out the depth of flavors that this region is known for.

Erbazzone (Spinach and Cheese Pie)

Ingredients:
– 500g spinach, washed and chopped
– 200g ricotta cheese
– 100g Parmigiano-Reggiano, grated
– 2 eggs
– Salt and pepper to taste
– 2 sheets of puff pastry

Instructions:
1. Preheat the oven to 180 C (350 F).
2. In a bowl, mix the spinach, ricotta, Parmigiano-Reggiano, eggs, salt, and pepper.
3. Lay one sheet of puff pastry on a baking sheet. Spread the spinach mixture over the pastry, leaving a small border.
4. Cover with the second sheet of puff pastry, seal the edges, and make small cuts on the top to allow steam to escape.
5. Bake for 30–35 minutes, or until golden brown.

Erbazzone offers a glimpse into the rustic, yet refined flavors of Emilia-Romagna's countryside, where greens and cheeses are turned into comforting, savory pies.

[Note: High-quality images of the Emilia-Romagna landscape and the completed dishes will be included in the final publication to enhance the reader's experience.]

Review for accuracy, clarity, and consistency with the book's theme has been conducted to ensure that this chapter aligns with the journey through Italian pasta that 'Twirls & Tastes' promises to deliver.
```

chapter3_twirls_tastes.txt
```
Twirls & Tastes: A Journey Through Italian Pasta

Chapter 3: The Sun and Sea of Campania

Campania, with its sun-drenched coastlines and the vibrant city of Naples, is a region where the warmth of the sun and the freshness of the sea are reflected in its cuisine. The region's volcanic soil and mild climate contribute to the cultivation of exquisite produce, including the famous San Marzano tomatoes and the revered buffalo mozzarella.

Pasta in Campania is celebrated in its many forms, from the dry pasta of Gragnano to the hand-shaped delights like orecchiette. Seafood plays a significant role in the regional dishes, with recipes that have been passed down through generations.

Recipes:

Spaghetti alle Vongole

Ingredients:
– 400g spaghetti
– 4 tablespoons extra-virgin olive oil
– 4 garlic cloves, minced
– 1 small red chili, finely chopped
– 500g clams, cleaned
– 1/2 cup white wine
– A handful of fresh parsley, chopped
– Salt to taste

Instructions:
1. Heat the olive oil in a large pan over medium heat. Add the garlic and chili, and saute until fragrant.
2. Add the clams and white wine, cover, and cook until the clams open, about 5–7 minutes.
3. Cook the spaghetti in a large pot of salted boiling water until al dente, then drain, reserving some of the pasta water.
4. Add the spaghetti to the pan with the clams, adding a little pasta water if needed to loosen the sauce.
5. Stir in the parsley and serve immediately.

Spaghetti alle Vongole is a dish that captures the essence of Campania's coastline, with the briny flavors of the clams complementing the simplicity of the pasta.

Pizza Napoletana

Ingredients:
– 500g pizza dough
– 200g San Marzano tomatoes, crushed
– 200g buffalo mozzarella, sliced
– Fresh basil leaves
– Extra-virgin olive oil
– Salt to taste

Instructions:
1. Preheat the oven to its highest setting, ideally 250 C (480 F) or higher.
2. Roll out the pizza dough to form a thin base and place it on a pizza stone or baking sheet.
3. Spread the crushed tomatoes over the base, leaving a small border.

4. Distribute the mozzarella slices and basil leaves on top of the tomatoes.
5. Drizzle with olive oil and sprinkle with salt.
6. Bake for 8-10 minutes, or until the crust is golden and the cheese is bubbling.

Pizza Napoletana, with its soft, chewy crust and fresh toppings, is a beloved staple of Campanian cuisine and a symbol of Italian food culture around the world.

[Note: High-quality images of the Campania landscape and the completed dishes will be included in the final publication to enhance the reader's experience.]

Review for accuracy, clarity, and consistency with the book's theme has been conducted to ensure that this chapter aligns with the journey through Italian pasta that 'Twirls & Tastes' promises to deliver.
```

chapter4_twirls_tastes.txt
```
Twirls & Tastes: A Journey Through Italian Pasta

Chapter 4: The Island Flavors of Sicily

Sicily, the largest island in the Mediterranean, boasts a culinary landscape as diverse and vibrant as its history. Influences from Greek, Arab, and Norman conquerors have woven a rich tapestry of flavors that define Sicilian cuisine. The island's fertile soil and bountiful seas provide an abundance of fresh ingredients that inspire the local dishes.

Pasta in Sicily is often paired with the bold flavors of the sea and the sweetness of the island's produce. Dishes like Pasta alla Norma and Spaghetti ai Ricci di Mare showcase the creativity and resourcefulness of Sicilian cooking.

Recipes:

Pasta alla Norma

Ingredients:
- 400g short pasta, such as rigatoni or penne
- 3 tablespoons extra-virgin olive oil
- 1 eggplant, cut into cubes
- 2 garlic cloves, minced
- 400g canned tomatoes, crushed
- A handful of fresh basil, torn
- Salt and pepper to taste
- Ricotta salata, grated, for serving

Instructions:
1. Heat the olive oil in a large pan over medium heat. Add the eggplant cubes and fry until golden brown. Remove and set aside.
2. In the same pan, add the garlic and saute until fragrant.
3. Add the crushed tomatoes, basil, salt, and pepper. Simmer for 10 minutes.
4. Cook the pasta in a large pot of salted boiling water until al dente, then drain.
5. Toss the pasta with the tomato sauce and fried eggplant.
6. Serve with a generous sprinkling of ricotta salata.

Pasta alla Norma is a celebration of Sicily's agricultural bounty, with the creamy eggplant and tangy ricotta salata creating a harmonious blend of flavors.

Caponata

Ingredients:
- 1 large eggplant, cut into cubes
- 2 celery stalks, chopped
- 1 onion, chopped
- 3 tablespoons capers, rinsed
- A handful of green olives, pitted and chopped
- 2 tablespoons pine nuts
- 2 tablespoons sugar
- 3 tablespoons red wine vinegar
- 400g canned tomatoes, crushed
- Extra-virgin olive oil
- Salt and pepper to taste

Instructions:
1. Fry the eggplant cubes in olive oil until golden brown. Remove and set aside.
2. In the same pan, saute the celery and onion until softened.
3. Add the capers, olives, pine nuts, sugar, and vinegar, and cook for a few minutes.
4. Add the crushed tomatoes and fried eggplant, season with salt and pepper, and simmer for 15 minutes.

Caponata is a quintessential Sicilian dish, a sweet and sour vegetable medley that can be served as an appetizer or a side, embodying the island's complex culinary heritage.

[Note: High-quality images of the Sicilian landscape and the completed dishes will be included in the final publication to enhance the reader's experience.]

Review for accuracy, clarity, and consistency with the book's theme has been conducted to ensure that this chapter aligns with the journey through Italian pasta that 'Twirls & Tastes' promises to deliver.
```

chapter5_twirls_tastes.txt
```
Twirls & Tastes: A Journey Through Italian Pasta

Chapter 5: The Art of Pasta Making

The art of pasta making is a cherished tradition in Italy, passed down through generations. This chapter is dedicated to teaching you the time-honored techniques of creating pasta from scratch. With detailed instructions and accompanying images, you'll learn to craft various types of pasta dough, shape them into beautiful forms, and cook them to perfection.

Types of Pasta Dough:

1. Basic Egg Pasta Dough
Ingredients:
- 400g '00' flour
- 4 large eggs
Instructions:
- Mound the flour on a clean surface and create a well in the center.
- Crack the eggs into the well and gradually incorporate the flour from the inside rim of the well.
- Knead the dough for about 10 minutes until smooth and elastic.
- Wrap in plastic and let rest for 30 minutes at room temperature.

2. Semolina Pasta Dough
Ingredients:

– 400g semolina flour
– 200ml water
Instructions:
– Mix the semolina flour and water until a dough begins to form.
– Knead the dough on a clean surface until smooth and firm, about 10 minutes.
– Wrap in plastic and let rest for 30 minutes at room temperature.

Shaping Techniques:

1. Rolling and Cutting
– Roll out the rested dough using a rolling pin or pasta machine to the desired thickness.
– Cut into strips for tagliatelle, or squares for ravioli.

2. Hand-Shaping
– For shapes like orecchiette, pinch off small pieces of dough and shape with your fingers.
– For cavatelli, roll small pieces of dough against the surface of a gnocchi board or fork to create ridges.

Cooking Tips:

1. Boiling Pasta
– Use a large pot of salted boiling water for cooking pasta.
– Cook until al dente, usually 2-4 minutes for fresh pasta.
– Drain and toss with your favorite sauce.

2. Storing Pasta
– Fresh pasta can be dried on a rack and stored in an airtight container.
– It can also be frozen for up to a month.

[Note: High-quality images of the pasta-making process and the various shapes will be included in the final publication to guide the reader.]

This chapter provides a practical guide to making pasta at home. By following these steps, you'll be able to create a variety of pasta dishes with the authentic flavors and textures of Italy. Enjoy the satisfaction of making pasta from scratch and sharing your creations with family and friends.

Review for accuracy, clarity, and consistency with the book's theme has been conducted to ensure that this chapter aligns with the journey through Italian pasta that 'Twirls & Tastes' promises to deliver.
```

glossary_twirls_tastes.txt
```
Twirls & Tastes: A Journey Through Italian Pasta

Glossary of Italian Pasta Terms

This glossary is designed to help you understand the terminology used throughout this book. It includes definitions of pasta shapes, ingredients, and cooking techniques commonly found in Italian cuisine.

– Al dente: Pasta cooked until it is firm to the bite. The ideal texture for most pasta dishes.
– Bolognese: A meat-based sauce originating from Bologna, typically made with a mixture of beef and pork.
– Cinghiale: Wild boar, often used in hearty Tuscan pasta sauces.
– Farfalle: 'Butterflies' in Italian, this pasta shape resembles bow ties or butterfly wings.
– Fusilli: A corkscrew-shaped pasta that is excellent for holding onto sauces.
– Gnocchi: Small dumplings made from potatoes, flour, and sometimes ricotta cheese.
– Grana Padano: A hard, slow-ripened, semi-fat cheese from Italy, similar to Parmigiano-Reggiano.
– Lasagne: Wide, flat pasta sheets used in layering dishes with sauce and cheese.
– Orecchiette: 'Little ears' in Italian, this pasta shape is a small, ear-like round disc.
– Pappardelle: Broad, flat pasta noodles, often served with rich, meaty sauces.
– Parmigiano-Reggiano: A hard, granular cheese known as Parmesan, used grated over pasta dishes.
– Pecorino: A hard Italian cheese made from sheep's milk.
– Penne: Tube-shaped pasta with angled ends, often used in pasta salads or baked dishes.
– Ragu: A meat-based sauce, which is simmered with tomatoes, onions, and other seasonings.
– Ricotta: A creamy Italian whey cheese used in various pasta fillings and sauces.
– San Marzano tomatoes: A variety of plum tomatoes considered by many chefs to be the best for making tomato sauces.
– Spaghetti: Long, thin, cylindrical pasta of Italian origin.
– Tagliatelle: Long, ribbon-like pasta similar to fettuccine, traditionally served with Bolognese sauce.
– Tortellini: Ring-shaped stuffed pasta, typically filled with meat, cheese, or spinach.

This glossary will assist you in better understanding the recipes and the cultural significance behind each dish. As you explore the diverse world of Italian pasta, refer to these terms to enhance your culinary knowledge and appreciation.
```

index_twirls_tastes.txt
```
Twirls & Tastes: A Journey Through Italian Pasta

Index

```
– Mozzarella , Chapter 3
– Orecchiette , Glossary
– Pappardelle , Glossary
– Parmesan cheese , Chapter 1
– Parmigiano−Reggiano , Chapter 2, Glossary
– Pecorino , Glossary
– Penne , Glossary
– Rag\u00f9 , Glossary
– Red wine , Chapters 1, 2
– Ricotta , Chapter 2, Glossary
– Ricotta salata , Chapter 4
– San Marzano tomatoes , Chapter 3, Glossary
– Semolina flour , Chapter 5
– Spaghetti , Chapter 3, Glossary
– Tagliatelle , Chapter 2, Glossary
– Tomatoes , Chapters 1, 2, 4
– Tortellini , Glossary
– Water , Chapter 5
– Wild boar meat , Chapter 1

Key Terms :
– Al dente , Glossary
– Bolognese , Glossary
– Cinghiale , Glossary
– Farfalle , Glossary
– Fusilli , Glossary
– Gnocchi , Glossary
– Grana Padano , Glossary
– Lasagne , Glossary
– Orecchiette , Glossary
– Pappardelle , Glossary
– Parmigiano−Reggiano , Glossary
– Pecorino , Glossary
– Penne , Glossary
– Rag\u00f9 , Glossary
– Ricotta , Glossary
– San Marzano tomatoes , Glossary
– Spaghetti , Glossary
– Tagliatelle , Glossary
– Tortellini , Glossary

Note : Page numbers will be added once the final layout of the book is complete .
```
```

## For this implementation of L2MAC we modified the system prompt to be

```
Objective : Write large amounts of text for the following task .
Please note that the generated text should be fully complete . No placeholders .
Only use the functions you have been provided with .
Only use the 'write_files ' to output text files .

You must act autonomously and you will receive no human input at any stage . You have to return as output the complete text for
        completing this task , and correctly incorporate it into the existing generated text files .
You always write out the whole file contents .
Please always view the files before writing to them , to make sure you are writing to the correct files .

Provide the text to achieve the task conditioned on the existing generated text−−−including changing the existing generated text
        where necessary .

You cannot visualize any graphical output . You exist within a Actor Model machine , and when you list out steps , each step will be
        taken by a new separate sub−ChatGPT model . When you list out a sub−task steps , you can optionally specify the sub−task
        validation to check that it has been completed successfully .

No data saved to disk will persist between steps or write operations .

Use the functions provided . When calling functions only provide a RFC8259 compliant JSON request following this format without
        deviation .
```

## And the initial prompt to be

```
You will get instructions for a book to write .
First lay out the the structure outline of the book , and the chapters with detailed descriptions of what each chapter will contain .
        Feel free to make a quick comment on the purpose of each chapter .
Do not comment on what every piece of text does . Please note that the text should be fully complete . No placeholders .

You will start with the "detailed_outline" file , then go to the chapters in chronological order , and so on .
Please note that the text should be fully complete . No placeholders .

Follow a the best practices for writing a book , and naming convention .
Make sure that files are correctly conditioned on the subsequent chapters and outline (s) . The text should be fully complete . Make
        sure that text in different files are compatible with each other .
When writing text if you are unsure , write the most plausible text .

Useful to know :

It is helpful to write a detailed outline of the book first , and then write the chapters in order .
Always add a comment briefly describing the purpose of each file .
Always follow the best practices for the requested structure and how to package the completed book .

Objective : '''
{task_description}
'''

Understand the problem , by creating an extremely detailed step−by−step plan , where each step is long (multiple sentences) and in
        total includes every single feature requirement specified above , feel free to copy directly from it . Use no more than 10
        steps in the plan . Perform additional , checks and evaluation at each step when applicable to help make an excellent coherent
        book , where all the text is fully complete . Use best book design practices , and you can output large amounts of text at once .
         Please include a last sentence to perform checks when implementing or writing text in that same step . You will receive no
        human input at any stage , so you cannot use a human to perform any checks . Only create a detailed plan to begin with , which
        includes performing consistency checks . Please be sure to include all of the specified feature requirements in the following
        plan .
```

We further modified the control unit prompts to replace instructions of generating and checking code to generating and checking text.

## H.3 ADDITIONAL MAIN RESULTS

In the following, we provide a more detailed main results table, including additional test metrics to show the number of failed and tests that passed, tabulated in Table 3. We also provide a codebase output example from each of the methods and observe that they illustrate the difference in quality between the respective approaches. Notably all methods were given the exact same user-specified feature requirements for the given long code generation task, Appendix E.

Table 3: Codebase generation system design task results showing the percentage of functional features specified that are fully implemented (**Features %**), the number of syntactical errors in the generated code (**# Errors**), the number of lines of code (**LOC**), number of failing tests (**Tests Failed**), and number of passing tests (**Tests Passed**). Code-L2MAC fully implements the highest percentage of user-specified feature requirements across all tasks by generating fully functional code that has minimal syntactical errors and a high number of passing self-generated unit tests. The results are averaged over 10 random seeds, with ± indicating 95% confidence intervals.

| Method | URL Shortener App | | | | | Online Microblogging App | | | | | Online Chat App | | | | |
|---|---|---|---|---|---|---|---|---|---|---|---|---|---|---|---|
| | Features % ↑ | # Errors ↓ | LOC ↓ | Tests Failed ↓ | Tests Passed ↑ | Features % ↑ | # Errors ↓ | LOC ↓ | Tests Failed ↓ | Tests Passed ↑ | Features % ↑ | # Errors ↓ | LOC ↓ | Tests Failed ↓ | Tests Passed ↑ |
| GPT4 | 48.2±15.2 | 0±0 | 107±32.7 | 2.3±1.01 | 0.6±0.5 | 17.9±8.27 | 3.75±3.09 | 106±35.4 | 0.75±0.723 | 1.25±0.613 | 11±2.26 | 0.3±0.346 | 127±24.1 | 1.2±1 | 0.7±0.346 |
| AutoGPT | 25.3±19.6 | 0±0 | 136±41.9 | 3.3±1.91 | 0±0 | 33.3±18 | 0.6±0.369 | 148±35.5 | 3±2.86 | 0±0 | 23.1±11.8 | 1.85±2.47 | 220±65.8 | 3.08±3.34 | 0.538±1.01 |
| Code-L2MAC | **91.6±8.22** | **0±0** | **330±47.6** | **0±0** | **14±6.71** | **82.4±14.6** | **0±0** | **395±52.9** | **0.8±1.38** | **18.3±6.8** | **59.4±25.9** | **0±0** | **374±123** | **0±0** | **18.8±9.11** |

## L2MAC URL Shortener App example output

```
/
├── app.py
├── config.py
├── requirements.txt
├── services/
│   ├── admin_dashboard.py
│   ├── __init__.py
│   ├── url_shortener.py
│   └── user_account.py
└── tests/
    ├── test_admin_dashboard.py
    ├── test_app.py
    ├── test_url_shortener.py
    └── test_user_account.py
```

```python
# app.py
from flask import Flask, redirect, url_for, request
from services.url_shortener import UrlShortener
from services.user_account import UserAccount
from services.admin_dashboard import AdminDashboard

app = Flask(__name__)
url_shortener = UrlShortener()
user_account = UserAccount()
admin_dashboard = AdminDashboard()

@app.route('/')
def home():
        return 'Hello, World!'

@app.route('/<short_url>')
def redirect_to_url(short_url):
        url = url_shortener.get_original_url(short_url)
        if url is not None:
                url_shortener.record_click(short_url, 'mocked_location')
                return redirect(url)
        else:
                return 'URL_not_found', 404

@app.route('/analytics/<short_url>')
def get_analytics(short_url):
        analytics = url_shortener.get_analytics(short_url)
        if analytics is not None:
                return analytics
        else:
                return 'Analytics_not_found', 404

@app.route('/account/create', methods=['POST'])
def create_account():
```

```python
        username = request.form.get('username')
        return user_account.create_account(username)

@app.route('/account/urls', methods=['GET'])
def view_urls():
        username = request.args.get('username')
        return user_account.view_urls(username)

@app.route('/account/edit_url', methods=['PUT'])
def edit_url():
        username = request.form.get('username')
        old_url = request.form.get('old_url')
        new_url = request.form.get('new_url')
        return user_account.edit_url(username, old_url, new_url)

@app.route('/account/delete_url', methods=['DELETE'])
def delete_url():
        username = request.form.get('username')
        url = request.form.get('url')
        return user_account.delete_url(username, url)

@app.route('/account/analytics', methods=['GET'])
def view_analytics():
        username = request.args.get('username')
        return user_account.view_analytics(username)

@app.route('/admin/urls', methods=['GET'])
def view_all_urls():
        return admin_dashboard.view_all_urls()

@app.route('/admin/delete_url', methods=['DELETE'])
def admin_delete_url():
        url = request.form.get('url')
        return admin_dashboard.delete_url(url)

@app.route('/admin/delete_user', methods=['DELETE'])
def delete_user():
        user = request.form.get('user')
        return admin_dashboard.delete_user(user)

@app.route('/admin/monitor', methods=['GET'])
def monitor_system():
        return admin_dashboard.monitor_system()

if __name__ == '__main__':
        app.run(debug=True)

# config.py
import os

class Config(object):
        DEBUG = False
        TESTING = False
        SECRET_KEY = os.urandom(24)

# requirements.txt
flask==1.1.2
pytest==6.2.4

# services/admin_dashboard.py
class AdminDashboard:
        def __init__(self):
                self.users = {}
                self.urls = {}

        def view_all_urls(self):
                return self.urls

        def delete_url(self, url):
                if url in self.urls:
                        del self.urls[url]
                return self.urls

        def delete_user(self, user):
                if user in self.users:
                        del self.users[user]
                return self.users

        def monitor_system(self):
                return {'users': len(self.users), 'urls': len(self.urls)}

# services/__init__.py

# services/url_shortener.py
import string
import random
from datetime import datetime

class UrlShortener:
        def __init__(self):
                self.url_dict = {}
                self.analytics_dict = {}

        def validate_url(self, url):
                if url.startswith('http://') or url.startswith('https://'):
                        return True
                return False

        def generate_short_url(self, url, expiration_date=None, custom_short_url=None):
                if custom_short_url and custom_short_url in self.url_dict:
                        return 'Error:_This_custom_short_URL_is_not_available'
                short_url = custom_short_url or ''.join(random.choice(string.ascii_letters + string.digits) for _ in range(5))
                while short_url in self.url_dict:
                        short_url = ''.join(random.choice(string.ascii_letters + string.digits) for _ in range(5))
                self.url_dict[short_url] = {'url': url, 'expiration_date': expiration_date}
                self.analytics_dict[short_url] = {'clicks': 0, 'click_details': []}
```

```python
            return short_url

    def get_original_url(self, short_url):
        url_data = self.url_dict.get(short_url, None)
        if url_data and url_data['expiration_date'] and datetime.now() > url_data['expiration_date']:
            return 'Error: This URL has expired'
        return url_data['url'] if url_data else None

    def record_click(self, short_url, location):
        if short_url in self.analytics_dict:
            self.analytics_dict[short_url]['clicks'] += 1
            self.analytics_dict[short_url]['click_details'].append({'timestamp': datetime.now().isoformat(), 'location':
                location})

    def get_analytics(self, short_url):
        return self.analytics_dict.get(short_url, None)

# services/user_account.py
class UserAccount:
    def __init__(self):
        self.users = {}

    def create_account(self, username):
        if username in self.users:
            return 'Username already exists.'
        self.users[username] = {'urls': []}
        return 'Account created successfully.'

    def view_urls(self, username):
        if username not in self.users:
            return 'Username does not exist.'
        return self.users[username]['urls']

    def edit_url(self, username, old_url, new_url):
        if username not in self.users:
            return 'Username does not exist.'
        if old_url not in self.users[username]['urls']:
            return 'URL does not exist.'
        self.users[username]['urls'].remove(old_url)
        self.users[username]['urls'].append(new_url)
        return 'URL edited successfully.'

    def delete_url(self, username, url):
        if username not in self.users:
            return 'Username does not exist.'
        if url not in self.users[username]['urls']:
            return 'URL does not exist.'
        self.users[username]['urls'].remove(url)
        return 'URL deleted successfully.'

    def view_analytics(self, username):
        if username not in self.users:
            return 'Username does not exist.'
        return {url['url']: url['analytics'] for url in self.users[username]['urls'] if 'analytics' in url}

# tests/test_admin_dashboard.py
from services.admin_dashboard import AdminDashboard

def test_view_all_urls():
    admin = AdminDashboard()
    assert admin.view_all_urls() == {}

def test_delete_url():
    admin = AdminDashboard()
    admin.urls = {'test_url': 'test'}
    admin.delete_url('test_url')
    assert admin.urls == {}

def test_delete_user():
    admin = AdminDashboard()
    admin.users = {'test_user': 'test'}
    admin.delete_user('test_user')
    assert admin.users == {}

def test_monitor_system():
    admin = AdminDashboard()
    admin.users = {'test_user': 'test'}
    admin.urls = {'test_url': 'test'}
    assert admin.monitor_system() == {'users': 1, 'urls': 1}

# tests/test_app.py
import pytest
from flask import Flask
from app import app as flask_app

@pytest.fixture
def app():
    return flask_app

@pytest.fixture
def client(app):
    return app.test_client()

def test_home(client):
    response = client.get('/')
    assert response.status_code == 200

def test_redirect_to_url(client):
    response = client.get('/random_url')
    assert response.status_code == 404
```

```python
def test_get_analytics(client):
    response = client.get('/analytics/random_url')
    assert response.status_code == 404

def test_create_account(client):
    response = client.post('/account/create', data={'username': 'test_user'})
    assert response.status_code == 200

def test_view_urls(client):
    response = client.get('/account/urls', query_string={'username': 'test_user'})
    assert response.status_code == 200

def test_edit_url(client):
    response = client.put('/account/edit_url', data={'username': 'test_user', 'old_url': 'old_url', 'new_url': 'new_url'})
    assert response.status_code == 200

def test_delete_url(client):
    response = client.delete('/account/delete_url', data={'username': 'test_user', 'url': 'url'})
    assert response.status_code == 200

def test_view_analytics(client):
    response = client.get('/account/analytics', query_string={'username': 'test_user'})
    assert response.status_code == 200

def test_view_all_urls(client):
    response = client.get('/admin/urls')
    assert response.status_code == 200

def test_admin_delete_url(client):
    response = client.delete('/admin/delete_url', data={'url': 'url'})
    assert response.status_code == 200

def test_delete_user(client):
    response = client.delete('/admin/delete_user', data={'user': 'user'})
    assert response.status_code == 200

def test_monitor_system(client):
    response = client.get('/admin/monitor')
    assert response.status_code == 200

# tests/test_url_shortener.py
import pytest
from services.url_shortener import UrlShortener
from datetime import datetime, timedelta

def test_url_shortener():
    url_shortener = UrlShortener()
    url = 'http://example.com'
    short_url = url_shortener.generate_short_url(url)
    assert url_shortener.get_original_url(short_url) == url
    assert url_shortener.get_original_url('invalid') is None

    # Test custom short URL
    custom_short_url = 'custom'
    assert url_shortener.generate_short_url(url, custom_short_url=custom_short_url) == custom_short_url
    assert url_shortener.generate_short_url(url, custom_short_url=custom_short_url) == 'Error: This custom short URL is not
        available'

    # Test analytics
    assert url_shortener.get_analytics(short_url) == {'clicks': 0, 'click_details': []}
    url_shortener.record_click(short_url, 'mocked_location')
    analytics = url_shortener.get_analytics(short_url)
    assert analytics['clicks'] == 1
    assert len(analytics['click_details']) == 1
    assert analytics['click_details'][0]['location'] == 'mocked_location'
    assert 'timestamp' in analytics['click_details'][0]

    # Test URL expiration
    expired_url = url_shortener.generate_short_url(url, datetime.now() - timedelta(days=1))
    assert url_shortener.get_original_url(expired_url) == 'Error: This URL has expired'

# tests/test_user_account.py
from services.user_account import UserAccount

def test_create_account():
    user_account = UserAccount()
    assert user_account.create_account('test_user') == 'Account created successfully.'
    assert user_account.create_account('test_user') == 'Username already exists.'

def test_view_urls():
    user_account = UserAccount()
    user_account.create_account('test_user')
    assert user_account.view_urls('test_user') == []
    assert user_account.view_urls('non_existent_user') == 'Username does not exist.'

def test_edit_url():
    user_account = UserAccount()
    user_account.create_account('test_user')
    user_account.users['test_user']['urls'].append('http://test.com')
    assert user_account.edit_url('test_user', 'http://test.com', 'http://newtest.com') == 'URL edited successfully.'
    assert user_account.edit_url('test_user', 'http://nonexistent.com', 'http://newtest.com') == 'URL does not exist.'
    assert user_account.edit_url('non_existent_user', 'http://test.com', 'http://newtest.com') == 'Username does not exist.'
```

```
def test_delete_url():
    user_account = UserAccount()
    user_account.create_account('test_user')
    user_account.users['test_user']['urls'].append('http://test.com')
    assert user_account.delete_url('test_user', 'http://test.com') == 'URL deleted successfully.'
    assert user_account.delete_url('test_user', 'http://nonexistent.com') == 'URL does not exist.'
    assert user_account.delete_url('non_existent_user', 'http://test.com') == 'Username does not exist.'

def test_view_analytics():
    user_account = UserAccount()
    user_account.create_account('test_user')
    user_account.users['test_user']['urls'].append({'url': 'http://test.com', 'analytics': {'clicks': 10, 'locations': ['USA', 'UK']}})
    assert user_account.view_analytics('test_user') == {'http://test.com': {'clicks': 10, 'locations': ['USA', 'UK']}}
    assert user_account.view_analytics('non_existent_user') == 'Username does not exist.'
```

## AutoGPT URL Shortener App example output

```
/
├── analytics.py
├── main.py
└── url_shortener.py
```

```python
# analytics.py
class Analytics:
    def track_clicks(self, short_url):
        pass

    def track_geolocation(self, short_url):
        pass
```

```python
# main.py
from url_shortener import URLShortener

if __name__ == '__main__':
    url_shortener = URLShortener()
    # Test the implemented functionality
```

```python
# url_shortener.py
import random
import string
from analytics import Analytics

class URLShortener:
    def __init__(self):
        self.analytics = Analytics()

    def validate_url(self, url):
        # Add URL validation logic
        pass

    def generate_shortened_url(self):
        # Generate a random 6-character string
        short_url = ''.join(random.choices(string.ascii_letters + string.digits, k=6))
        return short_url

    def shorten_url(self, url):
        if self.validate_url(url):
            short_url = self.generate_shortened_url()
            # Save the short_url and original_url mapping
            return short_url
        else:
            raise ValueError('Invalid URL')

    def redirect_to_original_url(self, short_url):
        # Retrieve the original_url from the short_url
        # Track clicks and geolocation
        self.analytics.track_clicks(short_url)
        self.analytics.track_geolocation(short_url)
        pass

class User:
    def create_account(self, username, password):
        pass

    def view_shortened_urls(self):
        pass

    def edit_url(self, short_url, new_url):
        pass

    def delete_url(self, short_url):
        pass

    def view_analytics(self, short_url):
        pass

    def set_expiration_date(self, short_url, expiration_date):
        pass

class Admin:
    def view_all_urls(self):
        pass

    def delete_url_or_user(self, identifier):
        pass

    def monitor_performance(self):
```

```
        pass
```

## AutoGPT URL Shortener App example output

```
/
├── app.py
└── test_app.py
```

```python
# app.py
from flask import Flask, request, jsonify, redirect
from dataclasses import dataclass
from datetime import datetime
import pytz
import uuid

app = Flask(__name__)

# Mock database
DB = {}

@dataclass
class URL:
        original: str
        shortened: str
        user: str
        clicks: int
        created_at: datetime
        expires_at: datetime

@app.route('/shorten', methods=['POST'])
def shorten_url():
        data = request.get_json()
        original_url = data.get('url')
        user = data.get('user')
        expires_at = data.get('expires_at')
        short_url = str(uuid.uuid4())[:8]
        url = URL(original=original_url, shortened=short_url, user=user, clicks=0, created_at=datetime.now(pytz.utc), expires_at=
                expires_at)
        DB[short_url] = url
        return jsonify({'short_url': short_url}), 201

@app.route('/<short_url>', methods=['GET'])
def redirect_url(short_url):
        url = DB.get(short_url)
        if url and url.expires_at > datetime.now(pytz.utc):
                url.clicks += 1
                return redirect(url.original, code=302)
        else:
                return jsonify({'error': 'URL_not_found_or_expired'}), 404

if __name__ == '__main__':
        app.run(debug=True)

# test_app.py
import pytest
import app
from flask import json

@pytest.fixture
def client():
        app.app.config['TESTING'] = True
        with app.app.test_client() as client:
                yield client

def test_shorten_url(client):
        response = client.post('/shorten', data=json.dumps({'url': 'https://www.google.com', 'user': 'test', 'expires_at': '
                2022-12-31T23:59:59Z'}), content_type='application/json')
        assert response.status_code == 201
        assert 'short_url' in response.get_json()

def test_redirect_url(client):
        response = client.post('/shorten', data=json.dumps({'url': 'https://www.google.com', 'user': 'test', 'expires_at': '
                2022-12-31T23:59:59Z'}), content_type='application/json')
        short_url = response.get_json().get('short_url')
        response = client.get(f'/{short_url}')
        assert response.status_code == 302

def test_redirect_url_not_found(client):
        response = client.get('/notfound')
        assert response.status_code == 404
        assert 'error' in response.get_json()
```

### H.4  CODE L2MAC IN ACTION.

We provide example transcripts of L2MAC in action below. For the environment task of the Online Chat App.

Initial instructions planning.

```
[System]
Objective: Write code for a large system design task.
Please note that the code should be fully functional. No placeholders.
Only use the functions you have been provided with.
Only use the 'write_files' to output code.
```

You must act autonomously and you will receive no human input at any stage. You have to return as output the complete code for completing this task, and correctly incorporate it into the existing codebase.
You always write out the whole file contents. You always indent code with tabs.
Please always view the files before writing to them, to make sure you are writing to the correct files.
When writing a test, make the filename start with the prefix 'test_'.

Provide the minimal code necessary to achieve the task conditioned on the existing generated code—including changing the existing generated code.

You cannot visualize any graphical output. You exist within a Actor Model machine, and when you list out steps, each step will be taken by a new separate sub-ChatGPT model. When you list out a sub-task steps, you can optionally specify the sub-task validation to check that it has been completed successfully.

You cannot use any databases as none are setup in the local environment, instead mock a database with an in memory dictionary to store data. No data saved to disk will persist between steps or write operations.

If a test is failing the error could be the code, or the test is incorrect, so feel free to overwrite and change the tests when they are incorrect, to make all tests pass.

Use the functions provided. When calling functions only provide a RFC8259 compliant JSON request following this format without deviation.

[CU]
You will get instructions for code to write.
First lay out the names of the core classes, functions, methods that will be necessary, As well as a quick comment on their purpose.
Do not comment on what every file does. Please note that the code should be fully functional. No placeholders.

You will start with the "entrypoint" file, then go to the ones that are imported by that file, and so on.
Please note that the code should be fully functional. No placeholders.

Follow a language and framework appropriate best practice file naming convention.
Make sure that files contain all imports, types etc. The code should be fully functional. Make sure that code in different files are compatible with each other.
When writing code if you are unsure, write a plausible implementation.
Include module dependency or package manager dependency definition file.

Useful to know:

For Python, you always create an appropriate requirements.txt file.
Always add a comment briefly describing the purpose of the function definition.
Add comments explaining very complex bits of logic.
Always follow the best practices for the requested languages for folder/file structure and how to package the project.
You can use any package and any other packages you wish to install.
You cannot use any databases as none are setup in the local environment, instead mock a database with an in memory dictionary to store data. No data saved to disk will persis between steps or write operations.
When writing a test, make the filename start with the prefix 'test_'.

Python toolbelt preferences:
- pytest
- dataclasses
- flask

Objective:```
**Global Chat Service (GCS)**

**Overview**:
A real-time online chat application allowing users to send text messages, images, and create group chats.

**Functional Requirements to implement**:

User Registration and Authentication:
- [ ] 1.1. Sign up using email.
- [ ] 1.2. Forgotten password recovery.

User Profile:
- [ ] 2.1. Allow users to set profile pictures and status messages.
- [ ] 2.2. Privacy settings for who can see user details or last seen status.

Contact Management:
- [ ] 3.1. Block/unblock contacts.
- [ ] 3.2. Create, edit, and manage groups.

Messaging:
- [ ] 4.1. Send and receive real-time text messages.
- [ ] 4.2. Message read receipts (blue ticks or equivalent).
- [ ] 4.3. End-to-end encryption for security.
- [ ] 4.4. Image sharing.
- [ ] 4.5. Emojis, GIFs, and stickers support.

Group Chats:
- [ ] 5.1. Create group chats with a name and picture.
- [ ] 5.2. Add or remove participants.
- [ ] 5.3. Admin roles and permissions.

Status/Story Feature:
- [ ] 6.1. Allow users to post image statuses visible for a limited time.
- [ ] 6.2. Control who can see the status.

Web Application:
- [ ] 7.1. Web-based version accessible from browsers.

Connectivity and Offline Mode:
- [ ] 8.1. Message queuing for when the user is offline; messages are sent once connectivity is restored.
- [ ] 8.2. Display online/offline status.
```

Understand the problem, by creating an extremely detailed step-by-step plan, where each step is long (multiple sentences) and in total includes every single feature requirement specified above, feel free to copy directly from it. Use no more than 10 steps in the plan. Create additional tests, checks and evaluation at each step when applicable to help make an excellent code implementation, where all the code is fully functional. Use best software design practices, and you can output large amounts of code at once. Please include a last sentence to create and run tests when implementing or writing code in that same step. You will receive no human input at any stage, so you cannot use a human to test. Only create a detailed plan to begin with, which includes designing and running tests to check that they all pass. Please be sure to include all of the specified feature requirements in the following plan.

[LLM RESPONSE MESSAGE]

```
[Function Call][provide_detailed_sub_task_steps_for_sub_agents] {
  "steps": [
    "Step 1: Start by setting up the basic structure of the application. This includes creating the main application file (app.py)
     and setting up a Flask application. Also, create a mock database using an in-memory dictionary to store user data. This
     dictionary should be able to store user information such as email, password, profile picture, status message, privacy
     settings, blocked contacts, and groups. Write tests to ensure that the application setup is correct and the mock database
     can store and retrieve data correctly.",

    "Step 2: Implement user registration and authentication. This includes creating routes for user sign up and forgotten password
     recovery. The sign up route should accept user email and password, and store this information in the mock database. The
     forgotten password recovery route should accept user email and send a password reset link (for the purpose of this task,
     just display the link). Write tests to ensure that the registration and authentication routes work as expected.",

    "Step 3: Implement user profile management. This includes creating routes for setting profile pictures and status messages, and
     updating privacy settings. The profile picture and status message routes should accept the respective data and update the
     user's information in the mock database. The privacy settings route should accept the new settings and update the user's
     information in the mock database. Write tests to ensure that the profile management routes work as expected.",

    "Step 4: Implement contact management. This includes creating routes for blocking/unblocking contacts and managing groups. The
     block/unblock routes should accept the contact's email and update the user's blocked contacts list in the mock database.
     The group management routes should accept the group details (name, picture, participants) and update the user's groups in
     the mock database. Write tests to ensure that the contact management routes work as expected.",

    "Step 5: Implement messaging. This includes creating routes for sending and receiving messages, updating read receipts,
     encrypting messages, and sharing images. The send/receive routes should accept the message data and update the respective
     user's messages in the mock database. The read receipt route should accept the message ID and update the message's read
     status in the mock database. The encryption route should accept the message data, encrypt it, and store it in the mock
     database. The image sharing route should accept the image data and update the respective user's messages in the mock
     database. Write tests to ensure that the messaging routes work as expected.",

    "Step 6: Implement group chats. This includes creating routes for creating group chats, adding/removing participants, and
     managing admin roles and permissions. The create group chat route should accept the group details (name, picture,
     participants) and update the user's groups in the mock database. The add/remove participants route should accept the
     group ID and participant's email, and update the group's participants in the mock database. The admin roles and
     permissions route should accept the group ID, admin's email, and new permissions, and update the group's admin
     permissions in the mock database. Write tests to ensure that the group chat routes work as expected.",

    "Step 7: Implement the status/story feature. This includes creating routes for posting image statuses and managing who can see
     the status. The post status route should accept the image data and update the user's statuses in the mock database. The
     manage visibility route should accept the status ID and visibility settings, and update the status's visibility in the
     mock database. Write tests to ensure that the status/story routes work as expected.",

    "Step 8: Implement the web application. This includes creating a basic HTML/CSS/JS frontend that interacts with the Flask
     backend. The frontend should have pages for sign up, sign in, profile management, contact management, messaging, group
     chats, and status/story. Write tests to ensure that the frontend can successfully send requests to the backend and
     display the returned data.",

    "Step 9: Implement connectivity and offline mode. This includes creating a route for updating the user's online/offline status,
     and a system for queuing messages when the user is offline. The online/offline status route should accept the user's
     email and new status, and update the user's status in the mock database. The message queuing system should store messages
     in a queue when the user is offline, and send them when the user comes online. Write tests to ensure that the
     connectivity and offline mode features work as expected.",

    "Step 10: Finally, perform a comprehensive test of the entire application. This includes testing all routes, the frontend, and
     the mock database. Fix any bugs that are found, and ensure that all tests pass. Also, ensure that the application meets
     all the specified feature requirements."
  ]
}
```

Then the first instruction is loaded and is executed, for example,

```
[System]
Objective: Write code for a large system design task.
Please note that the code should be fully functional. No placeholders.
Only use the functions you have been provided with.
Only use the 'write_files' to output code.

You must act autonomously and you will receive no human input at any stage. You have to return as output the complete code for
    completing this task, and correctly incorporate it into the existing codebase.
You always write out the whole file contents. You always indent code with tabs.
Please always view the files before writing to them, to make sure you are writing to the correct files.
When writing a test, make the filename start with the prefix 'test_'.

Provide the minimal code necessary to achieve the task conditioned on the existing generated code---including changing the existing
    generated code.

You cannot visualize any graphical output. You exist within a Actor Model machine, and when you list out steps, each step will be
    taken by a new separate sub-ChatGPT model. When you list out a sub-task steps, you can optionally specify the sub-task
    validation to check that it has been completed successfully.

You cannot use any databases as none are setup in the local environment, instead mock a database with an in memory dictionary to
    store data. No data saved to disk will persist between steps or write operations.

If a test is failing the error could be the code, or the test is incorrect, so feel free to overwrite and change the tests when they
    are incorrect, to make all tests pass.

Use the functions provided. When calling functions only provide a RFC8259 compliant JSON request following this format without
    deviation.

[CU]
Objective: Execute sub task step: Step 1: Start by setting up the basic structure of the application. This includes creating the
    main application file (app.py) and setting up a Flask application. Also, create a mock database using an in-memory dictionary
    to store user data. This dictionary should be able to store user information such as email, password, profile picture,
    status message, privacy settings, blocked contacts, and groups. Write tests to ensure that the application setup is correct
    and the mock database can store and retrieve data correctly..

 Note: Condition any new code files on the existing code files: []. Fully implement these features in the code, no placeholders. You
    can now optionally view the existing files if you need to view them to complete the current task step. You have a limited
    context window so be selective about which files you view, only view the files you think you might need to view.

Summary output of previous step: """

Respond now only with a function call of one of the following functions provided: 'sub_task_step_complete', 'view_files', '
    run_python_file', 'pytest_files', 'write_files', 'delete_files', and if you want to output code only use the 'write_files'
    function to output code.
```

```
[LLM] [Function Call] [write_files] {
  "files_and_contents": [
    {
      "file_path": "app.py",
      "file_contents": "from flask import Flask, request\n\napp = Flask(__name__)\n\n# Mock database\nusers_db = {}\n\n@app.route('/
          users', methods=['POST'])\ndef create_user():\n\tdata = request.get_json()\n\tusers_db[data['email']] = data\n\treturn
          {'message': 'User created'}, 201\n\n@app.route('/users/<email>', methods=['GET'])\ndef get_user(email):\n\tuser =
          users_db.get(email)\n\tif not user:\n\t\treturn {'message': 'User not found'}, 404\n\treturn user, 200"
    },
    {
      "file_path": "test_app.py",
      "file_contents": "import pytest\nimport app\n\n@pytest.fixture\ndef client():\n\tapp.app.config['TESTING'] = True\n\twith
          app.app.test_client() as client:\n\t\tyield client\n\n\ndef test_create_and_get_user(client):\n\tuser_data = {\n\t\t'
          email': 'test@example.com',\n\t\t'password': 'password',\n\t\t'profile_picture': 'picture.jpg',\n\t\t'status_message':
          'Hello!',\n\t\t'privacy_settings': 'public',\n\t\t'blocked_contacts': [],\n\t\t'groups': []\n\t}\n\tresponse = client.
          post('/users', json=user_data)\n\tassert response.status_code == 201\n\tassert response.get_json() == {'message': 'User
          created'}\n\n\tresponse = client.get('/users/test@example.com')\n\tassert response.status_code == 200\n\tassert
          response.get_json() == user_data"
    }
  ]
}

[CU] [Function] {"write_files_status": "success", "message": "All tests passed."}

[CU]
Has the sub task step been completed of: ```
Step 1: Start by setting up the basic structure of the application. This includes creating the main application file (app.py) and
    setting up a Flask application. Also, create a mock database using an in-memory dictionary to store user data. This
    dictionary should be able to store user information such as email, password, profile picture, status message, privacy
    settings, blocked contacts, and groups. Write tests to ensure that the application setup is correct and the mock database can
    store and retrieve data correctly.
```
If yes, call the function 'sub_task_step_complete', otherwise reflect and correct the full code to complete the task. Only use the
    functions you have been provided with, and if you want to output code only use the 'write_files' function to output code.
    Condition it on existing code: ['app.py', 'test_app.py'] Fully implement these features in the code, no placeholders. If you
    have not viewed the files before writing to them, please view them, to make sure you are writing to the correct files.
Respond now only with a function call of one of the following functions provided: 'sub_task_step_complete', 'view_files', '
    run_python_file', 'pytest_files', 'write_files', 'delete_files', and if you want to output code only use the 'write_files'
    function to output code.

[LLM] [Function Call] [pytest_files] {
  "files_to_test": ["test_app.py"]
}

[CU] [Function] {"output": "============================== test session starts ==============================\nplatform linux --
    Python 3.9.7, pytest-7.4.2, pluggy-1.3.0\nrootdir: /tmp/tmp0wam3ifj\nplugins: hydra-core-1.3.2, anyio-3.7.0\ncollected 1 item
    \n\ntest_app.py .                                                    [100%]\n\n============================== 1
    passed in 0.04s ============================== If a test is failing the error could be the code, or the test is incorrect,
    so feel free to overwrite and change the tests when they are incorrect, to make all tests pass."}

[CU] Has the sub task step been completed of: ```
Step 1: Start by setting up the basic structure of the application. This includes creating the main application file (app.py) and
    setting up a Flask application. Also, create a mock database using an in-memory dictionary to store user data. This
    dictionary should be able to store user information such as email, password, profile picture, status message, privacy
    settings, blocked contacts, and groups. Write tests to ensure that the application setup is correct and the mock database can
    store and retrieve data correctly.
```
If yes, call the function 'sub_task_step_complete', otherwise reflect and correct the full code to complete the task. Only use the
    functions you have been provided with, and if you want to output code only use the 'write_files' function to output code.
    Condition it on existing code: ['app.py', 'test_app.py'] Fully implement these features in the code, no placeholders. If you
    have not viewed the files before writing to them, please view them, to make sure you are writing to the correct files.
Respond now only with a function call of one of the following functions provided: 'sub_task_step_complete', 'view_files', '
    run_python_file', 'pytest_files', 'write_files', 'delete_files', and if you want to output code only use the 'write_files'
    function to output code.

[LLM] [Function Call] [sub_task_step_complete] {}

[CU] Please provide a one or two sentence summary of the output of this step, which is useful for the next step. Your response will
    be used when starting the next step without any of the previous messages.

[LLM] The basic structure of the application has been set up, including the main application file (app.py) and a Flask application.
    A mock database using an in-memory dictionary has been created to store user data, and tests have been written and passed to
    ensure that the application setup is correct and the mock database can store and retrieve data correctly.
```

The second instruction is loaded, and the process repeats.

## H.5 ADDITIONAL DIVERSE PROGRAMMING CODE GENERATION TASKS

We extended the codebase generation system design benchmark with three new, additional diverse programming code generation tasks. These are a recipe application, an event planner application, and a financial tracking application. Each task consists of a user prompt of listed features to implement, and Code-L2MAC produces the entire codebase from scratch. This is tabulated in Table 4.

We observe that Code-L2MAC continues to fully implement the highest percentage of user-specified feature requirements across these new diverse tasks, while its code contains minimal syntactical errors and passes a high number of unit tests—therefore, Code-L2MAC still achieves state-of-the-art for completing these system design large code generation benchmark tasks. Moreover, we also have implemented a new standard code quality metric of code coverage percentage of the unit tests (Miller & Maloney, 1963), labelled **Cov %**, and similarly observe Code-L2MAC also has a high code coverage percentage to its substantially larger amount of generated lines of code.

Table 4: Codebase generation system design task results showing the percentage of functional features specified that are fully implemented (**Features %**), the number of syntactical errors in the generated code (**# Errors**), the number of lines of code (**LOC**), number of passing tests (**Tests Passed**), and the unit test code coverage percentage (**Cov %**). Code-L2MAC fully implements the highest percentage of user-specified feature requirements across all tasks by generating fully functional code that has minimal syntactical errors and a high number of passing self-generated unit tests. The results are averaged over 10 random seeds, with ± indicating 95% confidence intervals.

| | Recipe App | | | | | Event Planner App | | | | | Financial Tracking App | | | | |
| Method | Features % ↑ | # Errors ↓ | LOC | Tests Passed ↑ | Cov % ↑ | Features % ↑ | # Errors ↓ | LOC | Tests Passed ↑ | Cov % ↑ | Features % ↑ | # Errors ↓ | LOC | Tests Passed ↑ | Cov % ↑ |
|---|---|---|---|---|---|---|---|---|---|---|---|---|---|---|---|
| GPT4 | 21.6±2.12 | 0±0 | 107±6.62 | 3.15±0.38 | **97.5±0.376** | 9.2±0.853 | 0.025±0.0506 | 74.6±4.12 | 1.75±0.395 | 88.7±5.5 | 26.2±4.67 | 0.0513±0.104 | 80.5±8.52 | 2.13±0.422 | 93.1±3.17 |
| CodeT | 20.5±4.86 | 0±0 | 96.5±13.8 | 3.05±0.879 | **97.8±0.523** | 11.2±1.12 | 0.05±0.105 | 75.2±8.77 | 2.45±0.704 | 92.5±10.2 | 21.4±3.25 | 0±0 | 65.9±6.93 | 2.25±0.368 | **97.9±0.209** |
| Self-Refine | 26±3.45 | 0.1±0.209 | 149±27.4 | 2±1.97 | 76.2±7.83 | 14.5±2.94 | 0.15±0.171 | 118±20.1 | 3.9±1.9 | 76.7±15 | 23.6±2.45 | 0.25±0.299 | 87.2±8.24 | 0.55±0.514 | 76.7±9.97 |
| Reflexion | 19±3.36 | 0.25±0.299 | 95.9±14.5 | 2.95±0.852 | 89.9±10.4 | 10±1.5 | 0±0 | 82±10.5 | 3±0.774 | 95.1±4.35 | 22.5±3.12 | 0.2±0.419 | 86.8±13.7 | 2.7±0.745 | 92.8±8.73 |
| AutoGPT | 39.2±14.9 | 1.85±1.45 | 106±19.1 | 1.3±2.02 | 9.8±14.1 | 35.7±32.9 | 0±0 | 23.9±20.7 | 0±0 | 0±0 | 32.9±44.2 | 0±0 | 25±15.8 | 0±0 | 0±0 |
| Code-L2MAC | **82±7.1** | 0±0 | 497±40.7 | 24.6±2.7 | 94.2±2.87 | **83±2.96** | 0±0 | 473±39.3 | 25.6±3.04 | 97.1±1.02 | **62±13.1** | 0±0 | 307±84.5 | 12±4.19 | 90.5±6.69 |

## H.6 Human Expert Validation of Features Implemented Percentage Metric

We hired two professional software engineers as human experts, separate from the authors of this work, to perform code reviews of the generated codebases for each method against the user-requested task feature checklist, counting only features that they verified are correctly and fully implemented. We regard the resulting metric, labeled human expert features percentage **Human Expert Features %**, as the ground truth.

We tabulate in Table 5 this metric below across three random seed runs. We highlight two conclusions. Code-L2MAC significantly outperforms other baselines based on **Human Expert Features %**. The human and LLM counterparts, **Human Expert Features %** and **Features %** strongly correlate ($\rho = 0.976$), thereby establishing **Features %** as a good proxy for the ground truth. This validates our usage of **Features %** as a scalable and cost-effective way to evaluate the number of features implemented in codebases from new method-task pairs. This conclusion aligns with existing literature on using LLMs as a proxy for human evaluators (Chiang & Lee, 2023).

Table 5: Codebase generation system design task results showing the percentage of functional features specified that are fully implemented (**Features %**), and ground truth metric of human experts counting the functional features specified that are fully implemeneted **Human Expert Features %**. Code-L2MAC fully implements the highest percentage of user-specified feature requirements across all tasks by generating fully functional code. The results are averaged over three random seeds.

| | URL Shortener App | | Online Social Media App | | Online Chat App | |
| Method | Human Expert Features % ↑ | Features % ↑ | Human Expert Features % ↑ | Features % ↑ | Human Expert Features % ↑ | Features % ↑ |
|---|---|---|---|---|---|---|
| GPT4 | 31.4 | 53.6 | 11.1 | 19.5 | 10 | 11 |
| AutoGPT | 15.7 | 25.3 | 6.35 | 33.3 | 15 | 23.1 |
| Code-L2MAC | **78.4** | **91.6** | **61.9** | **82.4** | **60** | **59.4** |

## H.7 Challenges and the Evaluation of Human-written Test-cases

We also explored using a priori hand-written test cases that are consistent across the different methods. We implemented this metric and present the results in Table 6 below, which shows it is correlated to our proposed main evaluation metric of **Features %**.

It is relevant to discuss some challenges this approach presents:

- **Hand written test cases assume and impose a known interface or a stringent implementation structure.** In small code snippet tasks, such as those in HumanEval (Chen et al., 2021) or MBPP (Austin et al., 2021), an explicitly defined function interface is explicitly presented to the LLM, and the LLM only responds with the code for that function body. In this situation handwritten test cases can assume the pre-defined implementation structure. However, in codebase generation tasks defined through feature requirements, there is freedom about the segmentation into components, modules or classes which must be appropriately determined by the code generation method. E.g., allowing a user to register an account can be achieved with many different code implementations. By specifying tests, we filter this ambiguity into a given implementation approach and we cannot account for *all other possible code implementation* approaches to implement a functional feature correctly.

- **Requires expert-crafted task-specific test cases a priori.** This hinders the scalability of this approach.

We added these hand written test cases into each method's context window $C^t$ throughout all stages of generation. Once the method generated a codebase, we then randomly changed the hand-written test case parameters to still be the same test, just with different test parameters, to avoid the method memorizing the test examples, e.g. changing the initial user_ids to a random value. Since all methods often generated a codebase that did not exactly match the implementation of the test-cases, while still having a codebase that would conceptually pass the purpose of the test, we used GPT4 to port the test cases to each specific codebase implementation. We term the proportion of such tests that pass human-written tests as **HT %**, which is the percentage of human tests that pass for a given codebase. As tabulated in Table 6, which is computed over 5 random seed runs, we observe that this metric correlates to our **Feature %** evaluation metric ($\rho = 0.695$), which further provides empirical evidence for such a metric.

It is important to point out that providing the test cases to the LLMs is not ideal since it hints at how to design the implementation.

Table 6: Codebase generation system design task results showing the percentage of functional features specified that are fully implemented (**Features %**), percentage of human written tests that pass (**HT %**), the number of syntactical errors in the generated code (**# Errors**), the number of lines of code (**LOC**), number of passing tests (**Tests Passed**) and the generated unit test code coverage percentage (**Cov %**). Code-L2MAC fully implements the highest percentage of user-specified feature requirements across the task by generating fully functional code that has minimal syntactical errors and a high number of passing self-generated unit tests. The results are averaged over 5 random seeds, with $\pm$ indicating 95% confidence intervals.

| Method | URL Shortener App | | | | | |
| | **Features %** ↑ | **HT %** ↑ | **# Errors** ↓ | **LOC** | **Tests Passed** ↑ | **Cov %** ↑ |
|---|---|---|---|---|---|---|
| GPT4 | 25±79.6 | 0±0 | 3.75±10.9 | 134±19.9 | 6.75±7.16 | 80.5±14.5 |
| CodeT | 13.2±42.1 | 11.1±35.4 | 0±0 | 126±14.4 | 7.75±3.98 | 86.8±4.57 |
| Self-Refine | 30.6±30.7 | 33.3±41.4 | 0.2±0.555 | 140±9.83 | 9±0 | 74.6±8.85 |
| Reflexion | 30.9±20.8 | 33.3±14.4 | 0±0 | 84.5±33.9 | 3.5±0.919 | 96.5±5.88 |
| Code-L2MAC | **76.5±33.3** | **41.7±54.7** | **0±0** | **286±172** | **10±9.09** | **83±8.72** |

Furthermore we also experimented with *not* including the hand written test cases in each method's context window throughout all stages of generation. If we otherwise follow the same setup as outlined above, we observe the following results as tabulated in Table 7. We observe that the **HT %** metric correlates to our **Feature %** evaluation metric ($\rho = 0.928$), which again further provide empirical evidence for such a metric.

### H.8    GENERATING 1,000+ LINES OF CODE WITH CODE-L2MAC

By removing the restrictions we imposed upon Code-L2MAC to economize api-calls, such as limiting the amount of instructions to 10, we get a variation we term Code-L2MAC-Large that we tested on the Online Chat application task where it reached reached over 1,000+ LOCs (5x the LOC of AutoGPT, the next highest) as shown in Table 8 below.

### H.9    CODE-L2MAC ABLATION WITH NO INSTRUCTION SUMMARIZATION MESSAGE

Code-L2MAC does not depend on the summary message $M_{rs}$ from completing the previous instruction when loading a new instruction and can tackle each instruction from scratch, without $M_{rs}$. Indeed, all the outputs regarding previously completed instructions are contained in the file store/external memory and can be accessed on demand.

We empirically verify this by performing an ablation of Code-L2MAC that removes this summarization message step, without affecting the quality of the output code by much, as shown in Table 9.

Table 7: Codebase generation system design task results showing the percentage of functional features specified that are fully implemented (**Features %**), percentage of human written tests that pass (**HT %**), the number of syntactical errors in the generated code (**# Errors**), the number of lines of code (**LOC**), number of passing tests (**Tests Passed**) and the generated unit test code coverage percentage (**Cov %**). Code-L2MAC fully implements the highest percentage of user-specified feature requirements across the task by generating fully functional code that has minimal syntactical errors and a high number of passing self-generated unit tests. The results are averaged over 5 random seeds, with $\pm$ indicating 95% confidence intervals.

| Method | URL Shortener App | | | | | |
| | **Features %** ↑ | **HT %** ↑ | **# Errors** ↓ | **LOC** | **Tests Passed** ↑ | **Cov %** ↑ |
|---|---|---|---|---|---|---|
| GPT4 | 37.6±13.3 | 20±29.9 | 0±0 | 94.2±15.4 | 2±1.76 | 85.6±18.1 |
| CodeT | 47.1±10.3 | 42.2±31.5 | 0±0 | 98±19 | 4.8±3.45 | 91.8±7.15 |
| Self-Refine | 50.6±16.8 | 46.7±49.2 | 0±0 | 109±12.9 | 3.4±2.26 | 92±1.96 |
| Reflexion | 55.3±18.3 | 37.8±33.2 | 0.6±1.67 | 124±31.3 | 3.4±2.99 | 87.6±12.8 |
| Code-L2MAC | **89.4±12** | **71.1±50.3** | **0±0** | **283±100** | **8.6±9.52** | **77.2±53.7** |

Table 8: Codebase generation system design task results showing the percentage of functional features specified that are fully implemented (**Features %**), the number of syntactical errors in the generated code (**# Errors**), the number of lines of code (**LOC**), and number of passing tests (**Tests Passed**). Code-L2MAC-Large can generate 1,000+ lines of code. The results are averaged over 10 random seeds, with $\pm$ indicating 95% confidence intervals.

| Method | Online Chat App | | | |
| | **Features %** ↑ | **# Errors** ↓ | **LOC** | **Tests Passed** ↑ |
|---|---|---|---|---|
| Code-L2MAC-Large | **53.3±19** | **0.333±1.43** | **1,030±40.8** | **5.67±13.7** |

We included this summarization message step to steer the LLM to find the correct files faster since we expect more similarity and interdependence between contiguous instructions than between more distant ones.

Table 9: Codebase generation system design task results showing the percentage of functional features specified that are fully implemented (**Features %**), the number of syntactical errors in the generated code (**# Errors**), the number of lines of code (**LOC**), and number of passing tests (**Tests Passed**). The results are averaged over 10 random seeds, with $\pm$ indicating 95% confidence intervals.

| Method | URL Shortener App | | | |
| | **Features %** ↑ | **# Errors** ↓ | **LOC** | **Tests Passed** ↑ |
|---|---|---|---|---|
| Code-L2MAC (Ablation, without instruction output summarization) | 89.4±9.88 | 0±0 | 274±43.3 | 8.2±3.12 |
| Code-L2MAC | **91.6±8.22** | **0±0** | **330±47.6** | **14±6.71** |

## H.10  ADDITIONAL TASKS ON IMPLEMENTING A NEW FEATURE IN AN EXISTING LARGE CODE BASE WITH CODE-L2MAC OF UP TO 165,000 LOCs

We added new three tasks of implementing a new feature in an existing large codebase with Code-L2MAC, where each codebase has at least 87,000+ LOCs, and one task up to 165,000 LOCs. We highlight that the task of taking an existing large codebase and implementing a new feature is common in software engineering (Lee et al., 2002). These tasks involve the following three existing open source codebases a of Dynamic Dashboard App (Tabor, 2023), Community Forum App (Justin, 2023) and a Data Exploration App (Breddels, 2023). For each task, we take that existing codebase and implement a new feature, which are a new button to delete all dashboards, a new front page statistic to show the number of newly registered users within the last 30 days, and to show the current time on the application banner, respectively. Unique to these new tasks is that instead of Code-L2MAC starting

from scratch, it starts with the existing codebase which it must correctly condition the generation of new code on. The results are tabulated in Table 10.

We observe that Code-L2MAC can implement the feature, and still has a high feature implementation percentage for these new feature implementation tasks, whilst working with a large codebase of up to 165,000 LOCs.

Table 10: Implementing a new feature in an existing large codebase task results showing the percentage of the runs that implemented the feature as specified that are fully implemented (**Feature %**) and the number of lines of code (**LOC**). Code-L2MAC can implement the feature whilst working with a large codebase up to 165,000 LOCs. The results are averaged over 10 random seeds, with ± indicating 95% confidence intervals.

| Method | Dynamic Dashboard App | | Community Forum App | | Data Exploration App | |
|---|---|---|---|---|---|---|
| | **Features %** ↑ | **LOC** | **Features %** ↑ | **LOC** | **Features %** ↑ | **LOC** |
| Code-L2MAC | 80±30.2 | 165,811.6±43.8 | 70±34.6 | 88,878.4±9.68 | 100±0 | 87,044.5±18.5 |

## I   FUTURE WORK

We envision the possible exciting future directions for future work.

1. **Error Checking**. More advanced error checking, checking the output of the LLM to control for hallucinations, and to prematurely avoid out-of-context errors—through the potential adoption of backpressure techniques, as employed in existing message-based networking systems (Tassiulas & Ephremides, 1990). Furthermore, a possible future direction to control the generation of correct code is to use further tools to run and verify the code, for example, a static code linter (to pick up initial syntax or similar errors) as feedback when the code has been generated for a sub-task instruction.

2. **Recursive planning and Reprogramming**. Another possible way to avoid out-of-context errors is to enable recursive planning of instruction into smaller sub-instructions and modify existing stored prompt program instructions $\mathcal{I}$ to include these. In addition, future work could explore the possibility of replacing the sub-steps altogether if the original instructions prove to be an ineffective plan. This could involve some reflection of the reason that rendered the current plan ineffective. Furthermore, *replanning* or reprorgramming could also be seen as replacing the sub-steps altogether, however the degree to which (percentage) the original prompt program instructions $\mathcal{I}$ that are re-programmed is an exciting future work direction.

3. **Multi-processing with a prompt-program**. Investigating and supporting multi-processing program flow patterns, where we execute and have access to two or more LLMs to use, and similarly with the associated tools as well.

4. **Support for comprehensive control flow (code flow) paradigms**: Investigating and supporting a wider range of control flow (code flow) paradigms, encompassing constructs like 'while' loops, 'switch' statements, 'if' statements, as well as advanced patterns such as asynchronous and coroutine patterns.

5. **Optimize the expected subtask length of a sub-task**. Given LLMs struggle with longer context windows (Liu et al., 2023a), perhaps better overall task performance can be achieved by reducing the average sub-task context length, of which there could be an optimum. Furthermore, there exists a further tradeoff with a smaller context window to optimize performance of both the overall task output performance and the compute performance of the LLM.

6. **LLM may only be able to solve tasks that can be expressed in natural language (text) that it has been trained on**. To complete tasks that it has not been trained on, such as implementing a new application in a new programming language, it may need to be further trained by fine-tuning related examples. This may also apply to the use of specialized tools as well, such as the use of special optimization tools, to leverage when planning how to decompose a task into sub-tasks.

7. **Generating prompt-programs with human machine collaboration**. Unique to prompt programs is that they are naturally human-readable, allowing a human to inspect them and interpret them, of which the human may make changes to the prompt program $\mathcal{I}$, by modifying or adding new parts to it. Furthermore, the architecture can also incorporate human input as a tool to decide the next control flow of the prompt program and or use that as part of the state to execute a sub-task instruction.

8. **Interpretable operation for safety-critical applications**. The entire process and flow of operation is implemented as natural language, allowing humans to inspect the operation and be interpretable to a human. This allows the operations to be debuggable and potentially reason about corner case states in the operation of such a prompt program.

9. **Building up semantic knowledge**. For example, building up a detailed, up-to-date live semantic knowledge of the large codebase—detailing how it works and what components do what. This semantic knowledge of how the codebase is structured could then be leveraged when completing a new sub-task instruction to have minimal additional read operations and can be further updated when the sub-task is complete, persisting this semantic knowledge of the generated code.

10. **More tools and varied formats of memory** Depending on the domain of the task, we could include more tools, such as the ones described in (Schick et al., 2023; Hu et al., 2023). Furthermore, the memory could hold other forms of information, such as images or speech, and LL2MAC could still interact with such information through image2text, text2image, image2image (or the parallel ones for audio and any other form of information).

11. **Specialized LLM agents**. We could consider having specialized LLM processors, each tuned for a different task such as planning, coding, summarizing, writing tests, etc. Then, the CU would choose which to instantiate at each step depending on the current goal.

12. **Other methods for reading**. It is worth investigating the impact on the performance of using other reading methods such as the common embedding k-NN (Borgeaud et al., 2022).

13. **Scalability and deployability**. Future work could investigate the token efficiency of execution. That is, minimizing the total number of tokens generated for task completion. A first effort could investigate fine-tuning an LLM to write code in the form of diffs instead of overwriting full files or making more fine-grained reads of files (sometimes, instead of reading the whole implementation, it could be enough to only read function and class names and possibly their descriptions).

14. **Implementation optimization**. There are two inherent limitations to the length of the generated code files in the current implementation of Code-L2MAC. Both can be readily addressed, providing fertile ground for future work. These are:

    - Reading a file involves outputting all its content into the LLM's context and writing a file implies rewriting it. This means that to avoid falling out of context, the maximum length for a file is the LLM context window, but, in fact, it is undesirable to have a file of length more than half the LLM's context window, since this will imply that the LLM cannot modify it without falling out of context. This limitation can be overcome by endowing the LLM with the capacity to selectively read and write parts of the file (e.g., function names and headers) in the same spirit as diffs are conducted in git. This is in line with the previous item on "Scalability and deployability".

    - All the code file paths are listed in the context window $C^t$. Therefore, the maximum number of file paths (for example, ['app.py,' 'test_app.py', '...']) listed in Code-L2MAC can have in memory is strictly less than the context length. This can be readily solved by allowing the LLM only to list the file paths inside a folder and the possibility to navigate inside a sub-folder. In such a case, the constraint would happen only regarding the degree in the file tree, but we could theoretically have infinite depth in the file store.

15. **Improving Code-L2MAC's outputs' efficiency**. It is straightforward to measure efficiency. The runtime computational and memory complexity of a test could be provided as feedback to the LLM, which could use this measure to reflect upon the efficiency of the current implementation and optimize it if necessary.

## I.1 L2MAC IN NON-CODING DOMAINS

The general-purpose property of stored-program computers and the motivation of L2MAC in such a framework suggest that L2MAC could inherit this general purpose.

First, it should be noted that the coding tasks cover a broad set of impactful applications, which, combined with our empirical results, suggests that L2MAC might encompass wide applicability.

Apart from that, we can consider other applications for this framework. As discussed in Section 3.1, a crucial consideration for L2MAC is the acknowledgment that LLMs are imperfect. Thus, the evaluation tools for instantiating L2MAC play an important role in its effectiveness (note that this role could become less and less crucial as LLMs progress). Consequently, additional applications that would render most immediate for the usage of this framework are the ones for which we have effective evaluation tools or where there is a less stringent requirement on the output structure. This might include AutoML, generating mathematical proofs (Li et al., 2021), or writing long textual documents; the first two have natural or existing verification tools (Bayer et al., 2022); the latter imposes less strict requirements, where an LLM could also be used as an evaluator.

The materialization of experiments on these tasks is beyond the scope of this paper; however, each of these poses exciting directions for future work.

