# OpenReview forum: "L2MAC: Large Language Model Automatic Computer for Extensive Code Generation"
_ICLR.cc/2024/Conference — ICLR 2024 poster_

### Official Review · Reviewer_Sz28 · 2023-10-30

**Soundness:** 4 excellent
**Presentation:** 3 good
**Contribution:** 4 excellent
**Rating:** 8
**Confidence:** 3

**Summary:**

The paper presents L2MAC, a framework for using large language models (LLMs) as automatic computers for long and consistent code generation. The framework consists of an LLM, an external memory (which stores both instructions and data),  and a control unit (which manages the interaction between the LLM and the external memory). The CU enables the LLM to execute a prompt program that contains a list of instructions to solve a user-given task. The CU also provides the LLM with precise read and write operations to access and update the memory, as well as error checking and correction mechanisms to ensure the quality and coherence of the generated code. The paper demonstrates the effectiveness of L2MAC by implementing Code-L2MAC, a practical instantiation of the framework for code generation tasks. Code-L2MAC can generate large code bases for system design tasks that require multiple components and features, outperforming existing SOTA methods such as GPT4 and AutoGPT.

**Strengths:**

+ The paper presents a novel approach with the introduction of the L2MAC framework, which augments LLMs by integrating memory and control mechanisms. This innovation stands as the first practical LLM-based stored-program computer.
+ The authors have instantiated the LLM SPC framework as Code-L2MAC, specifically tailored for intricate tasks like long code generation. The proposed method exhibits superior performance compared to state-of-the-art techniques.
+ The introduced benchmark and evaluation metrics for long code generation tasks are valuable for further research in this field.

**Weaknesses:**

+ In the manuscript, there appears to be an inconsistency in the usage of the terms "L2MAC" and "Code-L2MAC." The authors aim to differentiate between the overarching framework, denoted as L2MAC (which stands for LLM-based SPC), and its specific instantiation, referred to as Code-L2MAC, designed for long code generation tasks. Notably, the title of the paper erroneously employs "L2MAC" when "Code-L2MAC" would be more appropriate for the context of long code generation. Similar discrepancies are observed in the abstract, the first summarized contribution, Figure 1, and its accompanying text, among other sections. It is recommended that the authors undertake a meticulous revision to clearly delineate between these two terms. Additionally, while the L2MAC framework's primary focus is on the long code generation task, its potential applicability to a wider array of experiments should not be overlooked. To underscore the framework's versatility and generalizability, it would be advantageous for the authors to incorporate additional tasks.
+ Regarding the comparative analysis, the manuscript omits some pivotal baselines, notably the Reflecting LLMs. While the related works section enumerates three methodologies—Single LLMs, Reflecting LLMs, and Autonomous Agent LLMs with memory—the results presented in Table 2 only encompass those of GPT-4 and AutoGPT. This omission should be addressed to provide a more comprehensive evaluation.

**Questions:**

1. In Section 3.3.1, the authors mention that if none of the introduced errors are found, CU asks the LLM to summarize the generated output in the current context window. The degree of summarization determines whether contextual information will be lost, which is crucial for long context handling. Can you provide a detailed and clearer explanation?

---

> ### Author Response · Authors · 2023-11-17
> **Response to Reviewer Sz28 [Part 1/3]**
>
> We are delighted you appreciated the novelty of the L2MAC framework and the value of our benchmark. We structure our response as follows:
>
> * (A) Precise usage of L2MAC and Code-L2MAC
> * (B) L2MAC on Further Applications
> * (C) Reflecting LLM Baselines
> * (D) Summarization Between Instructions
>
> ---
>
> ### **(A) Precise Usage of L2MAC and Code-L2MAC**
>
> We agree with your comment on the precise use of L2MAC and Code-L2MAC and have meticulously gone through the main text to resolve these and clearly delineate between these two terms.
>
> **UPDATE:** We have meticulously gone through the paper and resolved these terms. We will change the title and the mentions within the abstract to Code-L2MAC where appropriate. At the rebuttal stage, we cannot change the title and the abstract, but we will update them for the camera-ready revision.
>
> ---
>
> ### **(B) L2MAC on Further Applications**
>
> We thank you for highlighting the _potential_ of L2MAC in _other applications_! Indeed, the general-purpose property of stored-program computers is fascinating, and the motivation of L2MAC in such a framework suggests it could inherit this property. We _restricted our claims_ to L2MAC's applicability to coding-related tasks to ensure they _stayed within_ what we could _verify empirically_. Still, we agree that it would be valuable to discuss the areas of applications where this framework could be explored in future work. We include these considerations in appendix I.1.
>
> First, _coding_ tasks cover a _broad_ set of impactful _applications_, which, combined with our empirical results, suggests that Code-L2MAC (and, by extension, thus, L2MAC) encompasses wide applicability. To strengthen this point, we increased the variety of our benchmark with new tasks, as discussed in the global response **(R2)**.
>
> Apart from that, we can consider other domains for this framework. As discussed in section 3.1, a crucial consideration for L2MAC is the acknowledgment that LLMs are imperfect. Thus, the _evaluation tools_ for instantiating L2MAC play an _important role_ in its effectiveness (note that this role could become less and less crucial as LLMs progress). Consequently, additional applications that would render most immediate for the usage of this framework are the ones for which we have effective evaluation tools or where there is a less stringent requirement on the output structure. This set might include _AutoML_, generating _mathematical proofs_ (Li et al., 2020), or writing _long textual documents_; the first two have natural or existing verification tools (Bayer et al., 2022); the latter imposes less strict requirements, where an LLM could also be used as an evaluator.
>
> Materializing experiments on these tasks is out of the scope of this paper, although each of them constitutes an exciting direction for future work.
>
> **UPDATE:** We have included the above discussion on the potential additional applications of L2MAC in appendix I.1. We have doubled the number of coding tasks in our benchmark to increase the variety of settings; see global response **(R2)**.
>
> ---

---

> ### Author Response · Authors · 2023-11-17
> **Response to Reviewer Sz28 [Part 2/3]**
>
> ### **(C) Reflecting LLM Baselines**
>
> We agree that comparing Code-L2MAC to reflecting LLM approaches would be valuable to provide a more comprehensive evaluation and to make our experiments consistent with the related work section. Consequently, we have included in all our experiments _three new baselines_ of more recent state-of-the-art reflecting code generation methods: **CodeT** (Chen et al., 2022), **Self-Refine** (Madaan et al., 2023), and **Reflexion** (Shinn et al., 2023). Code-L2MAC significantly outperforms these reflecting baselines throughout all tasks in the benchmark.
>
> As described in the global response **(R1)**, these three new reflecting baselines are now included in the updated main experimental results table 2, also shown below. Code-L2MAC still fully implements the highest percentage of user-specified feature requirements across all tasks, where its code has minimal syntactical errors and a high number of self-generated unit tests. Therefore, _Code-L2MAC is state-of-the-art_ for these large code generation benchmark tasks.
>
> **Table 2.** (Main experimental results averaged over ten random seeds)
>
> **URL Shortener App**
> | Method | Features % | # Errors | LOC | Tests Passed |
> | ------------ | -------------------- | ------------- | -------------- | ----------------- |
> | | ↑ | ↓ | | ↑ |
> | GPT4 | 53.6±10.5 | 0±0 | 119±21.1 | 2.56±0.95 |
> | CodeT | 52.9±6.74 | 0.05±0.105 | 110±11.8 | 3.6±0.513 |
> | Self-Refine | 47.9±8.53 | 0.05±0.105 | 124±15.7 | 3.65±1.15 |
> | Reflexion | 38.8±6.02 | 0.1±0.209 | 96.2±9.11 | 2.35±0.631 |
> | AutoGPT | 25.3±19.6 | 0±0 | 136±41.9 | 3.3±1.91 |
> | Code-L2MAC | **91.6±8.22** | **0±0** | **330±47.6** | **14±6.71** |
>
> **Online Social Media App**
> | Method | Features % | # Errors | LOC | Tests Passed |
> | ------------ | -------------------- | ------------- | -------------- | ----------------- |
> | | ↑ | ↓ | | ↑ |
> | GPT4 | 19.5±8.28 | 4.09±3.32 | 116±31.5 | 0.818±0.785 |
> | CodeT | 19.5±5.19 | 0.4±0.603 | 106±17.7 | 2.6±1.76 |
> | Self-Refine | 16.4±2.62 | 0.938±0.714 | 110±19.6 | 1.81±0.938 |
> | Reflexion | 15.2±8.05 | 2.53±1.69 | 122±24 | 1.33±2.44 |
> | AutoGPT | 33.3±18 | 0.6±0.369 | 148±35.5 | 3±2.86 |
> | Code-L2MAC | **82.4±14.6** | **0±0** | **395±52.9** | **18.3±6.8** |
>
> **Online Chat App**
> | Method | Features % | # Errors | LOC | Tests Passed |
> | ------------ | -------------------- | ------------- | -------------- | ----------------- |
> | | ↑ | ↓ | | ↑ |
> | GPT4 | 11±2.26 | 0.3±0.346 | 127±24.1 | 1.2±1 |
> | CodeT | 10.5±4.61 | 0±0 | 91.6±25.9 | 3.32±1.57 |
> | Self-Refine | 14.2±4.19 | 0.211±0.304 | 111±13.8 | 1.42±0.927 |
> | Reflexion | 10.2±3.08 | 0±0 | 76±6.88 | 2.85±0.822 |
> | AutoGPT | 23.1±11.8 | 1.85±2.47 | 220±65.8 | 3.08±3.34 |
> | Code-L2MAC | **59.4±25.9** | **0±0** | **374±123** | **18.8±9.11** |
>
> **UPDATE:** We share the reviewer's opinion about the importance of how Code-L2MAC compares to existing reflecting methods and have included _three new_ reflecting _baselines_ in our experiments in table 2.
>
> ---

---

> ### Author Response · Authors · 2023-11-17
> **Response to Reviewer Sz28 [Part 3/3]**
>
> ### **(D) Summarization Between Instructions**
>
> We agree that the degree of summarization affects whether contextual information will be preserved in the output, and this is, in fact, a shortcoming of the methods that rely on such summarization, as we suggest in the Related Work comparison with Autonomous Agents LLMs: _"(2) they compress the previous action history, thus usually losing critical information in the process. In code generation tasks, (2) indicates forgetting which files exist and their content"._
>
> You rightly point out that in section 3.1.1. we describe how when the LLM signals the current instruction has been completed, the CU calls the evaluator on the current state of the file store, and if _"none [errors] are found, it [the CU] asks the LLM to summarize the generated output in the current context window."_ Where this summary is provided to the LLM along with the next instruction $\mathcal{I}^{(k+1)}$ to tackle.
>
> However, our method does not depend upon this summarization message $M\_{rs}$ step and can tackle each instruction from scratch. Indeed, all the essential information regarding previous instructions outputs is contained in the file store/external memory and can be accessed on-demand. We empirically verify this by performing an _ablation_ of our method that _removes_ this _summarization message step without affecting the quality of the output code by much_, as shown below in table 8.
>
> **Table 8.**
> | Method | Features % | # Errors | LOC | Tests Passed |
> | ---------------------------------------------------------- | ----------------- | --------- | -------------- | ---------------- |
> | Code-L2MAC (Ablation, without instruction output summarization) | 89.4±9.88 | **0±0** | 274±43.3 | 8.2±3.12 |
> | Code-L2MAC | **91.6±8.22** | **0±0** | **330±47.6** | **14±6.71** |
>
> We included this summarization message step to steer the LLM to find the correct files faster since we expect more similarity and interdependence between contiguous instructions than between more distant ones.
>
> **UPDATE**: We evaluated an ablation of Code-L2MAC that omits the summarization message step when having completed an instruction and loading a new instruction. This ablation shows it does not significantly affect the code's quality. We incorporated these results in a **(new) appendix N**.
>
> ---
>
>
> **References**
>
> - Wenda Li, Lei Yu, Yuhuai Wu, and Lawrence C. Paulson. Isarstep: a benchmark for high-level mathematical reasoning, 2021.
> - Jonas Bayer, Christoph Benzmüller, Kevin Buzzard, Marco David, Leslie Lamport, Yuri Matiyasevich, Lawrence Paulson, Dierk Schleicher, Benedikt Stock, and Efim Zelmanov. Mathematical proof between generations, 2022.
> - Bei Chen, Fengji Zhang, Anh Nguyen, Daoguang Zan, Zeqi Lin, Jian-Guang Lou, and Weizhu Chen. Codet: Code generation with generated tests. arXiv preprint arXiv:2207.10397, 2022.
> - Aman Madaan, Niket Tandon, Prakhar Gupta, Skyler Hallinan, Luyu Gao, Sarah Wiegreffe, Uri Alon, Nouha Dziri, Shrimai Prabhumoye, Yiming Yang, et al. Self-refine: Iterative refinement with self-feedback. arXiv preprint arXiv:2303.17651, 2023.
> - Noah Shinn, Federico Cassano, Ashwin Gopinath, Karthik R Narasimhan, and Shunyu Yao. Reflexion: Language agents with verbal reinforcement learning. In Thirty-seventh Conference on Neural Information Processing Systems, 2023.

---

### Official Review · Reviewer_4yCT · 2023-10-31

**Soundness:** 3 good
**Presentation:** 3 good
**Contribution:** 3 good
**Rating:** 8
**Confidence:** 4

**Summary:**

The paper proposes L2MAC, a practical LLM-based stored-program automatic computer for long and consistent code generation. The experimental results show L2MAC outperforms GPT-4 and AutoGPT for a variety of tasks, including URL shortening, online microblogging, and online chat applications.

**Strengths:**

The paper introduces a practical LLM-based stored-program automatic computer framework-- L2MAC, for long code generation tasks. L2MAC can generate code for a variety of tasks, including URL shortening, online microblogging, and online chat applications, and outperform SOTA works.

**Weaknesses:**

1.	The performance of L2MAC is evaluated on URL shortening, online microblogging, and online chat applications. Can this method be used for other applications, such as programming?
2.	Is there a limit on the length of the generated code files?
3.	How to ensure the efficiency of generated code files?

**Questions:**

Please refer to weaknesses.

---

> ### Author Response · Authors · 2023-11-16
> **Response to Reviewer 4yCT [Part 1/2]**
>
> Thank you for your comments and suggestions, which have driven improvements in the paper! We organize our response as follows:
>
> * (A) Additional Tasks
> * (B) Practical Output Limits of Code-L2MAC
> * (C) How do we Ensure the Efficiency of Generated Code Files?
>
> ---
>
> ### **(A) Additional Tasks**
>
> We agree with your suggestion; as explained in the global response **(R2)**, we included three _new programming tasks_: a **recipe application**, an **event planner application,** and a **financial tracking application**. The new results are tabulated in table 4 (new appendix J) and also shown below.
>
> Code-L2MAC continues to achieve state-of-the-art results on these large code generation benchmark tasks.
>
> **Table 4.** (Three new task experimental results averaged over ten random seeds)
>
> **Recipe App**
> | Method | Features % | # Errors | LOC | Tests Passed |
> | ------------ | -------------------- | ------------- | -------------- | ----------------- |
> | | ↑ | ↓ | | ↑ |
> | GPT4 | 21.6±2.12 | 0±0 | 107±6.62 | 3.15±0.38 |
> | CodeT | 20.5±4.86 | 0±0 | 96.5±13.8 | 3.05±0.879 |
> | Self-Refine | 26±3.45 | 0.1±0.209 | 149±27.4 | 2±1.97 |
> | Reflexion | 19±3.36 | 0.25±0.299 | 95.9±14.5 | 2.95±0.852 |
> | AutoGPT | 39.2±14.9 | 1.85±1.45 | 106±19.1 | 1.3±2.02 |
> | Code-L2MAC | **82±7.1** | **0±0** | **497±40.7** | **24.6±2.7** |
>
> **Event Planner App**
> | Method | Features % | # Errors | LOC | Tests Passed |
> | ------------ | -------------------- | ------------- | -------------- | ----------------- |
> | | ↑ | ↓ | | ↑ |
> | GPT4 | 9.2±0.853 | 0.025±0.0506 | 74.6±4.12 | 1.75±0.395 |
> | CodeT | 11.2±1.12 | 0.05±0.105 | 75.2±8.77 | 2.45±0.704 |
> | Self-Refine | 14.5±2.94 | 0.15±0.171 | 118±20.1 | 3.9±1.9 |
> | Reflexion | 10±1.5 | 0±0 | 82±10.5 | 3±0.774 |
> | AutoGPT | 35.7±32.9 | 0±0 | 23.9±20.7 | 0±0 |
> | Code-L2MAC | **83±2.96** | **0±0** | **473±39.3** | **25.6±3.04** |
>
> **Financial Tracking App**
> | Method | Features % | # Errors | LOC | Tests Passed |
> | ------------ | -------------------- | ------------- | -------------- | ----------------- |
> | | ↑ | ↓ | | ↑ |
> | GPT4 | 26.2±4.67 | 0.0513±0.104 | 80.5±8.52 | 2.13±0.422 |
> | CodeT | 21.4±3.25 | 0±0 | 65.9±6.93 | 2.25±0.368 |
> | Self-Refine | 23.6±2.45 | 0.25±0.299 | 87.2±8.24 | 0.55±0.514 |
> | Reflexion | 22.5±3.12 | 0.2±0.419 | 86.8±13.7 | 2.7±0.745 |
> | AutoGPT | 32.9±44.2 | 0±0 | 25±15.8 | 0±0 |
> | Code-L2MAC | **62±13.1** | **0±0** | **307±84.5** | **12±4.19** |
>
> **Update:** We now include these new benchmark tasks for all the baselines in a **(new) appendix J** entitled "Additional Diverse Programming Code Generation Tasks".
>
> ---
>
> ### **(B) Practical Output Limits of Code-L2MAC**
>
> There are two inherent limitations to the length of the generated code files in the current implementation of Code-L2MAC. Both are discussed within the paper (section 4, footnote 7, and section 7), and both can be readily addressed, providing fertile ground for future work. These are:
>
> - _Code files should be smaller than the context window constraint $c$._ Reading a file involves outputting its whole content into the LLM's context, and writing a file implies rewriting it. This means that to avoid falling out of context, the maximum length for a file is the LLM's context window size, but in fact, it is undesirable to have a file of length more than half the LLM's context window size since this will imply that the LLM cannot modify it without falling out of context. This limitation can be overcome by endowing the LLM with the capacity to selectively read and write parts of the file (e.g., function names and headers) in the same spirit as diffs are conducted in git. We discuss this and its implications for the method's efficiency in Future Work (app. I).
> - _All the code file paths are listed in the context window $C^t$._ Therefore, the maximum number of file paths (e.g. ['app.py,' 'test\_app.py,' …]) listed Code-L2MAC can have in memory is strictly less than the context length. This can be readily solved by allowing the LLM only to list the file paths inside a folder and the possibility to navigate inside a sub-folder. In such a case, the constraint would happen only regarding the degree in the file tree, but we could theoretically have infinite depth in the file store.
>
> **UPDATE:** We incorporate a version of the above discussion into the existing future work appendix I.
>
> ---

---

> ### Author Response · Authors · 2023-11-16
> **Response to Reviewer 4yCT [Part 2/2]**
>
> ### **(C) How do we Ensure the Efficiency of Generated Code Files?**
>
> Thank you for this exciting question. Indeed, the efficiency of the generated code is crucial in many real-world applications. Although this dimension is not currently considered in our current implementation of Code-L2MAC, there is a straightforward way in which this could be incorporated since it is easy to quantify efficiency. The _runtime_ computational, and memory _complexity_ of a test could be provided as _feedback_ to the LLM, which could use this measure to reflect upon the efficiency of the current implementation and _optimize_ it if necessary.
>
> Nonetheless, this does not constitute a crucial component of our implementation, intended to act as the first instantiation of our framework. Other potential refinements are discussed in our Future Work (app. I), and we encourage (and consider) pursuing an optimized implementation of Code-L2MAC in future work.
>
> **UPDATE:** We included the above discussion in our Future Work section in appendix I as a new item.

---

> > ### Comment · Reviewer_4yCT · 2023-11-19
> >
> > Thanks for the authors’ responses. These have addressed my questions. I have changed my score to accept.

---

> > > ### Author Response · Authors · 2023-11-19
> > > **Gratitude for Revised Review and Score Increase**
> > >
> > > Thank you very much for your thoughtful consideration and the time you have dedicated to reviewing our paper. Your feedback was instrumental in enhancing our work, through extensive new experiments and explanations, and we are grateful for your increased score. Thank you once again!

---

### Official Review · Reviewer_4zmV · 2023-11-07

**Soundness:** 3 good
**Presentation:** 2 fair
**Contribution:** 3 good
**Rating:** 6
**Confidence:** 3

**Summary:**

The paper presents a LLM-based computer that uses LLM in its core as the `computation` engine and enables the model to interact with two different memories, `instruction registry` as a storage for prompts and user-defined instructions and `file store` as a means to store intermediate outputs. The authors create an interesting synergy with how conventional computers (e.g. Von Neumann architecture) operate and aims to replace the core compute and control engine with LLMs. As one of the applications, the author explored how such LLM-enabled computing platform can productively be used for the task of code generation.

**Strengths:**

$\mathtt{+}$ I found the synergy between conventional von Neumann architecture and L2MAC interesting and how the authors created a 1to1 mapping between different components in conventional computing platforms and their proposed design.

$\mathtt{+}$ The results for code generation tasks is promising.

**Weaknesses:**

$\mathtt{-}$ While I think the paper proposes an interesting idea, but I found the writing very challenging and difficult to understand and follow.

$\mathtt{-}$ While the general-purpose computers can excel work in a variety of task, L2MAC focuses on one particular task and it is not clear how such model can generalized to different application and programs.

$\mathtt{-}$ While the core idea is still new, most of the explored idea like self-refinement, using external memory, etc. have been explored before in the literature.

**Questions:**

This is an interesting and timely idea. While the authors only explore one application for such general purpose computer, it would be interesting to see what other applications this computer can enable. To be honest, I found the writing of the paper/formulating the idea challenging to follow and that makes contributions less clear.

(Q1) I appreciate the authors providing a comparison with the related work in Table1. I am wondering if it would make sense to have a comparison with those in the scope of code generation. I am curious to see how your approach compares with reflecting LLM techniques.

(Q2) I understand the choice of target application for your model, but how do you think L2MAC can be extended to cover other applications and domains?

(Q3) Can you clarify how do you generate test programs for each applications? Do you have another verifier to ensure the correctness of the unit test? What would happen if the unit test are limited coverage in testing the target application?

(Q4) I also find the metric of `feature %` to be confusing. As an end-user, I would like my program to be fully functional and correct. Why do you need to design a new metric for the evaluation? How do you relate your metric with correctness of the program? Is there any correlation here?

---

> ### Author Response · Authors · 2023-11-16
> **Response to Reviewer 4zmV [Part 1/5]**
>
> Thanks for your thoughtful comments and suggestions! We are glad you enjoyed the motivation of our framework through the von Neumann architecture. Please find our answers below and the corresponding updates to the revised submission. We structure our response as follows:
>
> * (A) Clarified Writing
> * (B) Generalizing to Other Tasks
> * (C) Comparison with Reflecting LLMs
> * (D) New Ground-truth Metrics and Discussion on Previous Ones
>   * (D.1) Ground-truth Human Expert Validation Justifies the Use of Feature %
>   * (D.2) Generated Tests' Coverage
>   * (D.3) Challenges of Using an Existing Evaluation Metrics
>
> ----
>
> ### **(A) Clarified Writing**
>
> Thank you for flagging this.
>
> **UPDATE**: We have carefully reviewed the manuscript and have _identified and fixed typos_ that could lead to confusion.
>
> ---
>
> ### **(B) Generalizing to Other Tasks**
>
> We thank you for highlighting the _potential_ of L2MAC in _other applications_! Indeed, the general-purpose property of stored-program computers is fascinating, and the motivation of L2MAC in such a framework suggests it could inherit this property. We _restricted our claims_ to L2MAC's applicability to coding-related tasks to ensure they _stayed within_ what we could _verify empirically_. Still, we agree that it would be valuable to discuss the areas of applications where this framework could be explored in future work. We include these considerations in appendix I.1.
>
> First, _coding_ tasks cover a _broad_ set of impactful _applications_, which, combined with our empirical results, suggests that Code-L2MAC (and, by extension, thus, L2MAC) encompasses wide applicability. To strengthen this point, we increased the variety of our benchmark tasks with new tasks, as discussed in the global response **(R2)**.
>
> Apart from that, we can consider other domains for this framework. As discussed in section 3.1, a crucial consideration for L2MAC is the acknowledgment that LLMs are imperfect. Thus, the _evaluation tools_ for instantiating L2MAC play an _important role_ in its effectiveness (note that this role could become less and less crucial as LLMs progress). Consequently, additional applications that would render most immediate for the usage of this framework are the ones for which we have effective evaluation tools or where there is a less stringent requirement on the output structure. This set might include _AutoML_, generating _mathematical proofs_ (Li et al., 2020), or writing _long textual documents_; the first two have natural or existing verification tools (Bayer et al., 2022); the latter imposes less strict requirements, where an LLM could also be used as an evaluator. Materializing experiments on these tasks is out of the scope of this paper, although each of them constitutes an exciting direction for future work.
>
> **UPDATE:** We have included the above discussion on the potential additional applications of L2MAC in appendix I.1. We have doubled the number of coding tasks in our benchmark to increase the variety of settings; see global response (R2).
>
> ---

---

> ### Author Response · Authors · 2023-11-16
> **Response to Reviewer 4zmV [Part 2/5]**
>
> ### **(C) Comparison with Reflecting LLMs**
>
> We agree that comparing Code-L2MAC to reflecting LLM techniques would be helpful. We thus included _three new baselines_ that represent state-of-the-art reflecting code generation methods: **CodeT** (Chen et al., 2022), **Self-Refine** (Madaan et al., 2023), and **Reflexion** (Shinn et al., 2023).
>
> We performed a complete re-run of these and the previous methods across all tasks, including three new ones; see the general response **(R2)**. We made the comparison fairer by providing these baselines with the same tools that Code-L2MAC uses; we provide the implementation details and hyperparameters in a newly expanded appendix F.
>
> As you rightly mentioned, our _central contribution_ is introducing the first practical _LLM-based stored-program computer_. In addition to that, as you point out, our implementation incorporates concepts of refinement and external memory as discussed in previous work, which is further discussed in the _related work_, section 5, and the extended related work (appendix D), which we do not claim as contributions. Specifically, we find the following differences between Code-L2MAC and reflecting LLM methods, such as Self-Refine and Reflexion to be:
>
> - Self-Refine refines the _most recent_ output from the LLM and Reflexion the _recent outputs_ that are only within the LLM's current context window $C^t$; whereas Code-L2MAC can refine the entire file store, encompassing all previous outputs from the LLM—allowing it to fix or improve earlier outputs from multiple instructions back, outside its context window, which are now erroring due to a new code implementation change. This enables Code-L2MAC to generate long, consistent, and interrelated code structures.
> - Self-Refine and Reflexion are constrained to operating on an output within the context window constraint $c$, whereas Code-L2MAC can manage a total output greater than the context window constraint through the L2MAC framework.
>
> **UPDATE:** We share the reviewer's curiosity about how Code-L2MAC compares to existing reflecting methods and have included _three new_ reflecting _baselines_ in our experiments (see R2 in the global response). We also agree that the novelty of memory that stems from the stored-program computer framework to existing memory augmentation literature might not be apparent, so we _updated_ our _extended related work_ to include a version of the above discussion.
>
> ---

---

> ### Author Response · Authors · 2023-11-16
> **Response to Reviewer 4zmV [Part 3/5]**
>
> ### **(D) New Ground-truth Metrics and Discussion on Previous Ones**
>
> (D.1) We start by addressing your questions regarding how Feature % correlates with the correctness of the program, and (D.2) we discuss your great point about code coverage. Finally, (D.3), we discuss why the existing code evaluation metrics through human-written tests are not appropriate for our benchmark.
>
> **(D.1) Ground-truth Human Expert Validation Justifies the Use of Feature %**
>
> To elucidate the motivation of **Feature %**, we note that the metric _"Is it fully functional?"_ is very ***sparse*** and does not reflect the degrees of correctness that an implementation can have. As an extreme case, it is tough to prove that a given code base contains no bugs, and in practice, most code bases contain bugs. This does not imply that they are not functional, although each time a bug is correctly fixed, we can agree that the code base has gotten more functional. Consequently, asking, _"To what degree is it functional?"_ is more reasonable. A way to quantify this is to ask _"Of the_ ***functions/features*** _that I expect this code base to fulfill correctly,_ ***how many are actually fulfilled*** _?"_. This is what feature % quantifies. As a proxy, and in line with previous literature (Chiang et al., 2023), we quantify this by having GPT4 assess what features specified by the user were implemented fully as code (see appendix E for example prompts).
>
> This being established, we agree with the added value of a _ground truth_ human evaluation of the resulting code that unambiguously validates the result is fully functional and fulfills the requirements in the task description.
>
> To that end, we _hired two professional software engineers as human experts_, separate from the authors of this work, to perform code reviews of the generated code bases for each method against the user-requested task feature checklist, counting only features that they verified were correctly and fully implemented. We regard the resulting metric, labeled **Human Expert Features %**, as the ground truth.
>
> We tabulate (in table 5 below) this metric across three random seed runs. We highlight two conclusions.
>
> - Code-L2MAC significantly outperforms other baselines based on **Human Expert Features %.**
> - The human and LLM counterparts, **Human Expert Features %** and **Features %,** strongly correlate ($\rho=0.976$), thereby establishing **Features %** as a good proxy for the ground truth. This validates our usage of **Features %** as a scalable and cost-effective way to evaluate the amount of features implemented in code bases from new method-task pairs. This conclusion aligns with existing literature on using LLMs as a proxy for human evaluators (Chiang et al., 2023).
>
> **Table 5.**
> | | URL Shortener App | Online Social Media App | Online Chat App |
> |-------|---------------------------------------|---------------------------------------|---------------------------------------|
> | Method | Human Expert Features % | Features % | Human Expert Features % | Features % | Human Expert Features % | Features % |
> | | ↑ | ↑ | ↑ | ↑ | ↑ | ↑ |
> | GPT4 | 31.4 | 53.6 | 11.1 | 19.5 | 10 | 11 |
> | AutoGPT | 15.7 | 25.3 | 6.35 | 33.3 | 15 | 23.1 |
> | Code-L2MAC | **78.4** | **91.6** | **61.9** | **82.4** | **60** | **59.4** |
>
> **UPDATE:** We now include the human expert validation results and an adaptation of the above discussion in an additional **(new) appendix K** labeled "Human Expert Validation of Features Implemented Percentage Metric." We also included the above justification for the Feature % metric in a new subsection of appendix G (evaluation metrics).
>
> ---

---

> ### Author Response · Authors · 2023-11-16
> **Response to Reviewer 4zmV [Part 4/5]**
>
> **(D.2) Generated Tests' Coverage**
>
> Regarding generated tests, we request that the LLM generate tests that verify the validity of the implemented code, which are also used for reflection. We agree that such tests are a proxy for such validity since they could come in the form of mock tests that do not reflect the validity of the code, but this does not seem to be the case, as shown by the fact that _Code-L2MAC can fix tests that start failing throughout the execution_ (mock tests would never fail), while for example, _AutoGPT accumulates failing tests_, as shown in section 6.2.
>
> We thank you for pointing out the notion of coverage for the tests, which opens the door to help understand more precisely the implications of the passed and failed tests. Correspondingly, we incorporated a new metric reflecting the _coverage percentage_ of the tested code. Using the standard test suite quality metric of code coverage percentage of the unit tests (Miller & Maloney, 1963), we observe that Code-L2MAC has a high code coverage percentage despite its substantially larger amount of generated lines of code. As shown in table 4, in the global response **(R2)**, Code-L2MAC achieves an average coverage percentage of 93.93% across three tasks.
>
> **UPDATE:** We now include a new coverage evaluation metric for all the new tasks and baselines in a **(new) appendix J** and include this metric in our evaluation metrics in appendix G.
>
> ---
>
> **(D.3) Challenges of Using an Existing Evaluation Metric of Human-written Test Cases to Validate the Code**
>
> We also explored using a priori hand-written test cases that are consistent across the different methods, which is the existing way to evaluate small code snippet generation tasks. We implemented this metric and present the results in table 6 below, which shows it is correlated to our main evaluation metric of **Features %.**
>
> It is relevant to start with some challenges this metric presents:
>
> 1. **Handwritten test cases assume and impose a known interface or a stringent implementation structure**. In small code, snippet tasks, such as those in HumanEval (Chen et al., 2021) or MBPP (Austin et al., 2021), an explicitly defined function interface is explicitly presented to the LLM, and the LLM only responds with the code for that function body. In this situation, handwritten test cases can assume the pre-defined implementation structure. However, in code base generation tasks defined through feature requirements, there is freedom about the segmentation into components, modules, or classes, which must be appropriately determined by the code generation method. For example, allowing users to register an account can be achieved with many code implementations. By specifying _tests, we filter this ambiguity into a given implementation approach,_ and we _cannot_ account for _all other possible code implementation_ approaches to implement a functional feature correctly.
> 2. **Requires expert-crafted task-specific test cases a priori.** This hinders the scalability of this approach.
>
> We added these handwritten test cases into each method's context window $C^t$ throughout all generation stages. Once the method generated a code base, we then randomly changed the hand-written test case parameters to still be the same test, just with different test parameters, to avoid the method memorizing the test examples, e.g., changing the initial user_ids to a random value. Since all methods often generated a code base that did not precisely match the implementation of the test cases while still having a code base that would conceptually pass the test purpose, we used GPT4 to port the test cases to each specific code base implementation. We term the proportion of such tests that pass human-written tests as **HT %**, the percentage of human tests that pass for a given code base. Table 6, which is computed over five random seed runs, shows that this metric correlates to our **Feature %** evaluation metric ($\rho=0.695$), which further provides empirical evidence for such a metric.
>
> **Table 6.**
> | Method | Features % | HT % | # Errors | LOC | Tests Passed | Cov % |
> |------------|------------------|------------------|------------------|---------------|--------------------|-------------------|
> | | ↑ | ↑ | ↓ | | ↑ | ↑ |
> | GPT4 | 25±79.6 | 0±0 | 3.75±10.9 | 134±19.9 | 6.75±7.16 | 80.5±14.5 |
> | CodeT | 13.2±42.1 | 11.1±35.4 | 0±0 | 126±14.4 | 7.75±3.98 | 86.8±4.57 |
> | Self-Refine| 30.6±30.7 | 33.3±41.4 | 0.2±0.555 | 140±9.83 | 9±0 | 74.6±8.85 |
> | Reflexion | 30.9±20.8 | 33.3±14.4 | 0±0 | 84.5±33.9 | 3.5±0.919 | 96.5±5.88 |
> | Code-L2MAC | **76.5±33.3** | **41.7±54.7** | **0±0** | **286±172** | **10±9.09** | **83±8.72** |
>
> **UPDATE:** We have now included an extended version of this discussion in a **(new) appendix L** entitled "Challenges and the Evaluation of Human-written Test-cases."
>
> ---

---

> ### Author Response · Authors · 2023-11-16
> **Response to Reviewer 4zmV [Part 5/5]**
>
> **References:**
>
> - Wenda Li, Lei Yu, Yuhuai Wu, and Lawrence C. Paulson. Isarstep: a benchmark for high-level mathematical reasoning, 2021.
> - Jonas Bayer, Christoph Benzmüller, Kevin Buzzard, Marco David, Leslie Lamport, Yuri Matiyasevich, Lawrence Paulson, Dierk Schleicher, Benedikt Stock, and Efim Zelmanov. Mathematical proof between generations, 2022.
> - Bei Chen, Fengji Zhang, Anh Nguyen, Daoguang Zan, Zeqi Lin, Jian-Guang Lou, and Weizhu Chen. Codet: Code generation with generated tests. arXiv preprint arXiv:2207.10397, 2022.
> - Aman Madaan, Niket Tandon, Prakhar Gupta, Skyler Hallinan, Luyu Gao, Sarah Wiegreffe, Uri Alon, Nouha Dziri, Shrimai Prabhumoye, Yiming Yang, et al. Self-refine: Iterative refinement with self-feedback. arXiv preprint arXiv:2303.17651, 2023.
> - Noah Shinn, Federico Cassano, Ashwin Gopinath, Karthik R Narasimhan, and Shunyu Yao. Reflexion: Language agents with verbal reinforcement learning. In Thirty-seventh Conference on Neural Information Processing Systems, 2023.
> - Chiang, Cheng-Han, and Hung-yi Lee. "Can Large Language Models Be an Alternative to Human Evaluations?." _arXiv preprint arXiv:2305.01937_ (2023).
> - Joan C Miller and Clifford J Maloney. Systematic mistake analysis of digital computer programs. Communications of the ACM, 6(2):58–63, 1963.
> - Mark Chen, Jerry Tworek, Heewoo Jun, Qiming Yuan, Henrique Ponde de Oliveira Pinto, Jared Kaplan, Harri Edwards, Yuri Burda, Nicholas Joseph, Greg Brockman, et al. Evaluating large language models trained on code. arXiv preprint arXiv:2107.03374, 2021.
> - Jacob Austin, Augustus Odena, Maxwell Nye, Maarten Bosma, Henryk Michalewski, David Dohan, Ellen Jiang, Carrie Cai, Michael Terry, Quoc Le, et al. Program synthesis with large language models. arXiv preprint arXiv:2108.07732, 2021.

---

### Official Review · Reviewer_v4fH · 2023-11-08

**Soundness:** 2 fair
**Presentation:** 3 good
**Contribution:** 2 fair
**Rating:** 6
**Confidence:** 4

**Summary:**

The paper investigates the challenges and capabilities of Large-Language Models (LLMs) in the context of code generation from natural language instructions. A central issue identified is the difficulty LLMs face when generating extensive programs due to the limitations imposed by context window length. To tackle this problem, the authors introduce a system, L2MAC, which augments LLMs with an instruction registry and a file store. Through the orchestration of a control unit, L2MAC breaks down code generation into smaller, more manageable steps, and permits the modification of previously generated code via the file store. Evaluations presented in the paper indicate that L2MAC delivers superior performance in three system design tasks compared to two established benchmarks, GPT-4 and AutoGPT.

**Strengths:**

1. **Relatively Novel Approach**: The paper presents a novel idea of employing a control unit, instruction registry, and a file store to enhance LLMs. Although the individual components have been introduced in prior work (planning, test case generation, using external tools, refining with code execution feedback), the application of these to a stored-program computer in this context seems to be a fresh approach.

2. **Detailed Descriptions**: The paper provides a thorough description of all the crucial modules. The inclusion of example prompts and turn-by-turn actions in the appendix is an added advantage, as it supports better comprehension and reproducibility.

3. **Addressing Key Challenges**: The paper tackles an important and complex problem - the generation of comprehensive and cohesive programs. It makes significant observations, such as emphasizing the need to enable LLMs to revise previously-generated code, which contributes to the understanding of the problem.

**Weaknesses:**

1. **Questionable Evaluation Metrics**: The paper employs several evaluation metrics that are based on LLMs rather than ground truths like human-written test cases. This approach raises concerns about the representation of these metrics in terms of code quality. For instance, 'Features %' is determined by a GPT-4 call and not by running and testing the code. Similarly, 'Tests Passed' is based on the LLM-generated test cases, which may not accurately reflect test coverage or code quality. The paper could improve its evaluation by incorporating well-established metrics, such as human-written test cases, and applying a consistent set of tests across different methods. [partially addressed in the rebuttal].

2. **Overclaiming**: Although the paper introduces an intriguing concept, it also overstates several claims. For example, it suggests that L2MAC could enable LLMs to generate virtually unbounded code, but in practice, it is still limited by the context window when breaking down tasks and reading/writing code files. Moreover, while the paper claims that L2MAC can generate large codebases, the evaluation only presents a modest code length (300-400 LOCs).

3. **Weak Baselines**: The paper fails to incorporate more state-of-the-art code generation baselines like CodeT (Chen et al., 2022), Self-refine (Madaan et al., 2023), and Reflexion (Shinn et al., 2023). The use of GPT-4 alone as a baseline appears to be a strawman argument, while AutoGPT is not renowned for superior code generation capability in any popular benchmark, such as HumanEval or MBPP [addressed in the rebuttal].

**Questions:**

1. Have you considered incorporating ground truths, such as human-written test cases, into your evaluation metrics? Were there any challenges in doing so? Additionally, have you conducted any evaluation using more established metrics, and if so, could you share the results?

2. Might the inclusion of more sophisticated code generation models such as CodeT, Self-refine, and Reflexion have provided a broader comparison of L2MAC's performance? If this is the case, could you possibly incorporate some of these comparative results in the paper?

---

> ### Author Response · Authors · 2023-11-16
> **Response to Reviewer v4fH [Part 1/4]**
>
> Thank you for your thoughtful comments and suggestions! We are glad you enjoyed the relevance of the problem of program cohesiveness that L2MAC tackles, the novelty of our approach through the stored-program computer, and the detailed descriptions. Please find our answers below and the corresponding updates to the revised submission.
>
> * (A) Additional Evaluation Metrics
>   * (A.1) Ground-truth Human Expert Validation Justifies the Use of Feature %
>   * (A.2) Extending Tests with Code Coverage
>   * (A.3) Human-written Test Cases to Measure the Number of Features Implemented
> * (B) Refining Claims
>   * (B.1) "extensive" Now Replaces "unbounded"
>   * (B.2) Additional Experiment Task Demonstrating Code-L2MAC Can Generate 1,000+ Lines of Code
> * (C) Three New State-of-the-art Code Generation Baselines
>
> ---
>
> ### **(A) Additional Evaluation Metrics**
>
> We agree with the reviewer that the choice of evaluation metrics is vital to provide a fair comparison with the baselines. Although there is precedent for using LLMs as a proxy for human evaluation (Chiang et al., 2023), we agree that it is desirable to get a fuller picture of what they represent and ideally validate such metrics by comparing how they perform against ground truth validation methods.
>
> **(A.1) Ground-truth Human Expert Validation Justifies the Use of Feature %**
>
> To that end, we **hired two professional software engineers** as _human experts_, separate from the authors of this work, to perform code reviews of the generated code bases for each method against the user-requested task feature checklist, counting only features that they verified were correctly and fully implemented. We regard the resulting metric, labeled **Human Expert Features %**, as the ground truth.
>
> We tabulate (in table 5 below) this metric across three random seed runs. We highlight two conclusions.
>
> - Code-L2MAC significantly outperforms other baselines based on **Human Expert Features %.**
> - The human and LLM counterparts, **Human Expert Features %** and **Features %,** strongly correlate ($\rho=0.976$), thereby establishing **Features %** as a good proxy for the ground truth. This validates our usage of **Features %** as a scalable and cost-effective way to evaluate the amount of features implemented in code bases from new method-task pairs. This conclusion aligns with existing literature on using LLMs as a proxy for human evaluators (Chiang et al., 2023).
>
>
>
> **Table 5.**
> | | URL Shortener App | Online Social Media App | Online Chat App |
> |-------|---------------------------------------|---------------------------------------|---------------------------------------|
> | Method | Human Expert Features % | Features % | Human Expert Features % | Features % | Human Expert Features % | Features % |
> | | ↑ | ↑ | ↑ | ↑ | ↑ | ↑ |
> | GPT4 | 31.4 | 53.6 | 11.1 | 19.5 | 10 | 11 |
> | AutoGPT | 15.7 | 25.3 | 6.35 | 33.3 | 15 | 23.1 |
> | Code-L2MAC | **78.4** | **91.6** | **61.9** | **82.4** | **60** | **59.4** |
>
> **UPDATE:** We now include the human expert validation results and an adaptation of the above discussion in an additional **(new) appendix K** labeled "Human Expert Validation of Features Implemented Percentage Metric."
>
> ---
>
> **(A.2) Extending Tests with Code Coverage**
>
> To get more insight into the implications of the self-generated "Tests passed" metric, we use the standard test suite quality metric of code coverage percentage of the unit tests (Miller & Maloney, 1963), and similarly observe Code-L2MAC has a high code coverage percentage despite its substantially larger amount of generated lines of code. As shown in table 4, in the global response **R2**, Code-L2MAC achieves an average coverage percentage of 93.93% across three tasks.
>
> **UPDATE:** We now include this new evaluation metric for all the new tasks and baselines in a **(new) appendix J** and include this metric in our evaluation metrics in appendix G.
>
> ---

---

> ### Author Response · Authors · 2023-11-16
> **Response to Reviewer v4fH [Part 2/4]**
>
> **(A.3) Human-written Test Cases to Measure the Number of Features Implemented**
>
> We also explored using a priori hand-written test cases that are consistent across the different methods. We implemented this metric and present the results in table 6 below, which shows it is correlated to our proposed main evaluation metric of **Features %.**
>
> It is relevant to discuss some challenges this metric presents:
>
> 1. **Handwritten test cases assume and impose a known interface or a stringent implementation structure**. In small code snippet tasks, such as those in HumanEval (Chen et al., 2021) or MBPP (Austin et al., 2021), an explicitly defined function interface is explicitly presented to the LLM, and the LLM only responds with the code for that function body. In this situation, handwritten test cases can assume the pre-defined implementation structure. However, in code base generation tasks defined through feature requirements, there is a freedom about the segmentation into components, modules, or classes, which must be appropriately determined by the code generation method. For example, allowing users to register an account can be achieved with many code implementations. By specifying _tests, we filter this ambiguity into a given implementation approach,_ and we _cannot_ account for _all other possible code implementation_ approaches to implement a functional feature correctly.
> 2. **Requires expert-crafted task-specific test cases a priori.** This hinders the scalability of this approach.
>
> We added these handwritten test cases into each method's context window $C^t$ throughout all generation stages. Once the method generated a code base, we then randomly changed the hand-written test case parameters to still be the same test, just with different test parameters, to avoid the method memorizing the test examples, e.g., changing the initial user_ids to a random value. Since all methods often generated a code base that did not precisely match the implementation of the test cases while still having a code base that would conceptually pass the test purpose, we used GPT4 to port the test cases to each specific code base implementation. We term the proportion of such tests that pass human-written tests as **HT %**, the percentage of human tests that pass for a given code base. Table 6, which is computed over five random seed runs, shows that this metric correlates to our **Feature %** evaluation metric ($\rho=0.695$), which further provides empirical evidence for such a metric.
>
> **Table 6.**
> | Method | Features % | HT % | # Errors | LOC | Tests Passed | Cov % |
> |------------|------------------|------------------|------------------|---------------|--------------------|-------------------|
> | | ↑ | ↑ | ↓ | | ↑ | ↑ |
> | GPT4 | 25±79.6 | 0±0 | 3.75±10.9 | 134±19.9 | 6.75±7.16 | 80.5±14.5 |
> | CodeT | 13.2±42.1 | 11.1±35.4 | 0±0 | 126±14.4 | 7.75±3.98 | 86.8±4.57 |
> | Self-Refine| 30.6±30.7 | 33.3±41.4 | 0.2±0.555 | 140±9.83 | 9±0 | 74.6±8.85 |
> | Reflexion | 30.9±20.8 | 33.3±14.4 | 0±0 | 84.5±33.9 | 3.5±0.919 | 96.5±5.88 |
> | Code-L2MAC | **76.5±33.3** | **41.7±54.7** | **0±0** | **286±172** | **10±9.09** | **83±8.72** |
>
> **UPDATE:** We have now included an extended version of this discussion in a **(new) appendix L** entitled "Challenges and the Evaluation of Human-written Test-cases."
>
> ---
>
> ### **(B) Refining Claims**
>
> Based on your comments, we (1) refined our claims of adapting any mention of Code-L2MAC can generate _unbounded_ output to Code-L2MAC can generate _extensive_ output, and (2) provide an additional experiment task demonstrating Code-L2MAC can generate 1,000+ lines of code.
>
> **(B.1) "extensive" Now Replaces "unbounded"**
>
> We thank you for spotting the source of confusion that our use of "virtually unbounded" represents. We agree that this can be misleading and revised the use of this expression to remove any trace of such a claim.
>
> **UPDATE:** We will change the title and abstract to change the word "unbounded" to "extensive" and have done this throughout the paper. We kindly note that at the rebuttal stage, we cannot change the title and the abstract. However, we will make this change during the camera-ready revision.
>
> ---

---

> ### Author Response · Authors · 2023-11-16
> **Response to Reviewer v4fH [Part 3/4]**
>
> **(B.2) Additional Experiment Task Demonstrating Code-L2MAC Can Generate 1,000+ Lines of Code**
>
> By removing the restrictions we imposed upon Code-L2MAC to economize API-calls, such as limiting the number of instructions to 10, we get a variation we term Code-L2MAC-Large that we tested on the Online Chat application task where it reached reached over 1,000+ LOCs (5x the LOC of AutoGPT, the next highest) as shown in table 7 below.
>
> **Table 7.**
> | Method | Features % | # Errors | LOC | Tests Passed |
> |-------------------|-----------------|-----------------|--------------------|------------------|
> | | ↑ | ↓ | | ↑ |
> | Code-L2MAC-Large | **53.3±19** | **0.333±1.43** | **1,030±40.8** | **5.67±13.7** |
>
> **UPDATE** : We now discuss Code-L2MAC-Large and this experiment as an additional **(new) appendix M**, entitled "Generating 1,000+ Lines of Code with Code-L2MAC".
>
> ---
>
> ### **(C) Three New State-of-the-art Code Generation Baselines**
>
> We agree with the importance of considering these three methods and thus included them as **three new baselines** that represent state-of-the-art reflecting code generation methods: **CodeT** (Chen et al., 2022), **Self-Refine** (Madaan et al., 2023), and **Reflexion** (Shinn et al., 2023).
>
> We performed a complete re-run of these and the previous methods across all tasks, including three new ones; see the general response **(R2)**. We made the comparison fairer by providing these baselines with the same tools that Code-L2MAC uses; we provide the implementation details and hyperparameters in a newly expanded appendix F.
>
> These three new reflecting baselines are now included in the updated main experimental results table 2, also shown below. Code-L2MAC still fully implements the highest percentage of user-specified feature requirements across all tasks, where its code has minimal syntactical errors and a high number of self-generated unit tests. Therefore, Code-L2MAC is state-of-the-art for completing these system design large code generation benchmark tasks.
>
> **Table 2.** (Main experimental results averaged over ten random seeds)
>
> **URL Shortener App**
> | Method | Features % | # Errors | LOC | Tests Passed |
> | ------------ | -------------------- | ------------- | -------------- | ----------------- |
> | | ↑ | ↓ | | ↑ |
> | GPT4 | 53.6±10.5 | 0±0 | 119±21.1 | 2.56±0.95 |
> | CodeT | 52.9±6.74 | 0.05±0.105 | 110±11.8 | 3.6±0.513 |
> | Self-Refine | 47.9±8.53 | 0.05±0.105 | 124±15.7 | 3.65±1.15 |
> | Reflexion | 38.8±6.02 | 0.1±0.209 | 96.2±9.11 | 2.35±0.631 |
> | AutoGPT | 25.3±19.6 | 0±0 | 136±41.9 | 3.3±1.91 |
> | Code-L2MAC | **91.6±8.22** | **0±0** | **330±47.6** | **14±6.71** |
>
> **Online Social Media App**
> | Method | Features % | # Errors | LOC | Tests Passed |
> | ------------ | -------------------- | ------------- | -------------- | ----------------- |
> | | ↑ | ↓ | | ↑ |
> | GPT4 | 19.5±8.28 | 4.09±3.32 | 116±31.5 | 0.818±0.785 |
> | CodeT | 19.5±5.19 | 0.4±0.603 | 106±17.7 | 2.6±1.76 |
> | Self-Refine | 16.4±2.62 | 0.938±0.714 | 110±19.6 | 1.81±0.938 |
> | Reflexion | 15.2±8.05 | 2.53±1.69 | 122±24 | 1.33±2.44 |
> | AutoGPT | 33.3±18 | 0.6±0.369 | 148±35.5 | 3±2.86 |
> | Code-L2MAC | **82.4±14.6** | **0±0** | **395±52.9** | **18.3±6.8** |
>
> **Online Chat App**
> | Method | Features % | # Errors | LOC | Tests Passed |
> | ------------ | -------------------- | ------------- | -------------- | ----------------- |
> | | ↑ | ↓ | | ↑ |
> | GPT4 | 11±2.26 | 0.3±0.346 | 127±24.1 | 1.2±1 |
> | CodeT | 10.5±4.61 | 0±0 | 91.6±25.9 | 3.32±1.57 |
> | Self-Refine | 14.2±4.19 | 0.211±0.304 | 111±13.8 | 1.42±0.927 |
> | Reflexion | 10.2±3.08 | 0±0 | 76±6.88 | 2.85±0.822 |
> | AutoGPT | 23.1±11.8 | 1.85±2.47 | 220±65.8 | 3.08±3.34 |
> | Code-L2MAC | **59.4±25.9** | **0±0** | **374±123** | **18.8±9.11** |
>
> **Update:** These three reflecting LLM baselines have been included in the main experimental results in the paper in table 2.
>
> ---

---

> ### Author Response · Authors · 2023-11-16
> **Response to Reviewer v4fH [Part 4/4]**
>
> **References:**
>
> - Chiang, Cheng-Han, and Hung-yi Lee. "Can Large Language Models Be an Alternative to Human Evaluations?." _arXiv preprint arXiv:2305.01937_ (2023).
> - Joan C Miller and Clifford J Maloney. Systematic mistake analysis of digital computer programs. Communications of the ACM, 6(2):58–63, 1963.
> - Mark Chen, Jerry Tworek, Heewoo Jun, Qiming Yuan, Henrique Ponde de Oliveira Pinto, Jared Kaplan, Harri Edwards, Yuri Burda, Nicholas Joseph, Greg Brockman, et al. Evaluating large language models trained on code. arXiv preprint arXiv:2107.03374, 2021.
> - Jacob Austin, Augustus Odena, Maxwell Nye, Maarten Bosma, Henryk Michalewski, David Dohan, Ellen Jiang, Carrie Cai, Michael Terry, Quoc Le, et al. Program synthesis with large language models. arXiv preprint arXiv:2108.07732, 2021.
> - Bei Chen, Fengji Zhang, Anh Nguyen, Daoguang Zan, Zeqi Lin, Jian-Guang Lou, and Weizhu Chen. Codet: Code generation with generated tests. arXiv preprint arXiv:2207.10397, 2022.
> - Aman Madaan, Niket Tandon, Prakhar Gupta, Skyler Hallinan, Luyu Gao, Sarah Wiegreffe, Uri Alon, Nouha Dziri, Shrimai Prabhumoye, Yiming Yang, et al. Self-refine: Iterative refinement with self-feedback. arXiv preprint arXiv:2303.17651, 2023.
> - Noah Shinn, Federico Cassano, Ashwin Gopinath, Karthik R Narasimhan, and Shunyu Yao. Reflexion: Language agents with verbal reinforcement learning. In Thirty-seventh Conference on Neural Information Processing Systems, 2023.

---

> ### Comment · Reviewer_v4fH · 2023-11-20
>
> Thank you for your detailed answer. The added results, including an expert's review, ground-truth test cases, and extra uses, greatly improve the evaluation and tackle most of my worries. As a result, I've raised my score by one level.
>
> However, I still have some worries about the paper's claims. I would like to see more careful claims and a discussion of limits. For example, even the extra tests with 1000 lines of code (LOC) may not be enough to count as a large code base. Also, the extra results with real-world test cases are now part of the input for creating code, which is not the usual way to test code quality.
>
> While I get that the proposed solution might not work well without seeing the test cases, I think the paper should make this method's limits clear. It would be even better if the authors could show results without giving the ground-truth test cases to the Language Models (LLMs).

---

> ### Author Response · Authors · 2023-11-21
> **Response to Reviewer v4fH Comment [Part 1/2]**
>
> We are glad you appreciated our improvements to the paper. Thank you again for your valuable input throughout the review process, and we are grateful for your increased score by one level.
> Once again, the issues you raised drove us toward further experiments that have proven insightful and beneficial for the paper. We organize our response as follows:
>
> * (A) Code-L2MAC Lines of Code Limits
> * (B) Discussion on Human-written Test Cases and Results Without Test Cases Given to the Methods
>   * (B.1) Discussion on Human-written Test Cases
>   * (B.2) Results Without Human-written Test Cases Given to the Methods
>
> ---
>
> ### **(A) Code-L2MAC Lines of Code Limits**
>
> We agree that there are limits to the number of lines of code (LOC) that Code-L2MAC can write effectively in the current implementation. Both are discussed within the paper (section 4, footnote 7, and section 7), and both can be readily addressed, providing fertile ground for future work. These are:
>
> * _Code files should be smaller than the context window constraint $c$_. Reading a file involves outputting its whole content into the LLM’s context, and writing a file implies rewriting it. This means that to avoid falling out of context, the maximum length for a file is the LLM’s context window size, but in fact, it is undesirable to have a file of length more than half the LLM’s context window size since this will imply that the LLM cannot modify it without falling out of context. This limitation can be overcome by endowing the LLM with the capacity to selectively read and write parts of the file (e.g., function names and headers) in the same spirit as diffs are conducted in git. We discuss this and its implications for the method’s efficiency in Future Work (app. I).
> * _All the code file paths are listed in the context window $C^t$_. Therefore, the maximum number of file paths (e.g. [‘app.py,’ ‘test_app.py,’ …]) listed Code-L2MAC can have in memory is strictly less than the context length. This can be readily solved by allowing the LLM only to list the file paths inside a folder and the possibility to navigate inside a sub-folder. In such a case, the constraint would happen only regarding the degree in the file tree, but we could theoretically have infinite depth in the file store.
>
> Although there are no definitive reasons why, with these improvements, Code-L2MAC would have an inherent LOC-wise upper bound, we agree that having a more precise notion of what “large” implies would be desirable. Indeed, the inherent limitations on the LLM’s capabilities could implicitly impose an LOC-wise upper limit to Code-L2MAC in practice.
>
> To that end, and to address your concern, we follow an experimental version of the “Proof by Induction” reasoning. Our original experiments show that Code-L2MAC can handle the base case: “Starting a code base from scratch.” We now run three new experiments/tasks corresponding to the inductive step: “Given an extensive code base, can Code-L2MAC correctly enlarge it?”
>
> We take three existing code bases between 87,000 and 165,000 LOCs and request that Code-L2MAC extend its functionalities with a new feature. Experiments show that Code-L2MAC is also capable of handling this task, as the following table 10 shows:
>
> **Table 10**
> | Method     | Features % (Dynamic Dashboard App) | LOC (Dynamic Dashboard App) | Features % (Community Forum App) | LOC (Community Forum App) | Features % (Data Exploration App) | LOC (Data Exploration App) |
> |------------|------------------------------------|-----------------------------|-----------------------------------|---------------------------|------------------------------------|----------------------------|
> | Code-L2MAC | 80±30.2                             | 165,811.6±43.8              | 70±34.6                            | 88,878.4±9.68             | 100±0                               | 87,044.5±18.5              |
>
> **UPDATE**: We incorporate a version of the discussion on Code-L2MAC limits into the existing future work appendix I. We also added the new experiments in a **(new) appendix O**, entitled ***“Additional Tasks on Implementing a New Feature in an Existing Large Code Base with Code-L2MAC of up to 165,000 LOCs”***, which includes their experimental setup details.
>
> ---

---

> ### Author Response · Authors · 2023-11-21
> **Response to Reviewer v4fH Comment [Part 2/2]**
>
> ### **(B) Discussion on Human-written Test Cases and Results Without Test Cases Given to the Methods**
>
> We kindly note that this evaluation method is only used in the set of experiments described in appendix L, labeled “Challenges and the Evaluation of Human-written Test-cases” (“A.3. Human-written Test Cases to Measure the Number of Features Implemented” in our first response). In contrast, the ground-truth human evaluation, coverage, and the rest of the experiments do not provide the LLM with the tests that will be used for the evaluation.
>
> Nonetheless, your remarks are relevant, and we divide them into two parts.
>
> ---
>
> **(B.1) Discussion on Human-written Test Cases**
>
> Indeed, as you correctly point out and as we tried to convey in our discussion on the challenges of using “a priori human-written test cases,” providing the test cases to the LLMs is not ideal since it hints at how to design the implementation.
>
> **UPDATE**: We thank you for highlighting the need for a more explicit discussion of the limitations of this evaluation method (which was included during the rebuttal in a new appendix L). We updated appendix L correspondingly to clearly state the limitations mentioned above on the results obtained when including the test cases as information for the method.
>
> ---
>
> **(B.2) Results Without Human-written Test Cases Given to the Methods**
>
> We thus agree that incorporating a re-run of the experiments in appendix L without allowing the method to see the test-cases would be insightful.
>
> Expanding on the original response of (A.3) and (B.1) above, and following your suggestion, we performed a complete re-run for all the baselines for this setting, where the methods did **not** have the human-written test-cases as part of their input (i.e., excluding the test-cases from the context window $C^t$), throughout all stages of generation. We present these new results in table 11. We observe that the **HT \%** metric correlates to our **Feature \%** evaluation metric ($\rho=0.928$), which further provides empirical evidence for such a metric.
>
> **Table 11**
> | Method      | Features % | HT % | # Errors | LOC | Tests Passed | Cov % |
> |-------------|------------|------|----------|-----|--------------|-------|
> | | ↑ | ↑ | ↓ | | ↑ | ↑ |
> | GPT4        | 37.6±13.3  | 20±29.9 | 0±0    | 94.2±15.4 | 2±1.76       | 85.6±18.1 |
> | CodeT       | 47.1±10.3  | 42.2±31.5 | 0±0  | 98±19     | 4.8±3.45     | 91.8±7.15 |
> | Self-Refine | 50.6±16.8  | 46.7±49.2 | 0±0  | 109±12.9  | 3.4±2.26     | 92±1.96   |
> | Reflexion   | 55.3±18.3  | 37.8±33.2 | 0.6±1.67 | 124±31.3 | 3.4±2.99   | 87.6±12.8 |
> | Code-L2MAC  | 89.4±12    | 71.1±50.3 | 0±0  | 283±100   | 8.6±9.52     | 77.2±53.7 |
>
>
> **UPDATE**: As you suggested, we have re-run these same experiments without providing the test cases to the LLM and incorporated these new results in an **(newly updated) appendix L**.
>
> ---
>
> Should any uncertainties linger, we invite you to share them with us before the author discussion period concludes. Your continued engagement is deeply appreciated, and we are at your disposal for any further clarifications. Thank you!

---

### Official Review · Reviewer_mgbF · 2023-11-09

**Soundness:** 4 excellent
**Presentation:** 3 good
**Contribution:** 3 good
**Rating:** 8
**Confidence:** 4

**Summary:**

Authors propose L2MAC tool that implements automatic generation of large code bases using LLMs. Authors implement context management to keep relevant context and summarize/compress previous context to keep it within context size bounds. They implement approach to read and write file data across all created files. They also implement generated output checker that runs static analysis and unit tests and processes error messages. Authors evaluate their tool on a benchmark set and demonstrate improved results compared to state-of-the-art baselines.

**Strengths:**

- Structured framework for LLM-based computation that can deal with limited context, file input/output and output evaluation and testing.
- Context handling that preserves information needed for the tasks and limits context to the context size
- Read and write implementation for files generated during subtasks. Demonstrated capabilities to write, then read and update files.
- Strongly improved results on benchmark tasks compared to strong baseline models/tools.

**Weaknesses:**

- File read/write implementation details are not clear. Please explain how your system decides what files to write, read, and update and how this is different from previous systems that did not have this functionality.
- Benchmark set is not described. It is not clear if the benchmarks are representative of large code base creation tasks. Evaluation is done on only 3 tasks. The number of tasks should be increased to show the versatility and that the results are not outliers.

Minor comments:
- Stored-program computer subsection does not seem to contribute much to the paper. Probably shorten or remove.
- Code-LLMatic is used instead of CodeL2MAC in couple places. Did not update old name?
- Footnotes on page 4 mostly do not add to the narrative. Remove?
- Figure 3 is placed after Figure 4 for some reason.

**Questions:**

- Figure 4 (c) - could you explain why for Code-L2MAC the figure shows the number of tests passed, while for other two tools it shows the number of tests failed? Are these numbers comparable? If so, why and how?
- Does the "checking of generated output" part contain novel contributions? It seems to me that other tools and approaches also have feedback loops where code is regenerated on errors. It would be good if this was clarified. (This is possibly a minor weakness).
- If I understand correctly, unlike autonomous tools that can add additional subtasks dynamically L2MAC creates a subtask list once at the beginning and does not subdivide or add additional tasks later. It seems that authors consider this to be a strength, because L2MAC will not go into hallucination subtask loop. However, this could also be a drawback since L2MAC may create subtask that needs to be subdivided later and it will not be able to do so. This should be evaluated. This also connects to the weakness of benchmark set that only has 3 benchmarks: perhaps other benchmarks would show the strength or weakness of this approach.

---

> ### Author Response · Authors · 2023-11-16
> **Response to Reviewer mgbF [Part 1/3]**
>
> Thanks for your thoughtful comments and suggestions! Please find our answers as follows, along with corresponding updates to the revised submission:
> * (A) Read/Write Implementation Details
> * (B) Extending and Describing the Benchmark
> * (C) Typos
> * (D) Questions
>   * (D.1) Figure 4 (c) Interpretation
>   * (D.2) Reflection in Code-L2MAC
>   * (D.3) Replanning
>
> ---
> ### **(A) Read / Write Implementation Details**
>
> We agree that a detailed description individually covering the read/write implementation would be valuable. The description in Section 4, appendix C, and appendix F.1 is entangled with the description of other components, which can induce confusion.
>
> The LLM interfaces with the control unit (CU) through calling functions, and thus, the read and write implementation can be fully described through a discussion of the functions that are provided to Code-L2MAC to that end, which are `read_files` and `write_files.`
>
> It is worth noting that the control unit always exposes the LLM to the _list of file paths_ that are already part of the code base (e.g., "file paths: ['app.py', 'tests/test\_app.py']"). The name of directories and files (i.e., file path) provides a semantical _prior_ for the _order_ in which the LLM will read each file, depending on the functionality it is looking for (among other elements, including always the current instruction $\mathcal{I}\^t$, and possibly previous dialog turn outputs and error responses).
>
> Note that if the LLM initially reads a file that does not contain what it was looking for or the desired functionality is spread across multiple files, the LLM can continue reading other files. Recall that when the context window is full, the CU prompts the LLM to summarize the relevant parts of the context window for the current instruction $\mathcal{I}\^t$. In this case, it could summarize the relevant files it needs to read if the functionality is spread across multiple files.
>
> Assume the LLM has already determined the name of the file `file_path` that it wants to read. Let us zoom into the functions (tools) provided to the LLM to signal intentions to the CU.
>
> - **`read_files(file_path)`** requests the CU to output into the context window $C^t$ the contents of `file_path`.
> - **`write_files(file_path, content)`** requests the CU to create or overwrite `file_path` and populate it with `content`.
>
> As the empirical results demonstrate, these functions perform well as components of Code-L2MAC. The future work appendix (app. I) discusses how these functions could be optimized.
>
> To address the remainder of your question, Code-L2MAC's reading and writing functionality is different from previous memory-augmented LLMs (we refer to the extended related work for more details, app. D) as these can be categorized as:
>
> 1. **Append only memory**: These methods explicitly extend the implicit knowledge of LLMs through an external corpus (Zhong et al. 2022b) to facilitate long conversational or document summarization tasks (Liang et al., 2023).
> 2. **Key-value memory**: These methods interface the LLM with a dictionary or a database (Hu et al., 2023), which does not apply to our automatic coding tasks.
>
> **UPDATE**: In full agreement with your comment about a lack of a central and thorough description of the read/write components, we provide a **(new) appendix C.1** in line with the above description to this end, labeled _"Read / Write Implementation Details"_.
>
>
> ---
> ### **(B) Extending and Describing the Benchmark**
>
> We agree that having a larger suite of experiments would be desirable to benchmark our method. To that end, as described in (**R2)** in the global response, we _extended_ the _benchmark_ with _three new_ programming tasks: coding a recipe platform, a financial tracker, and an event planner application. The results obtained on these tasks align with those in the original tasks; Code-L2MAC consistently outperforms other methods. See **(R2)** in the global response for the precise results.
>
> Our choice of tasks aims at diversity, and we rerun each method-task pair on ten seeds to provide meaningful confidence intervals in our results. The low variance of inter-task performance between tasks for each method and the consistent SOTA performance of Code-L2MAC suggests that these results are solid and representative of an extensive suite of tasks. We provide a full description of the benchmarks in appendices E and F.
>
> **UPDATE**: Per the reviewer's comments, we **duplicated** the number of **tasks.** We included the **results in a (new) appendix J** entitled "Additional Diverse Programming Code Generation Tasks" and extended the benchmark description accordingly in appendix E.
>
> ---
> ### **(C) Typos**
>
> Thank you for spotting the typos and the misordering of the figures.
>
> **UPDATE**: These issues have been solved now.

---

> ### Author Response · Authors · 2023-11-16
> **Response to Reviewer mgbF [Part 2/3]**
>
> ### **(D) Questions**
> ---
> **(D.1) Figure 4 (c) Interpretation**
>
> In Figure 4 (c), we show the tests that failed in red on top of the ones that passed in green for all methods. As you point out, Code-L2MAC accumulates very few errors. This is because when a new test fails or a previous one breaks, Code-L2MAC addresses the failures to _recover consistency_ in the code. In contrast, previous methods leave failures largely unattended, meaning failures accumulate as the code base grows and the number of passed tests stays low.
>
> **UPDATE**: We updated the caption to clarify this source of confusion by specifying they are **"stacked histograms"**. The above explanation of this result can be found in the last paragraph of section 6.
>
> ---
>
> **(D.2) Reflection in Code-L2MAC**
>
> A unique characteristic of our checks is that they are not only aimed at validating the immediate output (e.g., existing reflecting LLM methods) but rather at imposing _consistency_ on the file store/memory as a _whole_. This implies that while in previous settings, a failing test would require a change in the LLM's output, in Code-L2MAC, it can motivate revisiting and refactoring a pre-existing component in the file store to enhance its functionality and accommodate new use cases.
>
> Specifically, we find the following differences between Code-L2MAC and reflecting LLM methods, such as Self-Refine and Reflexion to be:
>
> - Self-Refine refines the _most recent_ output from the LLM and _Reflexion_ the _recent outputs_ that are only within the LLM's current context window $C^t$; whereas Code-L2MAC can refine the entire file store, encompassing all previous outputs from the LLM—allowing it to fix or improve earlier outputs from multiple instructions back, outside its context window, which are now erroring due to a new code implementation change. This enables Code-L2MAC to generate long, consistent, and interrelated code structures.
> - Self-Refine and Reflexion are constrained to operating on an output within the context window constraint $c$, whereas Code-L2MAC can manage a total output greater than the context window constraint through the L2MAC framework.
>
> **UPDATE**: Allow us to kindly reiterate that our central contribution is introducing the first practical LLM-based stored-program computer. As part of that, as you point out, our implementation incorporates concepts of refinement that should be properly contrasted with previous work. We previously did so in the _related work_ (section 5) and the extended related work (appendix D). As a consequence of this question, we have improved this comparison by including existing reflecting LLM methods as baselines against Code-L2MAC (see R2 in the global response), and we also improved the extended related work by including a version of this discussion in the (app. D).
>
> ---

---

> ### Author Response · Authors · 2023-11-16
> **Response to Reviewer mgbF [Part 3/3]**
>
> **(D.3) Replanning**
>
> We appreciate you bringing this up since we agree that this item needs clarification. First, we agree that allowing for (1) replanning if the original instructions prove to be an ineffective plan and (2) subdividing an instruction if needed would increase the flexibility of L2MAC. Indeed, the last sentence in Section 7 (Conclusion & Future Work) and the first item in appendix I (Future work) defer this extension as exciting future work.
>
> - **Section Conclusion & Future Work** : "... other aspects such as complex instruction flows, re-programming the instructions, … pose exciting open directions for future work."
> - **Appendix I: Future work** : "... another possible way to avoid out-of-context errors is to dynamically further break a sub-task down into smaller sub-tasks and modify the existing stored prompt-program instructions to include these …"
>
> Given the already successful empirical results, we deem these improvements not central to this paper; however, both provide fertile ground for future work.
>
> Nonetheless, we traced the concern of your question back to the following sentence in the Related Work (when discussing Autonomous Agent LLMs) _"When coding, these agents reprogram and reformulate their step plan at runtime, resulting in frequent deviations from the task and stalls in infinite loops (Wang et al., 2023; Sato et al., 2023)"_ which might be interpreted to suggest this is an inherent flaw in any method that admits reprogramming. However, the message we want to convey is not that the problem lies in the possibility of replanning or subdividing steps but instead in trusting the requirements of the task to be stored in context (as opposed to managed by both the Memory and a CU) while allowing for replanning. This combination can lead, as we show in the experiments (in particular, Figure 4.a.), to the original requirements being altered by the LLM, leading to a misaligned or potentially altogether different task. As mentioned, we observe this happening throughout both the original and the new tasks.
>
> **UPDATE**: We have updated the problematic sentence in the Related Work to "When coding, these agents reprogram and reformulate their step plan at runtime without safeguarding the original intentions, resulting in frequent deviations from the task.". Additionally, we have expanded our discussion in Future Work (app. I) regarding the exciting future work that might spark from exploring the possibility of replanning and recursively breaking down sub-tasks.
>
> ---
>
> **References:**
>
> - Zexuan Zhong, Tao Lei, and Danqi Chen. Training language models with memory augmentation. In Proceedings of the 2022 Conference on Empirical Methods in Natural Language Processing, pp. 5657–5673, 2022. 13
> - Xinnian Liang, Bing Wang, Hui Huang, Shuangzhi Wu, Peihao Wu, Lu Lu, Zejun Ma, and Zhoujun Li. Unleashing infinite-length input capacity for large-scale language models with self-controlled memory system. arXiv preprint arXiv:2304.13343, 2023.
> - Chenxu Hu, Jie Fu, Chenzhuang Du, Simian Luo, Junbo Zhao, and Hang Zhao. Chatdb: Augmenting llms with databases as their symbolic memory. arXiv preprint arXiv:2306.03901, 2023.
> - Aman Madaan, Niket Tandon, Prakhar Gupta, Skyler Hallinan, Luyu Gao, Sarah Wiegreffe, Uri Alon, Nouha Dziri, Shrimai Prabhumoye, Yiming Yang, et al. Self-refine: Iterative refinement with self-feedback. arXiv preprint arXiv:2303.17651, 2023.
> - Noah Shinn, Federico Cassano, Ashwin Gopinath, Karthik R Narasimhan, and Shunyu Yao. Reflexion: Language agents with verbal reinforcement learning. In Thirty-seventh Conference on Neural Information Processing Systems, 2023.

---

> > ### Comment · Reviewer_mgbF · 2023-11-18
> > **Thank you for author comments and changes**
> >
> > Thanks to the authors for extensive comments and changes to the paper. These have addressed my questions.

---

> > > ### Author Response · Authors · 2023-11-19
> > > **Appreciation for Feedback**
> > >
> > > Thank you very much for your valuable feedback and for taking the time to review our work. We are pleased that you appreciate the extensive new experiments, explanations, and changes to the paper, which have ultimately improved the paper. Thank you once again!

---

### Author Response · Authors · 2023-11-16
**Global Response [Part 1/2]**

We thank all reviewers for your thoughtful comments and suggestions! We respond to each reviewer with an individual rebuttal and share all _the main new experimental results_ here. These are:

- **(R1)** Three **new baselines** : CodeT (Chen et al., 2022), Self-Refine (Madaan et al., 2023), and Reflexion (Shinn et al., 2023).
- **(R2) Doubled** the benchmark **tasks**. We included tasks distinct from the previous ones: programming a recipe application, an event planner application, and a financial tracking application.

----
### **(R1) Three New State-of-the-art Code Generation Baselines**

We include **three new baselines** of more recent state-of-the-art reflecting code generation methods: **CodeT** (Chen et al., 2022), **Self-Refine** (Madaan et al., 2023), and **Reflexion** (Shinn et al., 2023). We performed a complete re-run of all these for all tasks, including three new additional tasks; see response **(R2)**. We make these new reflecting baselines more competitive by providing them with the same tools that Code-L2MAC uses and provide the implementation details and hyperparameters in a newly expanded appendix F.

These three new reflecting baselines are now included in the updated main experimental results table 2, also shown below. Code-L2MAC still fully implements the highest percentage of user-specified feature requirements across all tasks, where its code has minimal syntactical errors and a high number of self-generated unit tests. Therefore, Code-L2MAC is state-of-the-art for completing these system design large code generation benchmark tasks.


**Table 2.** (Main experimental results averaged over ten random seeds)

**URL Shortener App**

| Method | Features % | # Errors | LOC | Tests Passed |
| ------------ | -------------------- | ------------- | -------------- | ----------------- |
| | ↑ | ↓ | | ↑ |
| GPT4 | 53.6±10.5 | 0±0 | 119±21.1 | 2.56±0.95 |
| CodeT | 52.9±6.74 | 0.05±0.105 | 110±11.8 | 3.6±0.513 |
| Self-Refine | 47.9±8.53 | 0.05±0.105 | 124±15.7 | 3.65±1.15 |
| Reflexion | 38.8±6.02 | 0.1±0.209 | 96.2±9.11 | 2.35±0.631 |
| AutoGPT | 25.3±19.6 | 0±0 | 136±41.9 | 3.3±1.91 |
| Code-L2MAC | **91.6±8.22** | **0±0** | **330±47.6** | **14±6.71** |

**Online Social Media App**
| Method | Features % | # Errors | LOC | Tests Passed |
| ------------ | -------------------- | ------------- | -------------- | ----------------- |
| | ↑ | ↓ | | ↑ |
| GPT4 | 19.5±8.28 | 4.09±3.32 | 116±31.5 | 0.818±0.785 |
| CodeT | 19.5±5.19 | 0.4±0.603 | 106±17.7 | 2.6±1.76 |
| Self-Refine | 16.4±2.62 | 0.938±0.714 | 110±19.6 | 1.81±0.938 |
| Reflexion | 15.2±8.05 | 2.53±1.69 | 122±24 | 1.33±2.44 |
| AutoGPT | 33.3±18 | 0.6±0.369 | 148±35.5 | 3±2.86 |
| Code-L2MAC | **82.4±14.6** | **0±0** | **395±52.9** | **18.3±6.8** |


**Online Chat App**
| Method | Features % | # Errors | LOC | Tests Passed |
| ------------ | -------------------- | ------------- | -------------- | ----------------- |
| | ↑ | ↓ | | ↑ |
| GPT4 | 11±2.26 | 0.3±0.346 | 127±24.1 | 1.2±1 |
| CodeT | 10.5±4.61 | 0±0 | 91.6±25.9 | 3.32±1.57 |
| Self-Refine | 14.2±4.19 | 0.211±0.304 | 111±13.8 | 1.42±0.927 |
| Reflexion | 10.2±3.08 | 0±0 | 76±6.88 | 2.85±0.822 |
| AutoGPT | 23.1±11.8 | 1.85±2.47 | 220±65.8 | 3.08±3.34 |
| Code-L2MAC | **59.4±25.9**| **0±0** | **374±123** | **18.8±9.11** |

**Update:** Three reflecting LLM baselines have been included in the main experimental results in the paper in table 2.

---

> ### Author Response · Authors · 2023-11-16
> **Global Response [Part 2/2]**
>
> ### **(R2) Doubled the Benchmark Tasks**
>
> We are also excited about extending the benchmark with three new, _additional, diverse programming code generation tasks_. These are a **recipe application** , an **event planner application,** and a **financial tracking application.** Each task consists of a user prompt of listed features to implement, and Code-L2MAC produces the entire code base from scratch. Code-L2MAC continues to fully implement the highest percentage of user-specified feature requirements across these new diverse tasks. At the same time, its code contains minimal syntactical errors and passes a high number of unit tests. Therefore, Code-L2MAC still achieves state-of-the-art for completing these system design large code generation benchmark tasks. Moreover, we also have implemented a new standard code quality metric of code coverage percentage of the unit tests (Miller & Maloney, 1963), labeled **Cov %** , and similarly observe Code-L2MAC also has a high code coverage percentage to its substantially larger amount of generated lines of code.
>
>
> **Table 4.** (Three new task experimental results averaged over 10 random seeds)
>
> **Recipe App**
> | Method | Features % | # Errors | LOC | Tests Passed | Cov % |
> | ------------ | -------------------- | ------------- | -------------- | ----------------- | -------------- |
> | | ↑ | ↓ | | ↑ | ↑ |
> | GPT4 | 21.6±2.12 | 0±0 | 107±6.62 | 3.15±0.38 | **97.5±0.376** |
> | CodeT | 20.5±4.86 | 0±0 | 96.5±13.8 | 3.05±0.879 | **97.8±0.523** |
> | Self-Refine | 26±3.45 | 0.1±0.209 | 149±27.4 | 2±1.97 | 76.2±7.83 |
> | Reflexion | 19±3.36 | 0.25±0.299 | 95.9±14.5 | 2.95±0.852 | 89.9±10.4 |
> | AutoGPT | 39.2±14.9 | 1.85±1.45 | 106±19.1 | 1.3±2.02 | 9.8±14.1 |
> | Code-L2MAC | **82±7.1** | **0±0** | **497±40.7** | **24.6±2.7** | 94.2±2.87 |
>
> **Event Planner App**
> | Method | Features % | # Errors | LOC | Tests Passed | Cov % |
> | ------------ | -------------------- | ------------- | -------------- | ----------------- | -------------- |
> | | ↑ | ↓ | | ↑ | ↑ |
> | GPT4 | 9.2±0.853 | 0.025±0.0506 | 74.6±4.12 | 1.75±0.395 | 88.7±5.5 |
> | CodeT | 11.2±1.12 | 0.05±0.105 | 75.2±8.77 | 2.45±0.704 | 92.5±10.2 |
> | Self-Refine | 14.5±2.94 | 0.15±0.171 | 118±20.1 | 3.9±1.9 | 76.7±15 |
> | Reflexion | 10±1.5 | 0±0 | 82±10.5 | 3±0.774 | **95.1±4.35** |
> | AutoGPT | 35.7±32.9 | 0±0 | 23.9±20.7 | 0±0 | 0±0 |
> | Code-L2MAC | **83±2.96** | **0±0** | **473±39.3** | **25.6±3.04** | **97.1±1.02** |
>
> **Financial Tracking App**
> | Method | Features % | # Errors | LOC | Tests Passed | Cov % |
> | ------------ | -------------------- | ------------- | -------------- | ----------------- | -------------- |
> | | ↑ | ↓ | | ↑ | ↑ |
> | GPT4 | 26.2±4.67 | 0.0513±0.104 | 80.5±8.52 | 2.13±0.422 | 93.1±3.17 |
> | CodeT | 21.4±3.25 | 0±0 | 65.9±6.93 | 2.25±0.368 | **97.9±0.209** |
> | Self-Refine | 23.6±2.45 | 0.25±0.299 | 87.2±8.24 | 0.55±0.514 | 76.7±9.97 |
> | Reflexion | 22.5±3.12 | 0.2±0.419 | 86.8±13.7 | 2.7±0.745 | 92.8±8.73 |
> | AutoGPT | 32.9±44.2 | 0±0 | 25±15.8 | 0±0 | 0±0 |
> | Code-L2MAC | **62±13.1** | **0±0** | **307±84.5** | **12±4.19** | 90.5±6.69 |
>
> **Update:** We now include these new benchmark tasks for all the baselines in a **(new) appendix J** entitled "Additional Diverse Programming Code Generation Tasks".
>
> ---
> ### **References:**
>
> - Bei Chen, Fengji Zhang, Anh Nguyen, Daoguang Zan, Zeqi Lin, Jian-Guang Lou, and Weizhu Chen. Codet: Code generation with generated tests. arXiv preprint arXiv:2207.10397, 2022.
> - Aman Madaan, Niket Tandon, Prakhar Gupta, Skyler Hallinan, Luyu Gao, Sarah Wiegreffe, Uri Alon, Nouha Dziri, Shrimai Prabhumoye, Yiming Yang, et al. Self-refine: Iterative refinement with self-feedback. arXiv preprint arXiv:2303.17651, 2023.
> - Noah Shinn, Federico Cassano, Ashwin Gopinath, Karthik R Narasimhan, and Shunyu Yao. Reflexion: Language agents with verbal reinforcement learning. In Thirty-seventh Conference on Neural Information Processing Systems, 2023.
>
> - Joan C Miller and Clifford J Maloney. Systematic mistake analysis of digital computer programs. Communications of the ACM, 6(2):58–63, 1963.

---

### Meta-Review · Area_Chair_Sv2q · 2023-12-08

**Metareview:**

All reviewers were convinced and voted accept but no one recommended higher awards.

The method fundamentally queries GPT-4 for its steps and thus still inherits its limitations on each query. Reviewers suspect "virtually unbounded code structures" is overclaiming and not sufficiently transparent due to this simple reason.

The paper showed impressive performance on headline evaluations for their own system design task, which takes a series of requirements and generate the actual implementation of a system (e.g. an url shorterner app). While an interesting and important task, the benchmark is non-standard, possibly not ground truth, and not established, so there is significant hesitancy in recommending higher acceptance level and award. While there are a number of baselines methods, little efforts were made to tailor them for the system design setup. Most of the baselines explore different aspects such as error correction or aggregations whereas the proposed method focuses on guiding the LM through the process. The authors also chose not to tackle any established tasks where benefits of the their method could be relevant.

**Justification For Why Not Higher Score:**

Evaluation could be more comprehensive and competitive

**Justification For Why Not Lower Score:**

Reviewers agree that this is an interesting method

---

### Decision · Program_Chairs · 2024-01-16

Accept (poster)